# Faster Training of Neural ODEs Using Gauß–Legendre Quadrature

**Alexander Norcliffe***  
*University of Cambridge*

*alin2@cam.ac.uk*

**Marc Peter Deisenroth**  
*University College London*

*m.deisenroth@ucl.ac.uk*

**Reviewed on OpenReview:** *https://openreview.net/forum?id=f0FSDAy1bU*

## Abstract

Neural ODEs demonstrate strong performance in generative and time-series modelling. However, training them via the adjoint method is slow compared to discrete models due to the requirement of numerically solving ODEs. To speed neural ODEs up, a common approach is to regularise the solutions. However, this approach may affect the expressivity of the model; when the trajectory itself matters, this is particularly important. In this paper, we propose an alternative way to speed up the training of neural ODEs. The key idea is to speed up the adjoint method by using Gauß–Legendre quadrature to solve integrals faster than ODE-based methods while remaining memory efficient. We also extend the idea to training SDEs using the Wong–Zakai theorem, by training a corresponding ODE and transferring the parameters. Our approach leads to faster training of neural ODEs, especially for large models. It also presents a new way to train SDE-based models.

Code is available at: `https://github.com/a-norcliffe/torch_gq_adjoint`.

An associated video presentation is available at: `https://www.youtube.com/watch?v=pKbLwsqy8aM`.

## 1 Introduction

Neural ODEs (E, 2017; Chen et al., 2018) make an explicit connection between deep feedforward neural networks and dynamical systems. They take inspiration from the residual network architecture (He et al., 2016), where the $k + 1$-th hidden layer is related to the $k$-th by

$$\boldsymbol{z}_{k+1} = \boldsymbol{z}_k + f(\boldsymbol{z}_k, \boldsymbol{\theta}_k), \tag{1}$$

where $\boldsymbol{\theta}_k$ parameterises the learnable function $f$ at the $k$-th layer. Here, $\boldsymbol{z}_k \in \mathbb{R}^D$ for $k = 1, \ldots, K$, i.e., the dimensionality of the hidden units does not change as we progress through the $K$ layers.

We can interpret equation (1) as the Euler discretisation of a continuous-time dynamical system with a step size of 1. Neural ODEs are the limit of infinitesimally small step sizes, by directly parameterising the instantaneous rate of change of a state with a neural network at 'time' $t$ via

$$\frac{d\boldsymbol{z}}{dt} = f(\boldsymbol{z}, t, \boldsymbol{\theta}). \tag{2}$$

Equation 2 defines a continuous path between the input $\boldsymbol{z}(t_0)$ at an initial time $t_0$ and the state $\boldsymbol{z}(t_k)$ at a given later time $t_k$, which is evaluated using black-box ODE solvers.

---

*Majority of work done as a student at University College London September 2020 – September 2021.

Due to their continuous nature, neural ODEs are well suited to dealing with irregular time series (Chen et al., 2018; Kidger et al., 2020; Rubanova et al., 2019), allowing for analysis to be carried out without binning data. Neural ODEs also include the inductive bias that the data is generated from a dynamical system. This is particularly applicable in the natural sciences, for example, predator prey dynamics described by the Lotka–Volterra equations (Norcliffe et al., 2021; Dandekar et al., 2020). Additionally, neural ODEs can be applied to generative modelling. Continuous normalising flows (Chen et al., 2018; Grathwohl et al., 2019) build on normalising flows (Rezende & Mohamed, 2015) by allowing the flow to be defined by a vector field rather than a discrete change. Stochastic differential equations have also been used for score-based generative modelling (Song et al., 2020; Vahdat et al., 2021) showing strong results.

Neural ODEs are trained by calculating gradients of a scalar loss function $L$ and using a first-order optimiser, such as stochastic gradient descent, Adam (Kingma & Ba, 2015) or RMSProp (Tieleman & Hinton, 2012). Gradients can be calculated by directly backpropagating through the operations the ODE solver has taken; we call this the 'direct method'. An alternative method is the adjoint method (Pontryagin, 1987; LeCun et al., 1988; Chen et al., 2018), which solves a second ODE backwards in time. The direct method is faster but more memory intensive than the adjoint method (we include complexities of time and memory in Appendix C), in practice a trade-off must be made. The focus in this work is building on the adjoint method.

Solving these ODEs in general can be slow due to the computation demand of running an ODE solve (Lehtimäki et al., 2022). Speed is of particular interest when compute is limited, or more broadly for environmental concerns. We focus on this problem in this paper. Specifically, we focus on speeding up the training procedure of neural ODEs by providing a faster way to carry out the adjoint method while retaining its favourable memory footprint. The key idea is to use numerical integration, Gauß–Legendre quadrature, to solve one-dimensional definite integrals, which appear when computing gradients via the adjoint. These integrals are usually solved as differential equations, which slows down training considerably, compared with the Gauß–Legendre quadrature, which we propose using. This is a different way to approximate the solution to the *same* adjoint equation and should therefore produce the same gradients within numerical precision.

**Related Work**   There are various methods for speeding up neural ODEs. We can directly speed up the forward ODE solve by running it in a lower dimensional space using Model Order Reduction (Lehtimäki et al., 2022). We can also use regularisation techniques so that the trajectories are less complex, allowing the ODE solve to be faster (Onken et al., 2021). Kinetic regularisation (Finlay et al., 2020) includes terms in the loss function that penalise large velocities and Jacobians. Another method is to penalise large higher-order derivatives, such as the acceleration of the state, so that the learnt dynamics are easier to solve (Kelly et al., 2020). We can also include heuristics calculated by the solver in the loss function (Pal et al., 2021). This reduces the complexity of the dynamics by penalising small step sizes or large stiffness estimates. Further to this, we can also regularise the solutions without including any new terms in the loss. This can be done by uniformly sampling a terminal time $t_K$ (Ghosh et al., 2020). This encourages the model to reach the final state quickly (if the sampled end time is early), and then stay there (if the sampled end time is later), regularising the dynamics. Finally, we can directly restrict the dynamics to be faster to solve, for example (Xia et al., 2021; Nguyen et al., 2022), introduce damping style terms to the dynamics to regularize the ODE solutions. All these regularisation methods make inference faster, because solving the ODE is easier with simpler dynamics. However, this introduces restrictions on the trajectory, reducing expressivity of the model. This is particularly important in time-series applications, where the trajectory matters.

Instead of regularising the solutions, we can speed up the training process directly. Daulbaev et al. (2020) use a barycentric Lagrange interpolation to approximate the $z$ trajectory to solve the adjoint equations. This does not require a backwards ODE solve of $z$, which therefore speeds up the training process. However, this does require checkpoints of the trajectory $z$ to be stored during the forward solve, making the method less memory efficient than the adjoint method.

Seminorms (Kidger et al., 2021a) are the closest method to our work (that remains memory-efficient) by recognising the adjoint equations contain integrals. During the backward solve only a subset of the overall state is used by the ODE solver to estimate the truncation error, which is used to select the step size. This enables larger step sizes and faster solves. However, the adjoint integrals are still solved with ODE methods, which uses more computation than required. Rackauckas et al. (2020) use Gauß–Kronrod quadrature in place

of this to solve the integrals. However, this firstly requires more terms than other Gaussian quadratures and secondly requires dense solutions of the state and the adjoint state to carry out the quadrature, i.e., many checkpoints must be taken. This makes the method very memory intensive, which removes the advantage of using an adjoint based method over the direct method.

**Contributions** (i) We introduce the GQ method, a memory efficient and fast way to carry out the adjoint method using Gauß–Legendre quadrature, speeding up the training of neural ODEs. (ii) We show that this method can also be used to train neural SDEs using the Wong–Zakai theorem.

## 2 Neural ODEs

A general neural ODE consists of a dynamics function with learnable model parameters $\boldsymbol{\theta}$, an encoder with learnable parameters $\boldsymbol{\omega}$ and a decoder with learnable parameters $\boldsymbol{\phi}$. For an input $\boldsymbol{x}$ and given measurement times $\{t_0, t_1, ..., t_K\}$, the prediction at time $t_k$ is given by $\hat{\boldsymbol{x}}(t_k)$ by solving the ODE

$$\boldsymbol{z}(t_0) = h_e(\boldsymbol{x}(t_0), \boldsymbol{\omega}), \qquad \frac{d\boldsymbol{z}}{dt} = f(\boldsymbol{z}, t, \boldsymbol{\theta}), \qquad \hat{\boldsymbol{x}}(t_k) = h_d(\boldsymbol{z}(t_k), \boldsymbol{\phi}), \tag{3}$$

where the encoder $h_e$ and decoder $h_d$ are general functions. They can be identity operations so that the ODE can be thought of running in observation space, or they can be learnable functions so the ODE is latent (Rubanova et al., 2019).

Neural ODEs are trained by obtaining gradients of a scalar loss function $L$ and using a first-order optimiser of choice. ODE solvers, such as the Runge–Kutta solvers, consist of algebraic operations, all of which are differentiable. Therefore, one can directly backpropagate through an ODE solve to obtain gradients. However, when the dynamics are complex, this can lead to an arbitrarily large number of function evaluations for adaptive solvers, storing all of the intermediate activations during the solve, and the method becomes prohibitively memory intensive.

A less memory intensive method for training is using the adjoint method (Pontryagin, 1987; LeCun et al., 1988; Chen et al., 2018). This introduces the adjoint state $\boldsymbol{a_z}$, which obeys the (backwards-in-time) ODE

$$\frac{d\boldsymbol{a_z}}{dt} = -\boldsymbol{a}_z^T \frac{\partial f}{\partial \boldsymbol{z}}, \qquad \boldsymbol{a_z}(t_K) = \frac{\partial L}{\partial \boldsymbol{z}(t_K)}. \tag{4}$$

$dL/d\boldsymbol{z}(t_0)$ is given by $\boldsymbol{a_z}(t_0)$ and is used to train the encoder. A third state $\boldsymbol{a_\theta}$ is used to calculate the gradients for the dynamics function. It obeys the ODE

$$\frac{d\boldsymbol{a_\theta}}{dt} = -\boldsymbol{a}_z^T \frac{\partial f}{\partial \boldsymbol{\theta}}, \qquad \boldsymbol{a_\theta}(t_K) = \boldsymbol{0}. \tag{5}$$

The gradients of the loss $L$ with respect to the model parameters $\boldsymbol{\theta}$ are given by $\boldsymbol{a_\theta}(t_0)$. In the standard implementation of the method, these gradients are found by first solving the forward ODE for $\boldsymbol{z}(t)$ up to $t_K$, then solving the following concatenated ODE backwards in time

$$\frac{d}{dt} \begin{bmatrix} \boldsymbol{z} \\ \boldsymbol{a_z} \\ \boldsymbol{a_\theta} \end{bmatrix} = \begin{bmatrix} f(\boldsymbol{z}, t, \boldsymbol{\theta}) \\ -\boldsymbol{a}_z^T \nabla_{\boldsymbol{z}} f \\ -\boldsymbol{a}_z^T \nabla_{\boldsymbol{\theta}} f \end{bmatrix}, \qquad \begin{bmatrix} \boldsymbol{z} \\ \boldsymbol{a_z} \\ \boldsymbol{a_\theta} \end{bmatrix}(t_K) = \begin{bmatrix} \boldsymbol{z}(t_K) \\ \nabla_{\boldsymbol{z}(t_K)} L \\ \boldsymbol{0} \end{bmatrix}. \tag{6}$$

No intermediate activations are stored and so the method is memory efficient in the integration time. A fourth state $\boldsymbol{a_t}$ can be used to calculate gradients associated with measurement times. However, measurement times are typically not learnable functions[1], and gradients are not required. Further detail can be found in the original neural ODE paper (Chen et al., 2018). The method developed in this work directly applies to the case where we learn measurement times, and this is included in our code.

An important observation made by Kidger et al. (2021a) is that the differential equation for $\boldsymbol{a_\theta}$ does not actually contain $\boldsymbol{a_\theta}$. When the trajectories of $\boldsymbol{z}$ and $\boldsymbol{a_z}$ are known, $\dot{\boldsymbol{a}}_{\boldsymbol{\theta}} = -\boldsymbol{a}_z^T \nabla_{\boldsymbol{\theta}} f$ only depends on time.

---

[1]A noteable exception is adaptive depth neural ODEs (Massaroli et al., 2020).

They introduce seminorms to take advantage of this; this is when the ODE solver during the backward solve does not consider $a_\theta$ when calculating the error to choose a step size, because the error in $a_\theta$ does not grow significantly compared to $z$ or $a_z$ (Kidger et al., 2021a). While this allows for larger step sizes and faster training it still requires $a_z^T \nabla_\theta f$ to be calculated at every step. Instead we can rewrite the ODE solve in a way that it calculates the gradients associated with the parameters as a definite integral

$$\frac{dL}{d\theta} = \mathbf{0} + \int_{t_K}^{t_0} -a_z^T \frac{\partial f}{\partial \theta} dt = \int_{t_0}^{t_K} a_z^T(t) \frac{\partial f(t)}{\partial \theta} dt. \tag{7}$$

We can then use more advanced methods for solving definite integrals to compute the desired gradient faster and to the same level of accuracy. We only need to solve a smaller ODE for $[z, a_z]$. This will become significant when there are many parameters, making $a_\theta$ large. Importantly, despite having many parameters, the integration variable $t$ for computing the gradient in equation (7) is one-dimensional. Hence, we can solve many (easy) 1D integrals in parallel (one for each parameter), allowing us to use fast and accurate methods for solving 1D definite integrals. That is the key idea behind our approach, which we detail in the following.

## 3 Faster Training of Neural ODEs

In the following, we describe the methodology for faster training of neural ODEs. The key idea is to use more appropriate methods to calculate the definite integrals in the adjoint method given by equation (7); specifically we use Gauß–Legendre quadrature.

### 3.1 Gauß–Legendre Quadrature

Gaussian quadrature is a numerical integration method for calculating low-dimensional integrals as a weighted sum of function values

$$\int_a^b w(t) f(t) dt \approx \sum_{i=1}^n w_i f(\tau_i) \tag{8}$$

in the integration domain for given weights $w_i$, locations $\tau_i$, and integrand $f(t)$. Weights and locations are determined based on the locations of zeros of given (orthogonal) polynomials (Stoer & Bulirsch, 2002). Gaussian quadrature using $n$ terms in equation (8) is guaranteed to obtain the exact result for integrals of polynomials of up to degree $2n - 1$. Therefore, if the integrand can be well approximated by such a polynomial, this shall produce a good approximation to the integral. Gaussian quadrature is the fastest method for 1-D integrals outside of analytical solutions, it converges significantly faster than methods such as a Riemann Sum or the trapezoid rule; in particular solving with a differential equation solver requires $a_z^T(t) \frac{\partial f(t)}{\partial \theta}$ to be calculated more times than when using Gaussian quadrature (we give error bounds for the different integration methods in Appendix D). Therefore, we use Gaussian quadrature.

**Why Gauß–Legendre quadrature?** There are various flavours of Gaussian quadrature such as Gauß–Laguerre, Gauß–Hermite, Gauß–Jacobi. We specifically use Gauß–Legendre quadrature over other Gaussian quadratures for multiple reasons. Firstly, our integration intervals are both finite, ruling out Gauß–Laguerre quadrature which works on the $[0, \infty)$ interval and Gauß–Hermite quadrature which works on the $(-\infty, \infty)$ interval. Secondly and more specifically our integration interval is $[t_0, t_K]$, it includes the start and end time, rather than $(t_0, t_K)$, this excludes Gauß–Jacobi quadrature and Gauß–Chebyshev quadrature of the first kind. Thirdly, in general, Gaussian quadrature uses a weight function so that $I = \int_{t_0}^{t_k} w(t) f(t) dt$. Why do this? Consider the integral $\int_{-1}^{1} \sqrt{1 - t^2} f(t)$, this can be solved as $\int_{-1}^{1} w(t) \sqrt{1 - t^2} f(t) dt$ so that $w(t) = 1$ and we use Gauß–Legendre or we can solve it as $\int_{-1}^{1} w(t) f(t) dt$ where $w(t) = \sqrt{1 - t^2}$ and we use Gauß–Chebychev of the second kind. If we can write the integrand more simply as the product of a weight function and other function we may use other quadratures. In our case the dynamics function is a neural network and there is no clear way to write the dynamics as this product so we let $w(t) = 1$, which rules out Gauß–Chebyshev quadrature of the second kind. We do not use Gauß–Kronrod quadrature, an adaptive quadrature, because this requires the integrand to be stored at many locations in order to take advantage of

the adaptive nature of the scheme. This removes the memory efficiency of the method, and if memory is not a constraint it is often more useful to use direct backpropagation over any adjoint based method. Finally, we do not use other schemes such as Gauß–Lobatto quadrature because it does not converge as quickly, rather than $n$ points exactly solving a degree $2n - 1$ polynomial it only accurately solves up to degree $2n - 3$. After ruling out all of these option,s Gauß–Legendre quadrature is the only choice left.

## 3.2 Memory Efficiency

It is vital when using an adjoint-based method to be memory efficient in integration time. Otherwise directly backpropagating through the solver is the best method. This is the key flaw with the Gauß–Kronrod implementation (Rackauckas et al., 2020), in that it requires dense solutions of $z$ and $a_z$, consuming large amounts of memory. We can achieve memory efficiency by using a running total $g$ and add terms in the quadrature during the solve, rather than all at the end. That is, we initialize $g$ as a vector of zeros $\mathbf{0}$, and at each point in the quadrature sum $\tau_i$, we update $g$ to be $g + w_i a_z^T \nabla_\theta f(\tau_i)$, that way only needing two vectors at any point, the running total and the update. Without dense solutions, it is non-trivial to arbitrarily add terms to the sum because an ODE must be solved to query the integrand at time $t$. Therefore, we have to determine how many terms to use *before* starting the backward solve.

Preferably, the number of terms chosen adapts to the complexity of the problem, so that the smallest number of terms is used to compute an accurate gradient. Additionally, any information used to determine this must be gathered during the forward solve, and must be cheap to gather. Otherwise we do not reduce the computation, we just move it to a different process.

We opt for an empirical approach. During the forward solve, for effectively no extra computation, we count how many times the dynamics function is evaluated (NFE). When using an adaptive solver, the larger the NFE, the more complicated the trajectory. We make the assumption that if the (forward) trajectory of $z$ is complex then the corresponding gradient trajectory $a_z^T \nabla_\theta f$ is also complex, and more terms are required in the quadrature calculation. Considering the integration interval, larger intervals $[t_{k-1}, t_k]$ likely require more quadrature points to achieve a high accuracy. Knowing how these quantities should affect the number of terms, we propose the heuristic

$$n = \left\lceil C \frac{\text{NFE} \times (t_k - t_{k-1})}{(t_K - t_0)} \right\rceil, \tag{9}$$

where $C$ is a user chosen constant that tells us how the number of GQ terms scales with the NFE. If $C$ is larger, then we require more terms. We apply the ceiling function to ensure $n$ is both an integer and rounded up rather than down. We also enforce an upper bound of 64 quadrature points, which allows us to prescribe a polynomial of degree 127, to prevent a large number of terms slowing down the training. This does not negatively affect model performance in our experiments, see Appendix E.2 for an ablation study.

## 3.3 Algorithm

Algorithm 1 outlines an implementation of the GQ method. After initialising the state, the adjoint state and the running total, we then calculate the number of terms required using equation (9). The weights and locations for Gauß–Legendre quadrature are obtained by accessing a lookup table. We then loop through the weights and locations, solving the ODE up to that point, calculating $a_z^T \nabla_\theta f$ and adding to the running weighted sum. Finally we complete the backwards solve to obtain $a_z(t_0)$.

If the loss $L(\hat{x}(t_0), \hat{x}(t_1), ..., \hat{x}(t_K))$ depends on the state at multiple measurement times, there is a discontinuous change in the adjoint state $a_z$ at those times:

$$a_z(t_k^-) = a_z(t_k^+) + \frac{\partial L}{\partial z(t_k)}. \tag{10}$$

This discontinuity is accounted for by splitting the large integral into smaller integrals where the integrand is continuous and stepping through (for pseudocode we refer to Appendix A)

$$\int_{t_0}^{t_K} a_z^T \frac{\partial f}{\partial \theta} dt = \sum_{k=1}^{K} \int_{t_{k-1}}^{t_k} a_z^T \frac{\partial f}{\partial \theta} dt. \tag{11}$$

---

**Algorithm 1** Memory efficient implementation of the GQ method.

$\boldsymbol{z} \leftarrow \boldsymbol{z}(t_K), \quad \boldsymbol{a_z} \leftarrow \frac{\partial L}{\partial \boldsymbol{z}(t_K)}, \quad \boldsymbol{g} \leftarrow \boldsymbol{0}$

$n \leftarrow \text{GetNTerms}(t_0, t_K, \text{NFE}, C)$

$\boldsymbol{w}, \boldsymbol{\tau} \leftarrow \text{GetGQWeightsLocations}(n, t_0, t_K)$

$j \leftarrow n$

$t_{\text{prev}} \leftarrow t_K$

**while** $j \geq 1$ **do**

$\quad t_{\text{next}} \leftarrow \tau_j$

$\quad \begin{bmatrix} \boldsymbol{z} \\ \boldsymbol{a_z} \end{bmatrix} \leftarrow \text{ODESolve}(\begin{bmatrix} \boldsymbol{z} \\ \boldsymbol{a_z} \end{bmatrix}, \begin{bmatrix} f \\ -\boldsymbol{a_z}^T \frac{\partial f}{\partial \boldsymbol{z}} \end{bmatrix}, t_{\text{prev}}, t_{\text{next}})$

$\quad t_{\text{prev}} \leftarrow t_{\text{next}}$

$\quad \boldsymbol{g} \leftarrow \boldsymbol{g} + w_j \times \boldsymbol{a_z}^T \frac{\partial f}{\partial \boldsymbol{\theta}}$

$\quad j \leftarrow j - 1$

**end while**

$\begin{bmatrix} \boldsymbol{z} \\ \boldsymbol{a_z} \end{bmatrix} \leftarrow \text{ODESolve}(\begin{bmatrix} \boldsymbol{z} \\ \boldsymbol{a_z} \end{bmatrix}, \begin{bmatrix} f \\ -\boldsymbol{a_z}^T \frac{\partial f}{\partial \boldsymbol{z}} \end{bmatrix}, t_{\text{prev}}, t_0)$

**return** $\boldsymbol{g}, \boldsymbol{a_z}$

---

Overall, we have devised a fast and memory-efficient way to compute gradients in neural ODEs via the adjoint method. The key idea was to formulate the gradient of the loss w.r.t. the model parameters as a definite integral, and then to use Gaussian quadrature to solve this integral efficiently.

## 4 Faster Training of Neural SDEs

Neural ODEs have also been extended to learning stochastic differential equations (SDEs) in the form of neural SDEs (Tzen & Raginsky, 2019; Liu et al., 2019; Xu et al., 2021; Li et al., 2020; Kidger et al., 2021c;b). A Stratonovich SDE is given by

$$d\boldsymbol{z} = f(\boldsymbol{z}, t, \boldsymbol{\theta})dt + g(\boldsymbol{z}, t, \boldsymbol{\theta}) \circ dW_t, \tag{12}$$

where $f$ and $g$ are learnable functions of the drift and the diffusion and $W_t$ is a Wiener process. Neural SDEs can be applied to situations where there is inherent noise in the system, or an underlying random process affecting the dynamics, for example in mathematical finance (Black & Scholes, 1973).

Training neural SDEs suffers from similar speed and memory problems as neural ODEs. The GQ method from Section 3 can be extended to training neural SDEs by training a corresponding ODE and transferring the parameters to an SDE solver at test time. The Wong–Zakai theorem (Wong & Zakai, 1965) tells us that we can approximate an SDE using an ODE. For the Stratonovich SDE in equation (12) we consider the ODE

$$\frac{d\boldsymbol{z}_m}{dt} = f(\boldsymbol{z}_m, t, \boldsymbol{\theta}) + g(\boldsymbol{z}_m, t, \boldsymbol{\theta})\frac{dB_m}{dt}, \tag{13}$$

where $B_m(t)$ is a smooth approximation of a Wiener process determined by integer $m$ that improves as $m$ gets larger. An example could be a polynomial with degree $m$ (Foster et al., 2020). We include a subscript on $\boldsymbol{z}_m$ to show this is an approximation to the SDE. The Wong–Zakai theorem states that if $B_m(t) \to W_t$ as $m \to \infty$, then the solution $\boldsymbol{z}_m(t)$ to the ODE converges to a solution $\boldsymbol{z}(t)$ of the SDE . This allows us to approximate an SDE using an ODE (Londo & Villegas, 2016; Filip et al., 2019) and use the methods for training neural ODEs, such as our GQ method, to train neural SDEs and transfer the parameters at test time (Hodgkinson et al., 2021).

Our application requires the approximation of the Wiener process to be smooth and memory efficient in integration time; this prevents using interpolations through $m$ points, for example. We therefore use the Karhunen–Loève theorem, which says a Wiener process on the domain $[t_0, t_K]$ can be approximated by the Fourier series

$$W_t \approx \sqrt{2c} \sum_{i=1}^{m} \boldsymbol{\xi}_i \frac{\sin((i - \frac{1}{2})\pi(t - t_0)/c)}{(i - \frac{1}{2})\pi}, \tag{14}$$

where $c = t_K - t_0$ and $\boldsymbol{\xi}_i \sim \mathcal{N}(\mathbf{0}, \boldsymbol{I})$. We then differentiate this approximation, which yields

$$\frac{dB_m}{dt} = \sqrt{\frac{2}{c}} \sum_{i=1}^{m} \boldsymbol{\xi}_i \cos\left(\frac{(i - \frac{1}{2})\pi(t - t_0)}{c}\right). \tag{15}$$

This is not a significantly large memory overhead, and so $\boldsymbol{\xi}_i$ do not have to be resampled for each evaluation. However, if memory is scarce, we can further make this memory efficient by calculating equation (15) using a running sum, so that we iteratively sample $\boldsymbol{\xi}_i$ and then add the cosine, rather than storing all $\boldsymbol{\xi}_i$. We use pseudo-random numbers and re-seed a generator to guarantee we recover the same $B_m(t)$ throughout the forwards and backwards solves.

## 5 Experiments and Results

In this section, we compare the GQ method against the direct, adjoint and seminorm methods. The main aim of this work is to speed up the training of neural ODEs and SDEs, without compromising performance. Therefore, the key metric to compare methods is the training time, provided the final performances are the same, we discuss the number of function evaluations and why this is not the most reliable metric in our case in Appendix E.1. The GQ method is implemented in PyTorch building on the `torchdiffeq` library (Chen et al., 2018). Code is publicly available at: `https://github.com/a-norcliffe/torch_gq_adjoint`.

All experiments were run on NVIDIA GeForce RTX 2080 GPUs. We use the Dormand–Prince 5(4) solver with an absolute and relative tolerance of $1 \times 10^{-3}$ and $C = 0.1$ for the GQ method. We found that this value worked well, giving accurate gradients quickly, we run an ablation on this value in Appendix E.2. We train using the Adam optimiser (Kingma & Ba, 2015). For classification tasks we use constant integration times of $[0, 1]$. Additional experimental details such as task-dependent learning rates, information on datasets and further experiments (such as test performance against wall-clock time) are given in Appendix E.

### 5.1 Accuracy of Gradients

For Neural ODEs, it is difficult to determine the true gradient because the forward pass is approximated. To test the GQ method in an analytical setting, we have recreated and extended the experiment in ACA (Zhuang et al., 2020) and MALI (Zhuang et al., 2021). Here we use an analytical system that we can control with exact expressions for the loss and gradients. We consider the exponential growth system $\dot{z} = az \rightarrow z(t) = z_0 \exp(at)$, with a loss $L = z(T)^2 = z_0^2 \exp(2aT)$. The gradients with respect to initial condition $z_0$, parameter $a$ and integration time $T$ are: $2z_0 \exp(2aT)$, $2Tz_0^2 \exp(2aT)$ and $2az_0^2 \exp(2aT)$ respectively. In Figure 1 we plot the relative errors $\left|\frac{\text{True} - \text{Predicted}}{\text{True}}\right|$ in the loss and the different gradients for various methods as the integration time $T$ is increased. We plot relative errors since we use an exponentially growing system and want errors for small $T$ to be equally as visible as errors for large $T$. We use a value of 0.2 for $a$, 10.0 for $z_0$ and a range of 19-29 for $T$. The GQ method (blue) produces the same gradients as the standard adjoint method (pink), demonstrating that this is as accurate as the adjoint method, which supports the claim it is solving the same equation in a different way. We also see that the direct, standard and GQ adjoints give the same loss as we expect since they have the same forward solve. Importantly we see that the direct method gives different gradients to the adjoint methods which we expect since they are two different approaches (discretize-then-optimize vs optimize-then-discretize). MALI and ACA have been included to show how using different methods for the forward solve can change the results.

### 5.2 Memory Efficiency

We additionally investigate how much memory is used by each method on the analytical task. We test the direct, standard adjoint, GQ adjoint and ACA methods. We solve the same task up to $T = 15$ and then take the gradient using each method. The maximum memory consumption is then determined in bytes, and plotted against the solver tolerance (lower tolerance means more steps to solve). We plot these in Figure 2. We see that as expected the GQ method is constant in the tolerance, and the number of integration steps. Adjoint is also efficient, but uses slightly less memory. We hypothesize that this is because the adjoint method uses one function evaluation to calculate $[f(\boldsymbol{z}, t, \boldsymbol{\theta}), -\boldsymbol{a}_{\boldsymbol{z}}^T \nabla_{\boldsymbol{z}} f, -\boldsymbol{a}_{\boldsymbol{z}}^T \nabla_{\boldsymbol{\theta}} f]$, whereas GQ uses

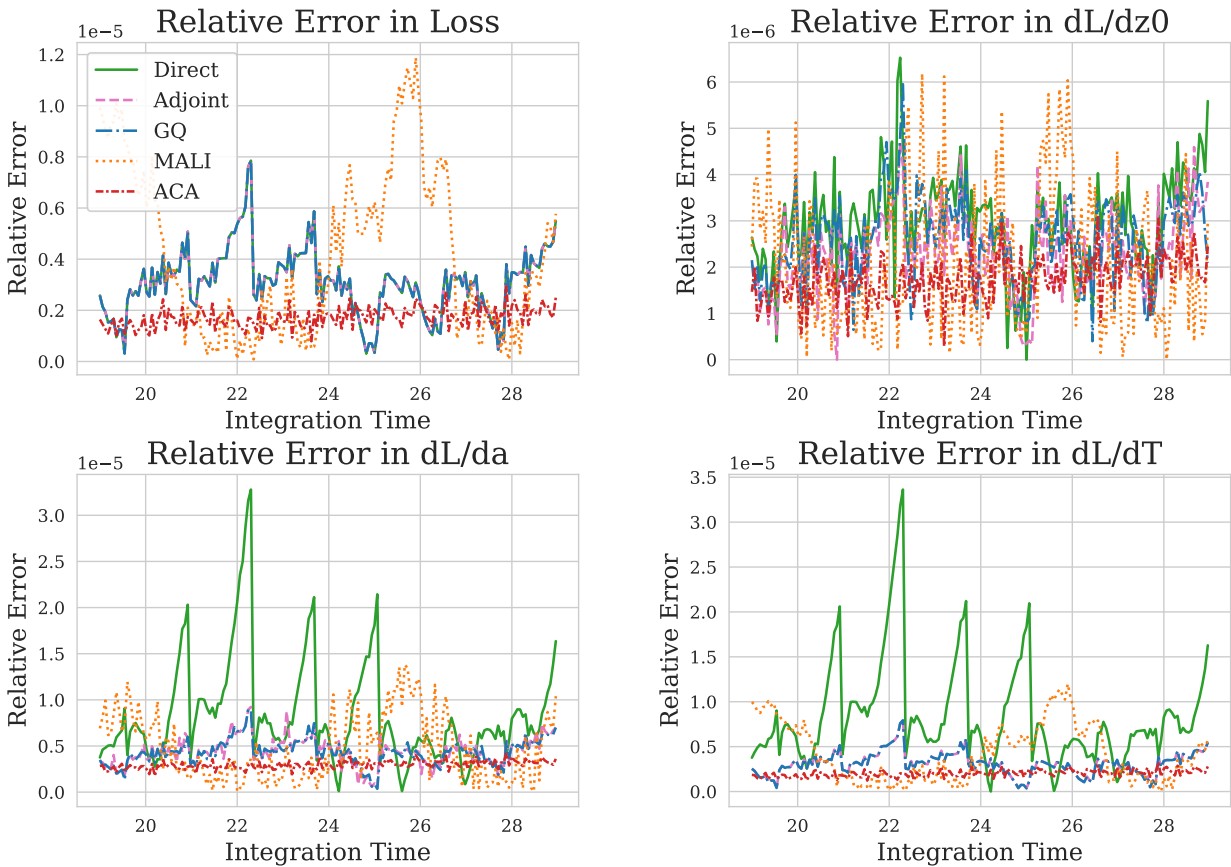

Figure 1: Absolute relative errors in losses and gradients for the toy gradient problem. We see that the GQ method performs well relative to the other well established methods, showing that it produces reliable gradients.

one evaluation to calculate $[f(\boldsymbol{z}, t, \boldsymbol{\theta}), -\boldsymbol{a}_{\boldsymbol{z}}^T \nabla_{\boldsymbol{z}} f]$ and another to calculate $\boldsymbol{a}_{\boldsymbol{z}}^T \nabla_{\boldsymbol{\theta}} f$ at the quadrature point, so there is a slightly higher memory usage for GQ, the main takeaway is that it is constant in the tolerance. ACA and the direct method scale poorly as expected.

### 5.3 Nested Spheres

The Nested Spheres experiment (Dupont et al., 2019; Massaroli et al., 2020) is an illustrative classification task. The function in $d$ dimensions consists of an inner sphere entirely surrounded by an outer sphere

$$g(\boldsymbol{x}) = \begin{cases} 0, & 0 \leq ||\boldsymbol{x}||_2 \leq r_1 \\ 1, & r_2 \leq ||\boldsymbol{x}||_2 \leq r_3 \end{cases}, \tag{16}$$

for $0 < r_1 < r_2 < r_3$. Vanilla neural ODEs cannot solve this problem, because any mapping preserves topology, preventing the two regions being linearly separable (Dupont et al., 2019). This can be seen in a plot of example 2D data in Figure 3.

We train an augmented neural ODE (Dupont et al., 2019) with three augmented dimensions for 100 epochs, with a batch size of 200 using the cross-entropy loss. We use a time-dependent, multilayer perceptron with softplus activations and two hidden layers as the dynamics function. We use different hidden widths to test the GQ method with varying numbers of parameters. We train across 10 seeds with results given in Figure 4. Training times are recorded as the time taken to complete 100 epochs.

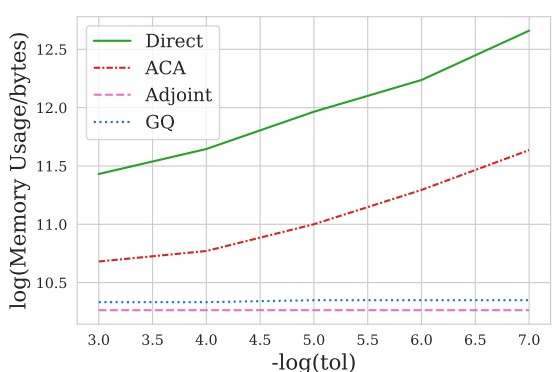

Figure 2: Maximum memory usage by each method on the analytical task.

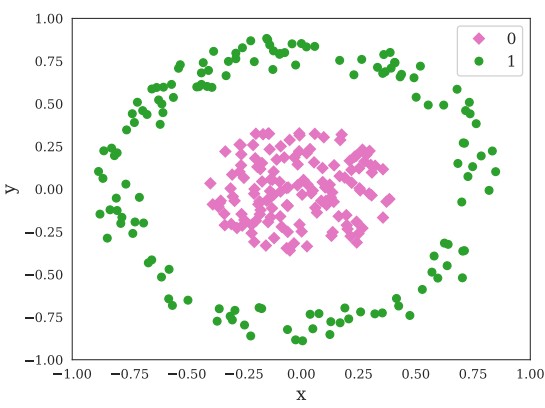

Figure 3: Example 2D data from the nested spheres training set. The two regions cannot be linearly separated by a vanilla neural ODE.

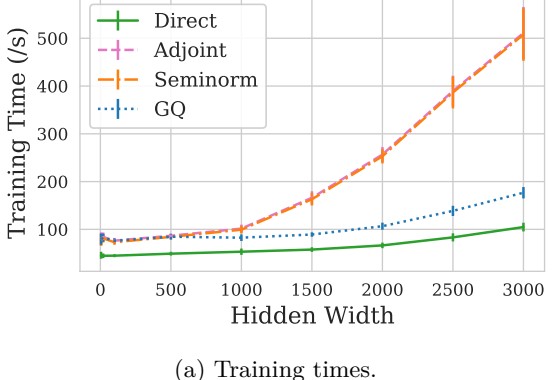

(a) Training times.

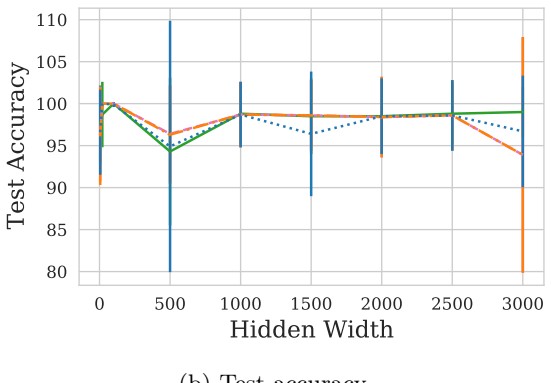

(b) Test accuracy.

Figure 4: Training times and test accuracies on the nested spheres task. The GQ method's training time scales well with model size compared to the adjoint and seminorm. The test accuracies are all comparable showing the method produces accurate gradients.

Directly backpropagating has the fastest training times, as expected. When the number of parameters is low, the methods have comparable training times. However, as the number of model parameters increases, the training times for the adjoint and seminorm methods increase more significantly than the GQ or direct methods. We also see that the models have the same accuracies on the test set (within a standard deviation) showing that the gradients produced by the GQ method are accurate.

## 5.4 Time Series

To test neural ODEs trained using the GQ method on time series, we consider sine curves $\ddot{x} = -x$, where the underlying ODE is second order. We train a second-order neural ODE (Norcliffe et al., 2020; Massaroli et al., 2020), where the initial position and velocity are given. Therefore, the model only has to learn the acceleration. We train using MSE as the loss for 250 epochs and a batch size of 15. This is repeated over 5 seeds to obtain means and standard deviations; results are given in Figure 5.

We see similar results as those in the nested spheres experiment. The direct method is the fastest. The time taken to train is about the same for a small model between the adjoint/seminorm methods and the GQ method. However, as the models get larger, the time taken scales far more favourably for the GQ method

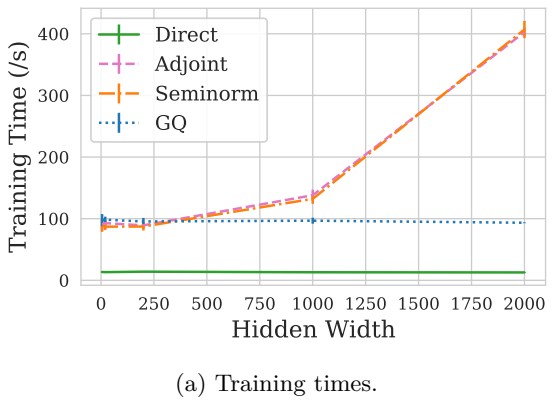 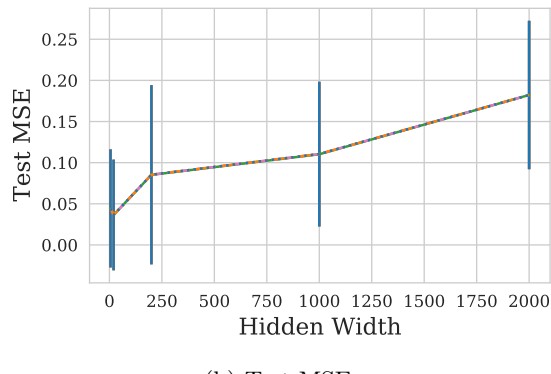

(a) Training times.           (b) Test MSE.

Figure 5: Training times and test MSEs on the sines task. The GQ method's training time scales well with model size compared to the adjoint and seminorm. The test MSEs are all comparable showing the method produces accurate gradients.

than the adjoint and seminorm methods. We also see that the final test MSEs are the same, with the same standard deviations, showing the methods produce the same gradients.

## 5.5 Image Classification

We now test the GQ method on more difficult classification. We consider image classification using the MNIST (LeCun et al., 1998) dataset. Further results are given on the CIFAR-10 (Krizhevsky, 2009) and SVHN (Netzer et al., 2011) datasets in Appendix E. We use convolutional initial downsampling layers and dynamics function and a fully connected final set of layers to obtain logits. We train for 15 epochs with a batch size of 16. We train using three random seeds to obtain means and standard deviations. We do not consider the direct method due to memory consumption. Results are given in Figure 6.

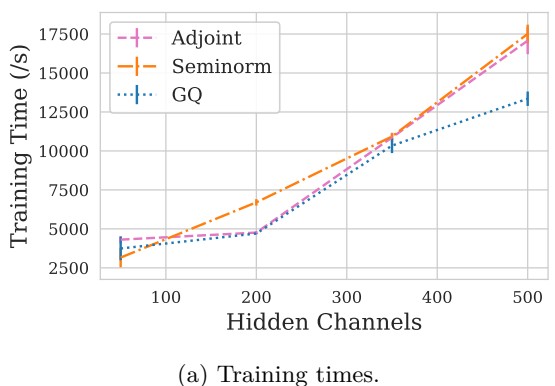 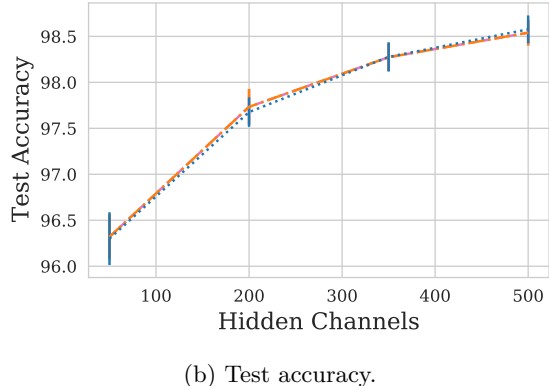

(a) Training times.           (b) Test accuracy.

Figure 6: Training times and test accuracies on the MNIST task. The GQ method's training time scales well with model size compared to the adjoint and seminorm. The test accuracies are all comparable showing the method produces accurate gradients.

As in the previous experiments we see that as the model (and the corresponding number of parameters) becomes large, the training time scales more favourably for the GQ method. The test accuracy does not change significantly between the methods either, so that the method produces accurate gradients.

### 5.6 Ornstein–Uhlenbeck Process

Finally, we consider how our method extends to training neural SDEs, by considering the Ornstein–Uhlenbeck (OU) process (Uhlenbeck & Ornstein, 1930; Kidger et al., 2021c;b). A one-dimensional OU process is governed by the SDE

$$dx = -\theta x dt + \sigma \circ dW_t. \tag{17}$$

We consider a harder variation of the SDE with time-dependent drift and diffusion

$$dx = (\mu t - \theta x)dt + (\sigma + \phi t) \circ dW_t \tag{18}$$

for scalar parameters $\mu$, $\theta$, $\sigma$ and $\phi$ and a one-dimensional Wiener process $W_t$. We train and evaluate using the KL divergence and a batch size of 40. We do not use an encoder or decoder; the model acts in observation space. The drift and diffusion are separate time dependent multilayer perceptrons, with two hidden layers. We use 10 cosines to approximate the Wiener process when training the corresponding ODE, we run an ablation over this value in Appendix F. We compare our method to directly backpropagating through an SDE solver and with the SDE adjoint method (Li et al., 2020), using the reversible Heun method (Kidger et al., 2021b) allowing for fast and reversible SDE solves. We use a relatively large step size of 0.01 for the SDE solvers to reduce their training time. We use the same SDE solver to evaluate parameters trained by the GQ method. Training times and test KL divergences are given in Figure 7.

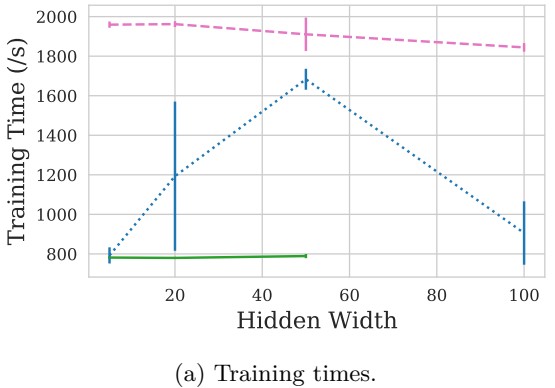
(a) Training times.

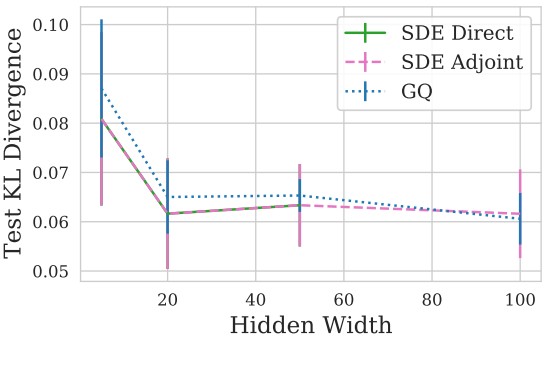
(b) Test KL divergence.

Figure 7: Training times and test KL divergences for different widths on the OU experiment. The SDE direct method is fast, but runs out of memory, the GQ method is faster than the SDE adjoint method. The methods have comparable test KL divergences.

The SDE direct method does not record values for a hidden width of 100. This is because the GPU ran out of memory. We see before this, that the direct method is the fastest, due to not having to solve a backwards SDE (or ODE) to obtain gradients. The GQ method is second fastest with the SDE adjoint method being slowest due to the time taken to solve the forward and backward SDEs. We do not see the previous effect of model size on training time because we are still in the 'small parameter regime', which is why the training time fluctuates rather than increasing with model size. The KL divergences for each method are approximately the same and within a standard deviation of each other, showing that the GQ method produces accurate gradients for neural SDEs as well as ODEs. The small difference likely comes from training a corresponding ODE, rather than the SDE directly.

## 6 Discussion and Practical Guidelines

We saw that the GQ method is a promising alternative to the standard adjoint method for training neural ODEs. One of the significant advantages of the GQ method is that it speeds up the training of neural ODEs without affecting the dynamics. In this way the GQ method is orthogonal to regularisation schemes, such

as STEER (Ghosh et al., 2020), kinetic regularisation (Finlay et al., 2020), regularising higher-order terms (Kelly et al., 2020) and training with solver heuristics (Pal et al., 2021).

When the model has many parameters, the GQ method can be significantly faster than the standard adjoint method depending on the state size. Additionally, the GQ method enforces no further restrictions, so it can be used to replace any existing models trained with the standard adjoint method, and it will speed up models without limiting their expressivity. We can also apply this method to train SDEs and, depending on the step size, the GQ method can be faster than the SDE adjoint method.

However, when the state is large, the improvement in speed is either negligible compared to the adjoint method or it can even be slower as shown in Appendix E. There are also no theoretical guarantees on the heuristic used to calculate the number of terms in equation (9). We did not experience cases where the method does not produce accurate gradients; however, that does not mean they do not exist. Hence, the GQ method should be used in situations with large models, and small states. If memory capacity is large, then the direct method will be the fastest method. In the case where the model is small and the state is large, the adjoint method will be the best method. The same holds for the SDE versions of these methods.

## 7 Conclusion and Future Work

We looked at how the training of neural ODEs can be made faster. We showed that the adjoint method, the current method of choice, inefficiently solves a definite integral using ODE methods, contributing to the slow training. We showed that we can use Gauß–Legendre quadrature to speed up the training and showed how to make this method memory efficient, a key requirement of neural ODE training at scale. Following this, we extended the method to training neural SDEs using the Wong–Zakai and the Karhunen–Loève theorems.

We tested both the speed of the GQ method and the reliability of its gradients on classification and time-series regression. In all of our experiments, we saw that training using the GQ method produces the same model performance as training with the direct or adjoint methods for ODEs and SDEs. We also saw that when the model size is large and the state is small, the GQ method outperforms the adjoint method significantly in terms of training time.

One of the significant advantages of the GQ method is that it speeds up the training of neural ODEs without affecting the dynamics. In this way the GQ method is orthogonal to regularisation schemes, such as STEER (Ghosh et al., 2020), kinetic regularisation (Finlay et al., 2020), regularising higher order terms (Kelly et al., 2020) and training with solver heuristics (Pal et al., 2021). Therefore, it would be interesting to see how much the method would speed up if these techniques were used in conjunction with the GQ method.

### Broader Impact Statement

In our work we investigate speeding up neural ODEs using Gauß–Legendre quadrature. We also extend the method to training neural SDEs. Our work is incremental and focuses on the speed of training rather than model performance. Therefore, we envision any large societal impacts coming more from applying differential equation models, rather than using our specific method to train.

**Applications** Neural ODEs and neural SDEs are still relatively new models and are yet to be used in any unethical situations or on any large scale to our knowledge. Differential equation based models have seen success in both time-series modelling and generative modelling. They will likely be applied further in these areas, such as predicting the evolution of a dynamical system in natural sciences, or generating images using score-based modelling. The models are still at risk of learning biases within a dataset. However, they are no more at risk of this than discrete models. In fact, there is evidence to show that they may be more robust in many cases than discrete models (Yan et al., 2020), due to the property of no crossing trajectories and preserving topology (Dupont et al., 2019).

**Environmental Implications** The contributions of this paper are to speed up existing methods for training neural ODEs. Therefore we anticipate a positive impact on the environment because less computation is needed to produce the same results for the same model.

**Datasets**  We do not use any datasets that could be considered to be sensitive. All experiments used synthetic data, with the exception of the well-known image datasets: MNIST, CIFAR-10 and SVHN.

**Code Release**

Our code is publicly available at: `https://github.com/a-norcliffe/torch_gq_adjoint`.

**Author Contributions**

**Alexander Norcliffe:**  Lead author. Developed the idea and full implementation. Ran the experiments. Joint effort writing the manuscript. Wrote the author responses during review.

**Marc Deisenroth:**  Senior author. Initial broad conceptualisation of the project and provided guidance on research directions. Joint effort writing the manuscript. Provided significant input to author responses. Provided access to the hardware to run the experiments.

**Acknowledgments**

At the time of publication, Alexander Norcliffe is supported by a grant from GlaxoSmithKline. We thank So Takao for his help understanding the Wong–Zakai theorem for the SDE adaptation of the GQ method. We would like to thank the anonymous reviewers and action editor Kevin Swersky for their time and efforts to review and constructively critique the paper. Our implementation relied heavily on the `torchdiffeq` library, we thank the authors Ricky Chen, Yulia Rubanova, Jesse Bettencourt and David Duvenaud for their work on this library and the Neural ODE paper.

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

## A  Multiple Measurement Times

In Section 3.3 we described the GQ method using an intial and terminal time only, here we consider multiple measurement times. The integrand is given by $\boldsymbol{a}_{\boldsymbol{z}}^T \nabla_{\boldsymbol{\theta}} f$ which can be integrated using Gauß–Legendre quadrature. However, when the loss depends on $K > 2$ measurement times $L(\hat{\boldsymbol{x}}(t_0), \hat{\boldsymbol{x}}(t_1), ..., \hat{\boldsymbol{x}}(t_K))$, there is a discontinuity in the adjoint state at the intermediate times given by

$$\boldsymbol{a}_{\boldsymbol{z}}(t_k^-) = \boldsymbol{a}_{\boldsymbol{z}}(t_k^+) + \frac{\partial L}{\partial \boldsymbol{z}(t_k)}. \tag{19}$$

The integrand is now not continuous, so Gaussian quadrature does not apply well to the integration domain $[t_0, t_K]$. However, quadrature still applies well to the sub-domains given by $[t_{k-1}, t_k]$, for $k = \{1, 2, ..., K\}$, because the integrand *is* continuous there. Therefore, we split the whole integration into many smaller ones and step through

$$\int_{t_0}^{t_K} \boldsymbol{a}_{\boldsymbol{z}}^T \frac{\partial f}{\partial \boldsymbol{\theta}} dt = \sum_{k=1}^{K} \int_{t_{k-1}}^{t_k} \boldsymbol{a}_{\boldsymbol{z}}^T \frac{\partial f}{\partial \boldsymbol{\theta}} dt, \tag{20}$$

applying quadrature to the smaller integrals. This more general implementation is given in Algorithm 2.

---

**Algorithm 2** Memory efficient implementation of the GQ method with $K$ measurement times.

$\boldsymbol{a}_{\boldsymbol{z}} \leftarrow \boldsymbol{0}, \quad \boldsymbol{g} \leftarrow \boldsymbol{0}$
$k \leftarrow K$
**while** $k \geq 1$ **do**              ▷ There is now an outer loop over $k$, which sums the smaller integrals.
    $\boldsymbol{z} \leftarrow \boldsymbol{z}(t_k), \quad \boldsymbol{a}_{\boldsymbol{z}} \leftarrow \boldsymbol{a}_{\boldsymbol{z}} + \frac{\partial L}{\partial \boldsymbol{z}(t_k)}$
    $n \leftarrow \text{GetNTerms}(t_{k-1}, t_k, \text{NFE}, C)$
    $\boldsymbol{w}, \boldsymbol{\tau} \leftarrow \text{GetGQWeightsLocations}(n, t_{k-1}, t_k)$
    $j \leftarrow n$
    $t_{\text{prev}} \leftarrow t_k$
    **while** $j \geq 1$ **do**
        $t_{\text{next}} \leftarrow \tau_j$
        $\begin{bmatrix} \boldsymbol{z} \\ \boldsymbol{a}_{\boldsymbol{z}} \end{bmatrix} \leftarrow \text{ODESolve}\left( \begin{bmatrix} \boldsymbol{z} \\ \boldsymbol{a}_{\boldsymbol{z}} \end{bmatrix}, \begin{bmatrix} f \\ -\boldsymbol{a}_{\boldsymbol{z}}^T \frac{\partial f}{\partial \boldsymbol{z}} \end{bmatrix}, t_{\text{prev}}, t_{\text{next}} \right)$
        $t_{\text{prev}} \leftarrow t_{\text{next}}$
        $\boldsymbol{g} \leftarrow \boldsymbol{g} + w_j \times \boldsymbol{a}_{\boldsymbol{z}}^T \frac{\partial f}{\partial \boldsymbol{\theta}}$
        $j \leftarrow j - 1$
    **end while**
    $\begin{bmatrix} \boldsymbol{z} \\ \boldsymbol{a}_{\boldsymbol{z}} \end{bmatrix} \leftarrow \text{ODESolve}\left( \begin{bmatrix} \boldsymbol{z} \\ \boldsymbol{a}_{\boldsymbol{z}} \end{bmatrix}, \begin{bmatrix} f \\ -\boldsymbol{a}_{\boldsymbol{z}}^T \frac{\partial f}{\partial \boldsymbol{z}} \end{bmatrix}, t_{\text{prev}}, t_{k-1} \right)$
    $k \leftarrow k - 1$
**end while**
**return** $\boldsymbol{g}, \boldsymbol{a}_{\boldsymbol{z}}$

---

Algorithm 2 is very similar to Algorithm 1, however there is now an outer loop over the measurement times indexed by $k$. The adjoint state is initialised at $\boldsymbol{0}$, and the discontinuity is applied at the beginning of each smaller integral. The smaller integral is then solved as in Algorithm 1 but using the integration domain $[t_{k-1}, t_k]$ rather than $[t_0, t_K]$.

NOTE: There is an important implementation detail. We do not restart a new solve except at the measurement times where there is the discontinuous change. That is, in the inner loop of Algorithm 2, and the only loop of Algorithm 1 we *continue* the ongoing solve and do not start a new one. This is because starting a solve involves initial steps with a fixed computational cost that we both wish and are able to avoid. We recommend looking at our code to see exactly how this is done.

## B    Training Neural SDEs

The final generalization of the GQ method is to stochastic differential equations. We train the drift $f$ and diffusion $g$ by training an ODE approximate solution of the SDE, using a Karhunen–Loève expansion. We can then use the GQ adjoint method to train this. The algorithm is exactly the same as Algorithm 2 for time-series or Algorithm 1 when only the end point matters. The main difference is that in those algorithms we have

$$\frac{d\boldsymbol{z}}{dt} = f(\boldsymbol{z}, t, \theta)$$

whereas for the SDE we have

$$\frac{d\boldsymbol{z}}{dt} = f(\boldsymbol{z}, t, \theta) + g(\boldsymbol{z}, t, \theta)\left[\sqrt{\frac{2}{t_K - t_0}}\sum_{i=1}^{m}\boldsymbol{\xi}_i \cos\left(\frac{(i - \frac{1}{2})\pi(t - t_0)}{t_K - t_0}\right)\right]$$

where $\boldsymbol{\xi}_i \sim \mathcal{N}(\boldsymbol{0}, \boldsymbol{I})$. Following this we use an SDE solver with the trained drift and diffusion functions at test time.

## C    Adjoint vs Direct

In our Introduction, Section 1 we stated that the direct method is faster than the adjoint but also more memory intensive. To make the statements more precise, consider a solve with state size $N_z$, dynamics function with $N_f$ hidden layers, number of function evaluations in the forward pass $N_t$ and number of evaluations in the backward pass $N_r$. Then the memory usage for the direct method is $\mathcal{O}(N_z N_f N_t)$ whereas for the standard adjoint method it is $\mathcal{O}(N_z N_f)$, which is why it is often referred to as constant with respect to the integration time, since it does not depend on $N_t$. This is because the direct method has to store every activation every time the dynamics function is evaluated, whereas the adjoint method only has to store the activations from one function evaluation. The computational complexity is then $\mathcal{O}(2N_z N_f N_t)$ for the direct method and $\mathcal{O}(N_z N_f (N_t + N_r))$ for the adjoint method, but the state in the backward solve is far larger increasing the computation. These values are taken from Table 1 from Zhuang et al. (2021), and they represent the dominant terms in the scaling.

It should be noted that the direct and adjoint methods do give different results, certainly for the gradients. Depending on the task and the parameter loss landscape this may not result in a difference in the final performance, the models may settle in different but equally good local optima. Or the loss landscape might be sufficiently shaped such that even with different gradients they still reach the same minimum, for example if the loss landscape is convex (this is very rarely the case). However it is still the case that the gradients are different, this is evident in our analytical experiment in Figure 1 where the errors in the loss between direct and adjoint are the same but the errors in the gradient are different. Interestingly for this experiment the errors are larger for the direct than the adjoint method, this is in contrast to (Gholami et al., 2019). We hypothesize this is because in ANODE the dynamics are learnt so that images converge on a final value to be turned into a distribution over classes, whereas in our exponential experiment the dynamics diverge, therefore for practical purposes where we do not want diverging dynamics it is likely that the direct method will produce preferable gradients. One example where this is not the case is Xu et al. (2023) which shows using the leapfrog ODE solver can produce incorrect gradients when directly backpropagating which oscillate around the true ODE gradient. Whilst comparisons with the direct method are not this main paper's aim, we do stress that care should be taken when selecting a method, since speed and memory consumption are factors but also in some cases the gradients will be different as well, which can lead to different results.

## D    Numerical Integration

**Error Rates of Numerical Integration Method**    In Section 3.1 we stated that Gaussian Quadrature is the fastest method for numerically calculating 1D integrals. We give precise error bounds here to demonstrate this. Assume we wish to solve the integral given by $\int_a^b f(t)dt$, below we give three common techniques for numerically solving the integral and the error bounds.

**Newton-Cotes** Newton-Cotes integration techniques split the interval $[a, b]$ into $N$ *equidistant* intervals. Then a polynomial of a given degree is fit on each interval which can be integrated. Classic examples are:

- Trapezoid rule, this fits a linear interpolant to each interval. The error is $\mathcal{O}\left(\frac{f^2(\xi)}{N^2}\right)$, where $f^m(\xi)$ refers to the largest $m$-th derivative on the interval $[a, b]$ located at $\xi$.

- Simpson's rule, this fits quadratic interpolants on the intervals. The error is $\mathcal{O}\left(\frac{f^4(\xi)}{N^4}\right)$.

- Boole's rule, this fits quartic interpolants on the intervals. The error is $\mathcal{O}\left(\frac{f^6(\xi)}{N^6}\right)$.

In the above, $f^m(\xi)$ is a constant, it has been included in the error complexity to show that if the true function is exactly a polynomial of a given degree the error can be zero. For example, if the function is exactly linear the second derivative everywhere is zero, so the Trapezoid and other Newton-Cotes rules give the exact results. However the crucial point is that these are still constant with respect to $N$ so the errors for the Trapezoid, Simpson and Boole's rules are $\mathcal{O}\left(\frac{1}{N^2}\right)$, $\mathcal{O}\left(\frac{1}{N^4}\right)$ and $\mathcal{O}\left(\frac{1}{N^6}\right)$. Therefore for Newton-Cotes the error is $\mathcal{O}\left(\frac{1}{N^k}\right)$ where $k$ depends on the rule for interpolation.

**Gaussian Quadrature** Gaussian Quadrature *does not use equidistant points*, the points are given by principled locations on the interval - the roots of orthogonal polynomials which we can look up in a table. This allows us to approximate a $2N - 1$ degree polynomial with only $N$ points in the quadrature sum. The error using Gaussian Quadrature with weight function 1 is bounded by $\frac{(b-a)^{2N+1}(N!)^4}{(2N+1)((2N)!)^3} f^{2N}(\xi)$. There are two key terms in this error, the first is $f^{2N}(\xi)$, as we use more points we model a higher degree polynomial unlike Newton-Cotes which stays constant. As mentioned we are therefore already able to exactly solve a polynomial of degree $2N - 1$, so if the function is well approximated by such a polynomial we will have an accurate approximation of the integral. The other key term is $((2N)!)^3$ in the denominator, which dominates the other terms in the error bound showing the error will shrink very quickly with $N$, significantly faster than if the error is $\mathcal{O}\left(\frac{1}{N^k}\right)$.

Note that the above assumes that $f$ is differentiable, if there are discontinuities in $f$ the integral is split up to make up for that (as described in Appendix A).

**Monte Carlo Integration** Monte Carlo methods approximate the integral by sampling many points and calculating the mean of these. $\int_a^b f(t)dt = \int_a^b p(t)\frac{f(t)}{p(t)}dt \approx \frac{1}{N}\sum_{i=1}^N \frac{f(t_i)}{p(t_i)}$ $t_i \sim p(t)$. By sampling the points they are *not equidistant but also not in principled locations*. The distribution of these Monte Carlo estimates follows a normal distribution with the mean being the true integral and the variance is $\mathcal{O}\left(\frac{1}{N}\right)$, so the error using Monte Carlo is $\mathcal{O}\left(\frac{1}{\sqrt{N}}\right)$. Monte Carlo has the worst error rates of the described methods which is why it is only used for high dimensional integrals where it is not possible to apply the other methods.

Therefore we use Gaussian Quadrature for these integrals since it has the best error rate. All of this information has been taken from Stoer & Bulirsch (2002)

**Shifting Integration Domain** The weights and locations in Gauß–Legendre quadrature are only defined on the interval $[-1, 1]$. And so these can only be used to approximate integrals of the form $\int_{-1}^1 f(t)dt$, whereas our integral is on the general domain given by the start and end times of the ODE solve $[t_0, t_K]$. Fortunately with a simple change of variables we can scale and shift our domain to be $[-1, 1]$. Consider the integral given by

$$I = \int_{t_0}^{t_K} f(t)dt,$$

by using the change of variables $\tilde{t} = \frac{2}{t_K - t_0}t - \frac{t_K + t_0}{t_K - t_0}$ we can rewrite the integral as

$$I = \int_{t_0}^{t_K} f(t)dt = \int_{-1}^1 f\left(\frac{t_K - t_0}{2}\tilde{t} + \frac{t_K + t_0}{2}\right)\frac{t_K - t_0}{2}d\tilde{t}$$

And this can then be approximated using Gauß–Legendre quadrature

$$\int_{t_0}^{t_K} f(t)dt \approx \sum_{i=1}^{n} w_i f\left(\frac{t_K - t_0}{2}\tau_i + \frac{t_K + t_0}{2}\right)\frac{t_K - t_0}{2}$$

allowing us to use Gauß–Legendre quadrature in our setting.

## E   Experimental Details

In this section we provide further details and results for our experiments. For all experiments we use the Dormand–Prince 5(4) solver with an absolute and relative tolerance of $1 \times 10^{-3}$. For the GQ method we use $C = 0.1$. We train using the Adam optimiser (Kingma & Ba, 2015). For classification tasks we use constant integration times of $[0, 1]$. For fair comparison, all experiments were run on NVIDIA GeForce RTX 2080 GPUs. Minor hyperparameter tuning of the batch size and learning rate was carried out initially on each task to obtain acceptable model performance. However, the key metric to compare methods is training time, provided the model performances are the same or very similar. We first calculate the approximate number of function evaluations below and justify why it is not the most indicative metric of speed in our scenario.

### E.1   Number of Function Evaluations

Unlike in other methods for speeding up the adjoint method, the number of function evaluations is not indicative of wall-clock time for the GQ method. This is because in previous methods during the backward solve one function evaluation is used each time to calculate the velocity of $[\boldsymbol{z}, \boldsymbol{a_z}, \boldsymbol{a_\theta}]$ which is $[f(\boldsymbol{z}, t, \boldsymbol{\theta}), -\boldsymbol{a_z}^T\nabla_{\boldsymbol{z}}f, -\boldsymbol{a_z}^T\nabla_{\boldsymbol{\theta}}f]$. And the gradient is still found as an ODE. However in our method we use one function evaluation to calculate the velocity of $[\boldsymbol{z}, \boldsymbol{a_z}]$ which is $[f(\boldsymbol{z}, t, \boldsymbol{\theta}), -\boldsymbol{a_z}^T\nabla_{\boldsymbol{z}}f]$ and another function evaluation to calculate the term in the quadrature given by $\boldsymbol{a_z}^T\nabla_{\boldsymbol{\theta}}f$. So number of function evaluations is not indicative of time to obtain the gradient. The reason we are faster is because the number of function evaluations does not take into account the time to backpropagate to calculate $\boldsymbol{a_z}^T\nabla_{\boldsymbol{z}}f$, or $\boldsymbol{a_z}^T\nabla_{\boldsymbol{\theta}}f$, as well as generally solving a differential equation with state size $2|\boldsymbol{z}| + |\boldsymbol{\theta}|$ (accounting for $[\boldsymbol{z}, \boldsymbol{a_z}, \boldsymbol{a_\theta}]$) versus one with size $2|\boldsymbol{z}|$ (accounting for $[\boldsymbol{z}, \boldsymbol{a_z}]$). Nevertheless we can make approximate calculations for how many function evaluations are calculated:

The standard adjoint method produces approximately half the number of function evaluations during the backward solve as were performed during the forward solve. This is shown in Figure 3 of Chen et al. (2018) (the original Neural ODE paper). Our method then uses the number of evaluation points introduced by our heuristic $\left\lceil C\frac{\text{NFE}(t_k - t_{k-1})}{(t_K - t_0)} \right\rceil$. In our case we use $C = 0.1$ giving us $\left\lceil \frac{\text{NFE}}{10}\frac{(t_k - t_{k-1})}{(t_K - t_0)} \right\rceil$, and if we assume we aren't working with a time series, then this simplifies to $\left\lceil \frac{\text{NFE}}{10} \right\rceil$. And so the standard adjoint method uses $\frac{\text{NFE}}{2}$ function evaluations and the GQ adjoint uses approximately $\frac{\text{NFE}}{10}$, so in relative terms the GQ method uses $\frac{1}{5}$ the number of evaluations as the standard adjoint.

Note that this is in fact likely an overestimate. In Kidger et al. (2021a) it is demonstrated that if treating $\boldsymbol{a_\theta}$ as an integral rather than differential equation far fewer function evaluations during the backward solve are required. That paper only uses the error estimates of $[\boldsymbol{z}, \boldsymbol{a_z}]$ during the backward solve using an adaptive solver, the error estimate of $\boldsymbol{a_\theta}$ is not included. Similarly since we are only solving the ODE for $[\boldsymbol{z}, \boldsymbol{a_z}]$ the error estimate in our case only comes from there as well. In Kidger et al. (2021a) Figure 1 and Table 1 it is shown that the number of backward function evaluations is significantly fewer in this scenario (40%–62% fewer steps as quoted), and so we can expect even fewer evaluations.

By using a range of 40%–60% fewer evaluations given we can put all of this together. If we use NFE function evaluations during the forward solve:

- The standard adjoint uses 0.5NFE evaluations of $[f(\boldsymbol{z}, t, \boldsymbol{\theta}), -\boldsymbol{a_z}^T\nabla_{\boldsymbol{z}}f, -\boldsymbol{a_z}^T\nabla_{\boldsymbol{\theta}}f]$

- The seminorm adjoint uses between 0.2NFE and 0.3NFE evaluations of $[f(\boldsymbol{z}, t, \boldsymbol{\theta}), -\boldsymbol{a_z}^T\nabla_{\boldsymbol{z}}f, -\boldsymbol{a_z}^T\nabla_{\boldsymbol{\theta}}f]$

- The GQ adjoint uses between 0.2NFE and 0.3NFE evaluations of $[f(\boldsymbol{z}, t, \boldsymbol{\theta}), -\boldsymbol{a}_{\boldsymbol{z}}^T \nabla_{\boldsymbol{z}} f]$ and 0.1NFE evaluations of $\boldsymbol{a}_{\boldsymbol{z}}^T \nabla_{\boldsymbol{\theta}} f$

And so we can see how the GQ method uses fewer evaluations compared to both the standard and seminorm adjoints, its just that this is not captured by the standard NFE measure.

### E.2 Nested Spheres

**Dataset** We consider a two-dimensional nested spheres task, where the classification is given by

$$g(\boldsymbol{x}) = \begin{cases} 0, & 0 \leq ||\boldsymbol{x}||_2 \leq r_1 \\ 1, & r_2 \leq ||\boldsymbol{x}||_2 \leq r_3 \end{cases}. \tag{21}$$

We use the values $r_1 = 0.4$, $r_2 = 0.7$ and $r_3 = 0.9$, 140 example points from the training set are given in Figure 3. The training set is made of 1000 randomly sampled points from each class (making 2000 data points in total). The test set is made of 50 randomly sampled points from each class.

**Model** Our model is an augmented neural ODE (Dupont et al., 2019) with 3 augmented dimensions, zero-augmentation (Massaroli et al., 2020) and a final linear layer to obtain logits. The dynamics function is a time-dependent multilayer perceptron with two hidden layers and softplus activation. The two hidden layers have the same width which we vary to test the methods on different sized models.

**Training** We train the model for 100 epochs, with a batch size of 200, cross entropy loss and learning rates given in Table 1. We repeat the experiment with 10 seeds to obtain means and standard deviations.

Table 1: Learning rates for the nested spheres experiment.

| Hidden Width | Learning Rate |
|:---:|:---:|
| 5 | $5 \times 10^{-3}$ |
| 20 | $5 \times 10^{-3}$ |
| 100 | $1 \times 10^{-3}$ |
| 500 | $1 \times 10^{-3}$ |
| 1000 | $2 \times 10^{-4}$ |
| 1500 | $2 \times 10^{-4}$ |
| 2000 | $1 \times 10^{-4}$ |
| 2500 | $1 \times 10^{-4}$ |
| 3000 | $1 \times 10^{-4}$ |

**Loss Curves** We saw in Section 5.3 that the GQ method's training times scale well with model size compared to the adjoint and seminorm methods, as expected. We also saw that all methods have similar final accuracies demonstrating the gradients produced by the GQ method are accurate. We look further into this by plotting test loss curves during training, these are given in Figure 8.

The loss curves match each other, showing further that the gradients computed with the GQ method are accurate. We also see that in wall clock time the GQ method reduces the loss faster than the adjoint and seminorm methods.

**Effect of C** Next we carry out an ablation study to see the effect of $C$ on the training time and accuracy. We train the same model, with a hidden width of 1500 using the GQ method. We train with different values of $C$ and repeat with 5 seeds to obtain means and standard deviations. The results are given in Table 2.

We see that the time to train is lowest when $C$ is lowest, because the fewest terms are used, however we also see that the final accuracy is compromised as a result. This is what we would expect, the number of terms

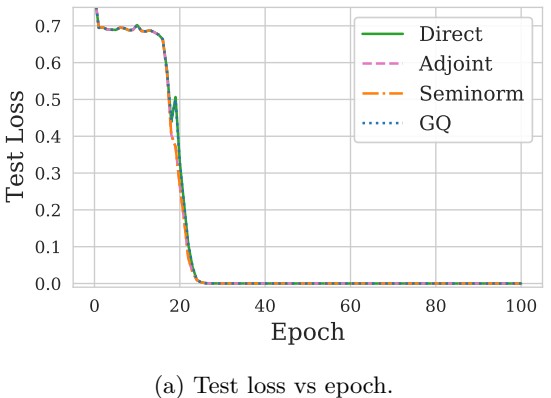

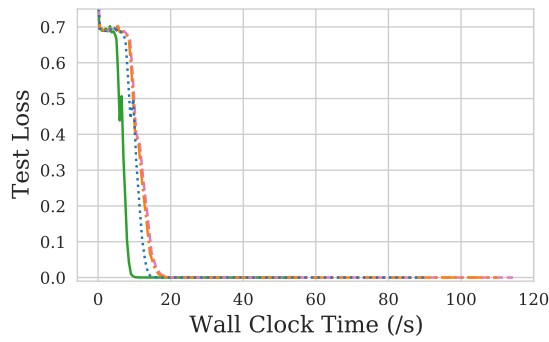

(a) Test loss vs epoch.              (b) Test loss vs wall clock time.

Figure 8: Test loss curves during training of the largest model on the nested spheres task. We see that the loss curves approximately match for all methods and that the GQ method reduces the loss faster than the adjoint and seminorm methods.

Table 2: The effect of the user defined $C$ on training time and final accuracy on the nested spheres experiment. The best values are in bold. When $C = 1 \times 10^2$ and $C = 1 \times 10^5$, both regimes reached the maximum number of quadrature points (64), resulting in the same performance.

| $C$ | Time to Train (/s) | Final Accuracy (%) |
|---|---|---|
| $1 \times 10^{-3}$ | **61.4 ± 3.0** | 79.0 ± 26.9 |
| $1 \times 10^{-1}$ | 89.8 ± 3.1 | 95.6 ± 8.8 |
| $1 \times 10^2$ | 141.0 ± 2.4 | **95.8 ± 8.4** |
| $1 \times 10^5$ | 141.7 ± 2.4 | **95.8 ± 8.4** |

used is not enough so the gradients are inaccurate. We see that using a very high $C$ leads to a slightly higher accuracy than when $C$ is 0.1. However, this is insignificant compared to the increase in training time, and the accuracies are easily within a standard deviation of each other. There is no increase in training time between $C$ being $1 \times 10^2$ and $1 \times 10^5$ because the number of terms in the quadrature is capped at 64.

### E.3 Time Series

**Dataset** Our dataset consists of sine curves, given by the second order ODE $\ddot{x} = -x$. For a given initial position $x(0) = x_0$ and initial velocity $v(0) = v_0$ the position as a function of time is

$$x(t) = x_0 \cos(t) + v_0 \sin(t). \tag{22}$$

To create a trajectory for the dataset, a random position and random velocity are sampled uniformly on $[-1, 1]$, equation (22) is then used to give the position at a given time between 0 and $2\pi$. The training set consists of 15 trajectories and the test set contains 5. For the main experiment we use 50 uniformly spaced measurement times in the range $[0, 2\pi]$ including the end points.

**Model** We use a second order neural ODE as our model (Norcliffe et al., 2020; Massaroli et al., 2020). In this case the model is given the initial position *and* the initial velocity and predicts the acceleration. The acceleration is learnt by a time-independent multilayer perceptron, that takes both position and velocity as input. The MLP has two hidden layers of varying width and softplus activations.

**Training** We train the model for 250 epochs, with a batch size of 15, MSE loss and learning rates given in Table 3. We repeat the experiment with 5 seeds to obtain means and standard deviations.

Table 3: Learning rates for the sines experiment.

| Hidden Width | Learning Rate |
|:---:|:---:|
| 5 | $2 \times 10^{-2}$ |
| 20 | $8 \times 10^{-3}$ |
| 200 | $5 \times 10^{-4}$ |
| 1000 | $3 \times 10^{-5}$ |
| 2000 | $1 \times 10^{-5}$ |

**Loss Curves**   To further test if the GQ method produces accurate gradients, we plot the test MSE curves during training in Figure 9.

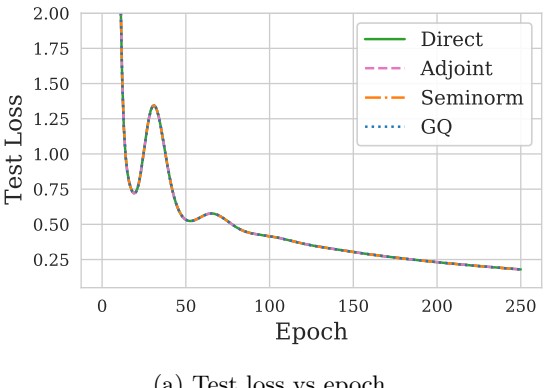 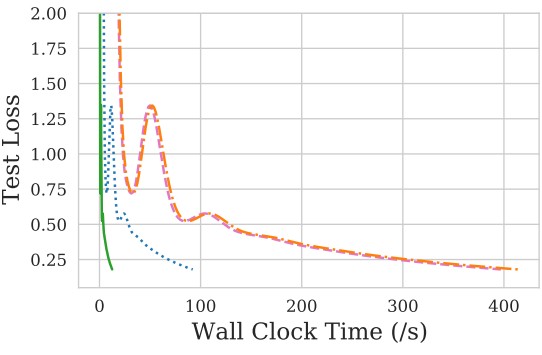

(a) Test loss vs epoch.                  (b) Test loss vs wall clock time.

Figure 9: Test loss curves during training of the largest model on the sines task. We see that the loss curves match for all methods and that the GQ method reduces the loss faster than the adjoint and seminorm methods.

The loss curves match, showing further that the gradients computed with the GQ method are accurate. We also see that in wall clock time the GQ method reduces the loss faster than the adjoint and seminorm methods.

**Effect of Regularity**   Here we carry out an ablation study to investigate the effect of having regular and irregular measurement times during training. We also test the case where there are only 10 measurement times. In order to use batches, all of the initial states are concatenated to one large state, and the ODE runs on the larger system. For time series, the solver outputs a prediction for the whole concatenated state at every unique measurement time and a mask is applied if necessary, so that only the relevant times for each batch are used to calculate the loss. Because we use batch sizes greater than one, having irregular times requires the model to output the predictions at many times, so we would expect this to be slower than when using regular measurement times. We repeat the above experiment using a model with hidden width 1000, using different numbers of measurement times, and changing whether the times are regularly spaced on the domain or not. The results are given in Table 4.

We see that having irregular times increases the time to train significantly, as expected. The final MSEs are not significantly different with the standard deviations overlapping. The irregular MSEs are slightly lower because using irregular training times is effectively giving the model many different measurement times to train on. However, the decrease in MSE is negligible compared to the increase in training time.

Table 4: The effect of regular/irregular times on the sines experiment on training time and final test MSE. The best values are in bold.

| No. Times | Regularity | Time to Train (/s) | Final MSE |
|---|---|---|---|
| 10 | Regular | **48.5 $\pm$ 1.3** | 0.08 $\pm$ 0.07 |
| | Irregular | 239.2 $\pm$ 2.9 | **0.07 $\pm$ 0.06** |
| 50 | Regular | **96.7 $\pm$ 4.9** | 0.11 $\pm$ 0.09 |
| | Irregular | 1880.5 $\pm$ 607.6 | **0.09 $\pm$ 0.08** |

### E.4 Image Classification

**Datasets** We use standard image datasets for the image recognition experiment. We use the MNIST dataset (LeCun et al., 1998), which consists of handwritten digits 0–9. We use the CIFAR-10 dataset (Krizhevsky, 2009) which consists of natural images of 10 classes: Airplane, Automobile, Bird, Cat, Deer, Dog, Frog, Horse, Ship and Truck. And we use the SVHN dataset (Netzer et al., 2011), which is a harder version of MNIST consisting of natural images of the digits 0–9, obtained from house numbers in Google Street View.

**Model** For each dataset, our model consists of initial downsampling layers, a time-independent neural ODE with time domain $[0, 1]$ and fully connected layers, to produce a vector of logits. We vary the number of hidden channels (nhidden) in the neural ODE dynamics function to vary model size. The downsampling layers are given in order below:

- Batch Normalisation
- ReLU activation
- $3 \times 3$ convolution going to 10 channels with no padding
- $2 \times 2$ max pool
- ReLU activation
- $3 \times 3$ convolution going to 10 channels with no padding

The neural ODE dynamics function layers are then given by:

- $3 \times 3$ convolution going from 10 channels to nhidden channels with zero padding
- Softplus activation
- $3 \times 3$ convolution going from nhidden channels to nhidden channels with zero padding
- Softplus activation
- $3 \times 3$ convolution going from nhidden channels to 10 channels with zero padding (must have padding and go to 10 channels so the ODE layers preserve the size of the state)

An important implementation detail is that our implementation only works with vectors. So included in the downsampling layers is a flattening operation at the end, additionally the ODE function has an unflattening operation at the start and a flattening operation at the end.

The fully connected layers at the end of the classifier are given by a one hidden layer multilayer perceptron, with hidden width 50 and a ReLU activation.

**Training** We train the models for 15 epochs, with a batch size of 16 and cross entropy loss. The learning rate used for MNIST was $1 \times 10^{-5}$ and the learning rate used for CIFAR-10 and SVHN was $6 \times 10^{-5}$. We run each experiment using 3 seeds to obtain means and standard deviations. We do not consider the direct method due to memory limitations.

**Additional Results** In addition to the results on MNIST in Section 5.5, we also present further results on CIFAR-10 and SVHN here; this includes loss curves and results for different batch sizes. We first plot the loss curves from the MNIST experiment in Figure 10.

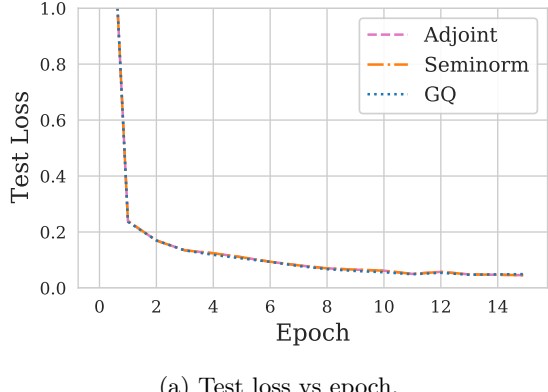
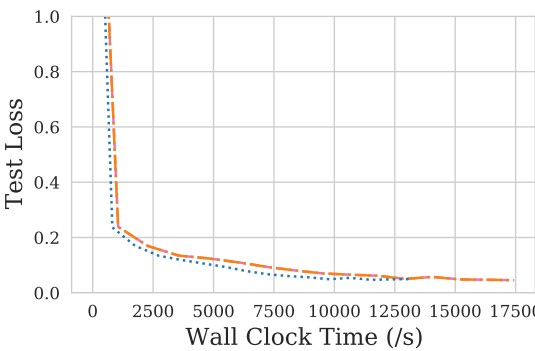

(a) Test loss vs epoch.              (b) Test loss vs wall clock time.

Figure 10: Test loss curves during training of the largest model on the MNIST task using a batch size of 16. We see that the loss curves match for all methods and that the GQ method reduces the loss slightly faster than the adjoint and seminorm methods.

We see that the loss curves match, providing further evidence that the GQ method produces accurate gradients. The GQ method also reduces the loss faster than the adjoint and seminorm methods.

We also carry out the same experiments on the CIFAR-10 and SVHN datasets, results for the CIFAR-10 dataset are given in Figure 11.

We see that the GQ method scales more favourably than the adjoint and seminorm methods for large models, however the effect is far less noticeable than for the nested spheres and sines tasks. The final accuracies are comparable showing the gradients are accurate. In further support of this, the test loss curves match for the methods, with the GQ method reducing the loss faster than the adjoint and seminorm methods.

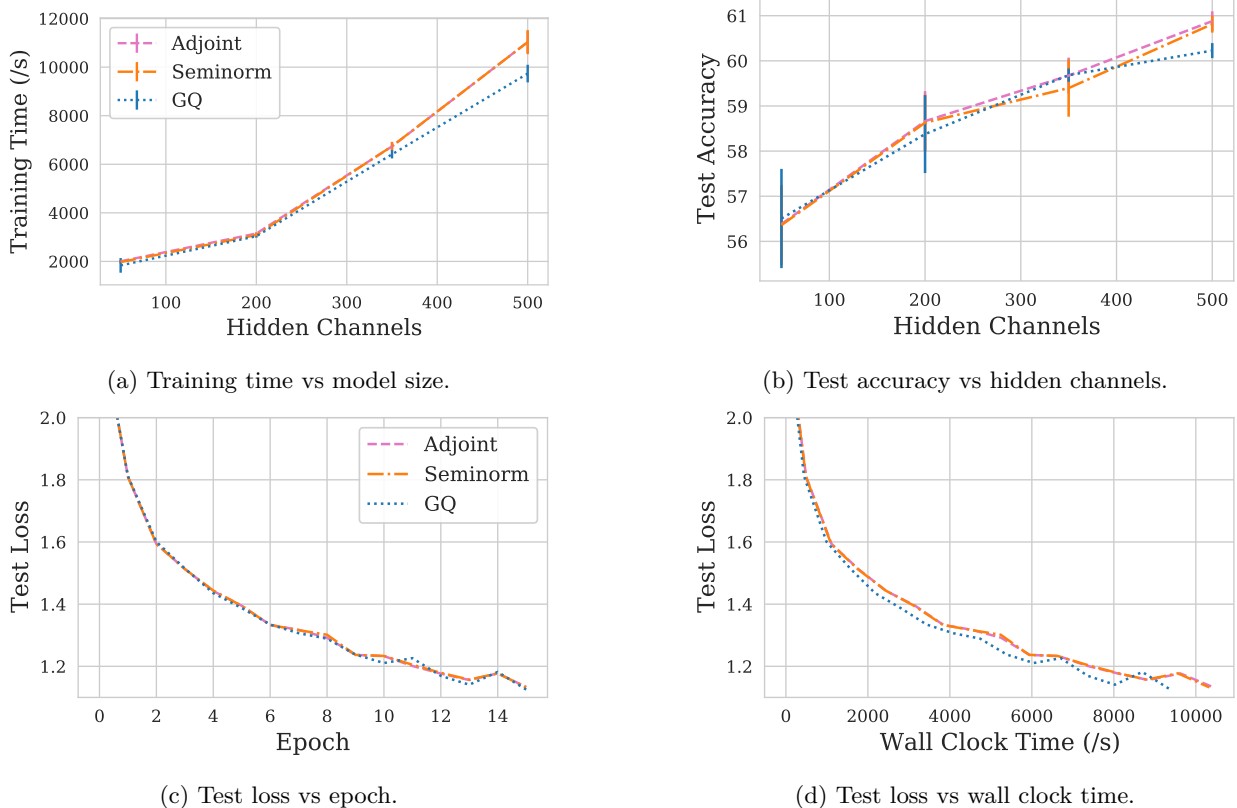

(a) Training time vs model size.

(b) Test accuracy vs hidden channels.

(c) Test loss vs epoch.

(d) Test loss vs wall clock time.

Figure 11: Training times and test accuracies for models of varying size on the CIFAR-10 dataset with a batch size of 16. Loss curves for the largest model are also plotted in the second row. The GQ method scales well to large models, the loss curves in the second row are approximately the same for each method and the GQ method reduces the loss faster than the adjoint and seminorm methods.

We carry out the same experiment on the SVHN dataset with results given in Figure 12.

(a) Training time vs model size.

(b) Test accuracy vs hidden channels.

(c) Test loss vs epoch.

(d) Test loss vs wall clock time.

Figure 12: Training times and test accuracies for models of varying size on the SVHN dataset with a batch size of 16. Loss curves for the largest model are also plotted in the second row. The GQ method scales well to large models, the loss curves in the second row are approximately the same for each method and the GQ method reduces the loss faster than the adjoint and seminorm methods.

We see that the GQ method scales more favourably than the adjoint and seminorm methods for large models. The final accuracies are comparable showing the gradients are accurate. In further support of this, the test loss curves match for the methods, with the GQ method reducing the loss faster than the adjoint and seminorm methods.

**Effect of Batch Size**   In this ablation study we see how the state size affects the training time. The adjoint state is the same size as the state and so as this increases, the effect of using the GQ method will be expected to be less noticeable, as it takes longer to solve the ODE regardless of the model size. We test this by using a larger batch size in the image recognition tasks, we use a batch size of 32 doubling the size of the state. We also look at MNIST with a comparably large batch size of 256. We plot the training times for models of different sizes, in Figure 13.

We see that for larger state sizes the advantage of using the GQ method significantly decreases, only scaling slightly more favourably than the adjoint and seminorm methods. This is because the computational cost to solve the differential equation for $[\boldsymbol{z}, \boldsymbol{a_z}]$ is significantly larger than solving the integral of $\boldsymbol{a_z}^T \nabla_{\boldsymbol{\theta}} f$, regardless of whether it is solved with quadrature or as an ODE. In the extreme case, using a batch size of 256, we actually see that the GQ method is slower than the adjoint and seminorm methods. This is likely because the time taken to calculate terms in the quadrature sum becomes significant when the state is large. Whereas they are automatically calculated when solving it as an ODE, because the backward state is concatenated, allowing $\nabla_{\boldsymbol{\theta}} f$ to be calculated along with $\nabla_{\boldsymbol{z}} f$ in one backward pass of $f$.

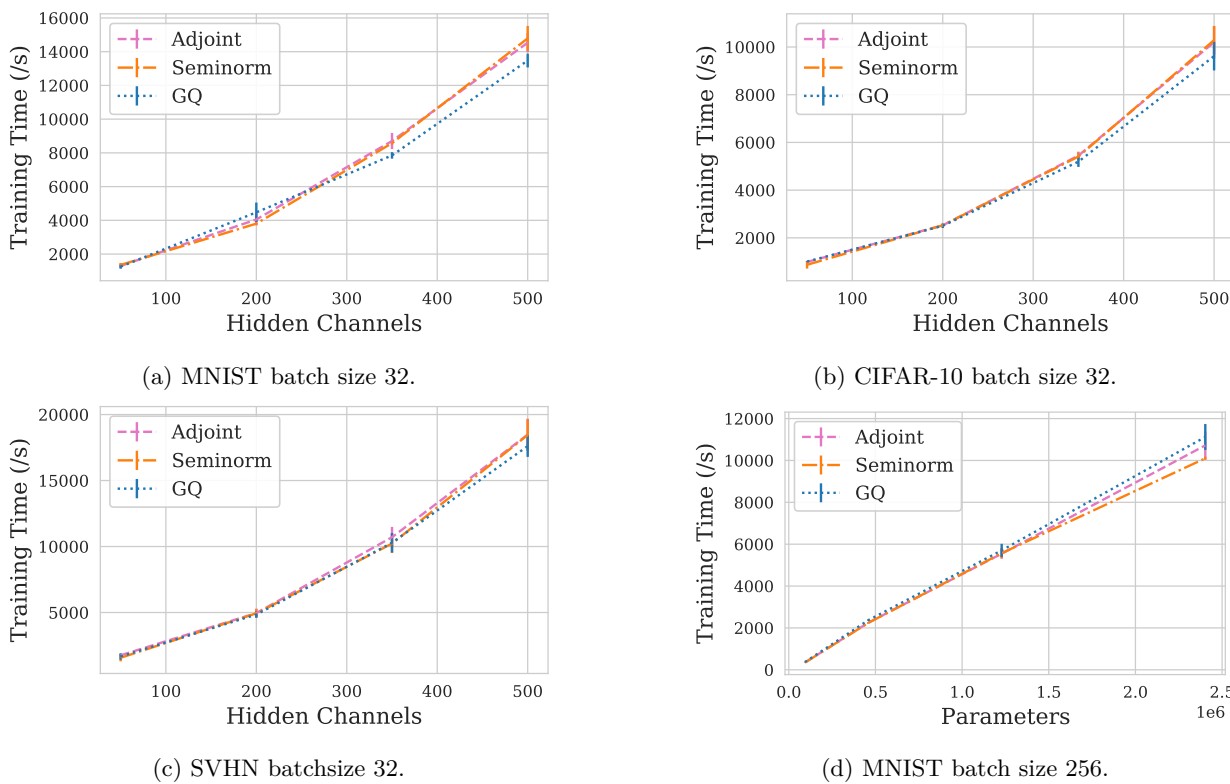

Figure 13: Training times on the image classification tasks for larger batch sizes. For a batch size of 32 the GQ method still scales more favourably, however the effect is significantly reduced. For a batch size of 256, the GQ method is actually slower than the adjoint and seminorm methods.

Therefore, as discussed in Section 6, the GQ method is best applied to situations where memory is limited (otherwise the direct method is best), where the state size is small and the model has many parameters. An example is time-series analysis with complex dynamics.

## F  Training SDEs

**Dataset**   We consider the one-dimensional time-dependent OU process given by the Stratonovich SDE

$$dx = (\mu t - \theta x)dt + (\sigma + \phi t) \circ dW_t, \tag{23}$$

where we have a time dependent drift and diffusion. We use the values $\mu = 0.2$, $\theta = 0.1$, $\sigma = 0.6$, $\phi = 0.15$. The data is made by sampling an initial condition uniformly on $[-3, 3]$, and then solving the SDE for times between $[0, 10]$ with time intervals of 0.1. This is done using the `sdeint` Python library. We solve the SDE 45 times for each initial condition to generate enough samples, the training data consists of 200 initial conditions, and the test data consists of 20 initial conditions.

**Model**   Our neural SDE runs in observation space, so we do not use an encoder or decoder. The drift and diffusion are learnt by two separate, time-dependent, multilayer perceptrons. Each MLP has two hidden layers of varying width and uses softplus activations.

**Training**   We train the model for 100 epochs with a batch size of 40 initial conditions and the approximate KL divergence as the loss as well as evaluation metric. The learning rates for each hidden width are given in Table 5. We run each experiment using 5 seeds to obtain means and standard deviations. We use the reversible-Heun solver (Kidger et al., 2021b) to produce reversible solves (Zhuang et al., 2021), with a step size of 0.01. We use the same solver to evaluate the parameters trained by the GQ method.

Table 5: Learning rates for the Ornstein Uhlenbeck experiment.

| Hidden Width | Learning Rate |
|:---:|:---:|
| 5 | $1 \times 10^{-3}$ |
| 20 | $5 \times 10^{-4}$ |
| 50 | $1 \times 10^{-4}$ |
| 100 | $1 \times 10^{-4}$ |

**Loss Curves**   To further test if the GQ method produces accurate gradients for neural SDEs, we plot the test KL divergences during training in Figure 14.

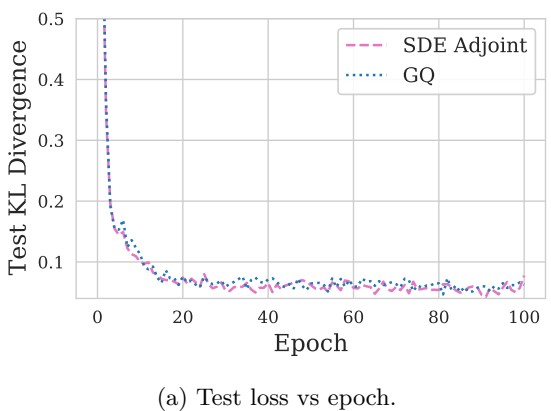

(a) Test loss vs epoch.

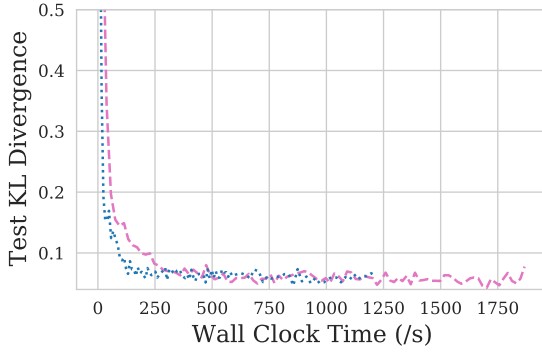

(b) Test loss vs wall clock time.

Figure 14: Test loss curves during training of the largest model on the OU task. We see that the loss curves match for both methods and that the GQ method reduces the test loss slightly faster than the SDE adjoint method.

We see that the loss curves approximately match, they don't exactly match due to the stochastic element of the task. The approximate match shows that the GQ method and the Wong-Zakai theorem produces accurate gradients and can be used to train neural SDEs. Additionally, the GQ method reduces the loss faster than the SDE adjoint method.

**Effect of Number of Cosines**   Next we consider how the number of cosines affects the results, with fewer cosines the ODE solution is a worse approximation to the SDE solution, so we would expect a larger KL divergence. However, with fewer cosines each function evaluation is faster, so we would also expect training to be faster. The important case is when there are no cosines used, because only the drift is being trained. We test this by carrying out the previous experiment with hidden width 20, using different numbers of cosines to represent the Wiener process. The training times and test KL divergences are given in Table 6.

We see what is expected, the training time increases with the number of cosines used and the KL divergence decreases with the number of cosines. We see that for more than 5 cosines, the improvement in performance is small compared to the increase in training time.

**Learnt Solutions**   Finally, to see how the trained model performs, we visualise some of the learnt solutions in Figure 15 when the hidden width is 20. The learnt distribution matches the true distribution well. The learnt samples appear like they could have been sampled from the true distribution, providing qualitative evidence that the GQ method works for training SDEs.

Table 6: Effect of number of cosines on the OU experiment. The best values are in bold.

| No. Cosines | Training Time (/minutes) | Mean KL Divergence ($\times 10^{-2}$) |
|---|---|---|
| 0 | $\mathbf{9.62 \pm 0.09}$ | $75.55 \pm 32.30$ |
| 1 | $11.34 \pm 0.04$ | $28.01 \pm 2.48$ |
| 5 | $12.97 \pm 0.06$ | $7.49 \pm 0.86$ |
| 10 | $19.88 \pm 6.23$ | $6.50 \pm 0.74$ |
| 25 | $33.13 \pm 0.96$ | $6.18 \pm 0.68$ |
| 50 | $51.97 \pm 0.69$ | $\mathbf{6.12 \pm 0.66}$ |

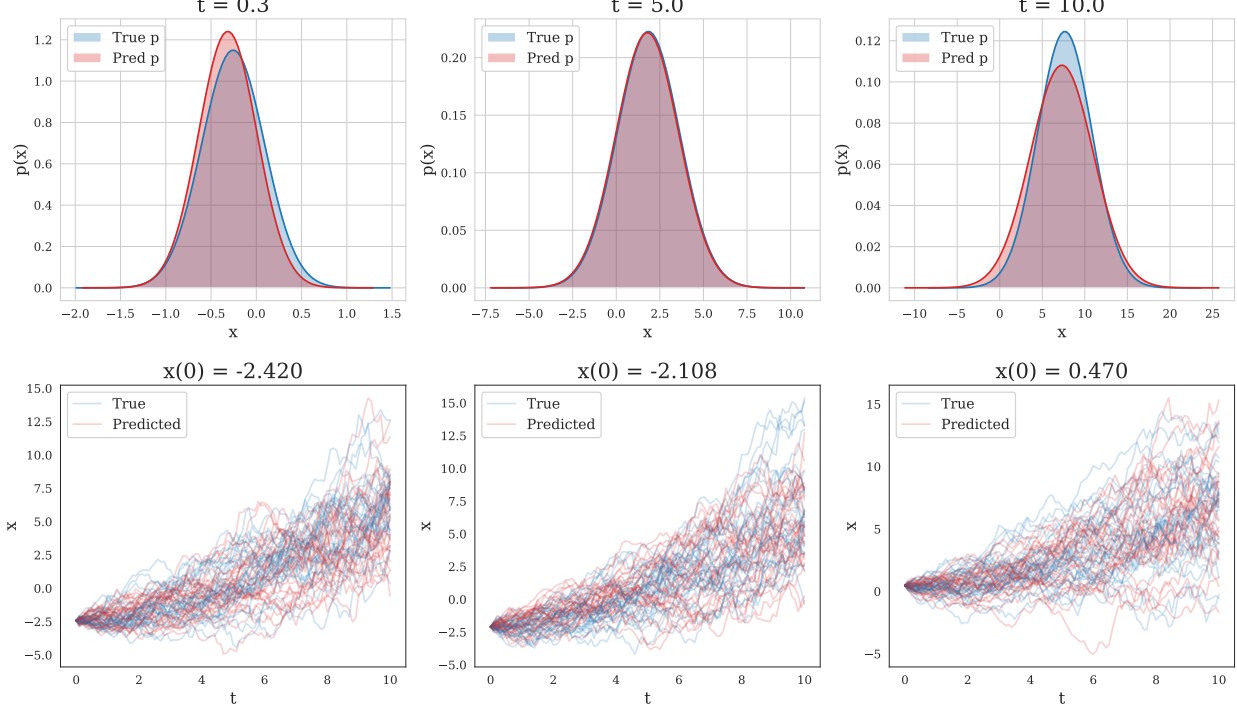

Figure 15: SDE predictions using the trained model, with a hidden width of 20, on the OU experiment. In the top row we plot Gaussian distributions, where the mean and standard deviation are calculated from the samples. These are plotted for given times from the same initial condition. We see that the predictions match well. In the bottom row are samples from three different initial conditions. We see that the predicted samples are similar to the true samples from the SDE.

# G Crossing Trajectories

Finally we consider the additional task of the crossing trajectories problem Massaroli et al. (2020), originally called $g_{1d}$ in Dupont et al. (2019). It represents a simple problem that vanilla Neural ODEs cannot solve. The problem is to map $[[+1], [-1]]$ to $[[-1], [+1]]$, using an ODE. In one dimension, this is impossible because ODE trajectories cannot cross.

The standard solution to this problem is to augment the state Dupont et al. (2019), this is where the initial condition has its dimensionality increased with zeros

$$\mathbf{z}_0 = \begin{bmatrix} \mathbf{x}_0 \\ \mathbf{0} \end{bmatrix}. \tag{24}$$

The new state $\mathbf{z}$ is then evolved according to a new ODE and can be projected back into the observation space by directly selecting the relevant dimensions, alternatively a projection could be learnt as a single linear layer or a more complex function.

We use a state with 3 augmented dimensions, 4 in total. We train the model for 200 epochs and record the time as well as the MSE during training, using the MSE as a loss function. For each model this is repeated 10 times to obtain means and standard deviations, using the same seeds between methods for fair comparison.

The ODE function is a multi-layer perceptron, with two hidden layers of varying width, and a softplus activation. After the ODE evolution we then select the first dimension of the augmented state. We use different learning rates depending on the width of the hidden layers which are given in Table 7.

Table 7: Learning rates for the crossing trajectories experiment.

| Hidden Width | Learning Rate |
| --- | --- |
| 5 | $1 \times 10^{-2}$ |
| 20 | $1 \times 10^{-2}$ |
| 100 | $1 \times 10^{-3}$ |
| 500 | $1 \times 10^{-3}$ |
| 1000 | $7 \times 10^{-4}$ |
| 1500 | $1 \times 10^{-4}$ |
| 2000 | $1 \times 10^{-4}$ |
| 2500 | $1 \times 10^{-4}$ |
| 3000 | $1 \times 10^{-4}$ |

The results of the time taken and final MSE for each hidden width, are presented in Figure 16. We can see that the direct method is the fastest, followed by the GQ method. The seminorm method is slightly faster on the whole than the standard adjoint method as expected. In particular we can see that the GQ method scales well to the number of parameters, whereas the other two adjoint methods do not. This is because during the backwards solve, the two adjoint methods calculate $\mathbf{a}_{\mathbf{z}}^T \frac{\partial f}{\partial \theta}$ at every function evaluation (with the seminorm method taking fewer steps). The GQ method on the other hand only calculates this term for evaluation points in the quadrature calculation, and therefore scales better to larger models. This also explains why the training time for the adjoint methods appears to scale approximately linearly with the number of parameters. For small models the time improvement is negligible.

We see that the final MSEs are the same, which are all low showing that the problem has been solved. The only exception being for the largest model where the learning rate may have been too high. We see that the losses during training match, so the gradients produced by the GQ method are (approximately) the same as the other methods. We also see that the loss is lowered in a shorter wall clock time using the GQ method than the other adjoint methods as expected.

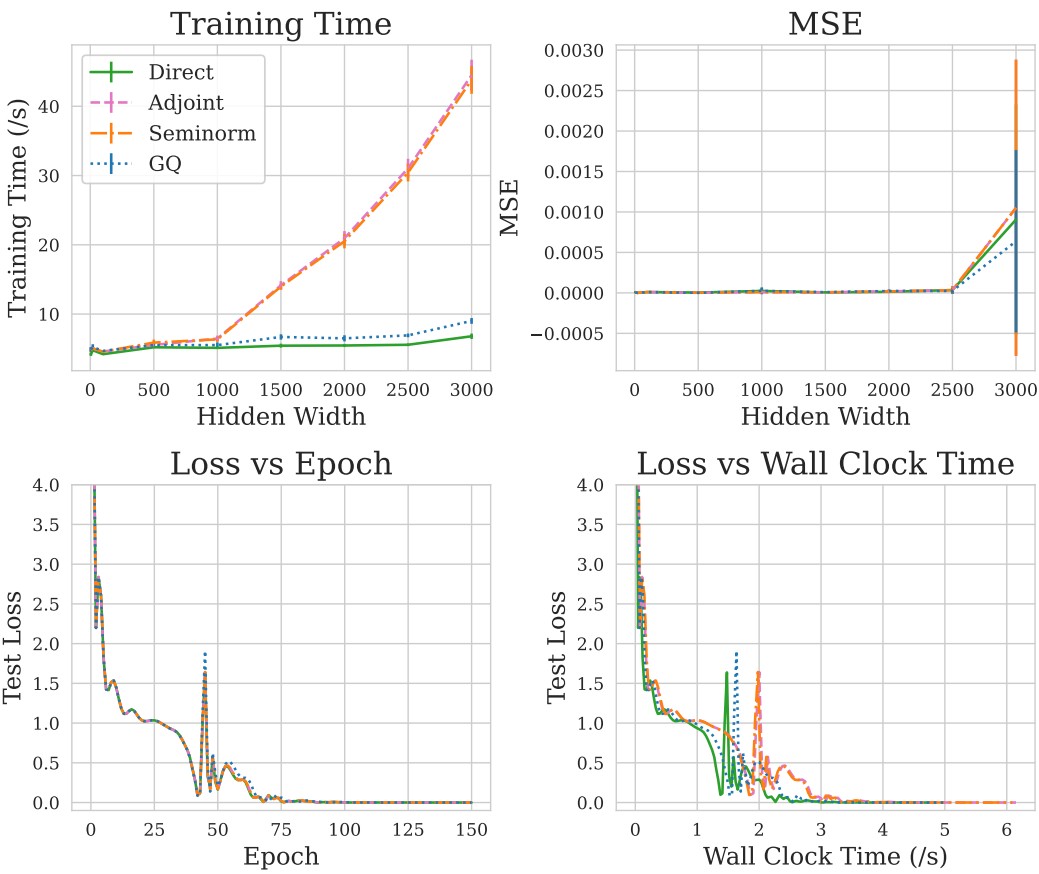

Figure 16: Training times and MSEs for the crossing trajectories task. The GQ method's training time scales well with model size compared to the adjoint methods. The test MSEs are the same. In the second row we plot the loss during training for the largest model. We see the loss curves all match and that the GQ method reduces the loss faster than the adjoint methods.

