# OpenReview forum: "Faster Training of Neural ODEs Using Gauß–Legendre Quadrature"
_TMLR — Accepted by TMLR_

### Review · Reviewer_8wu8 · 2023-04-28

**Summary Of Contributions:**


Training Neural ODEs is slow since they require numerical solvers. A common approach to speed Neural ODEs up is to regularise them, however this can also affect the expressivity of the model (particularly in time-series modelling, where the trajectory itself matters).

This paper proposes a method to make training Neural ODEs with the adjoint method faster. The method of Gauß–Legendre Quadrature is used for numerical integration, instead of using solvers. This approach is also extended into a new way to train Neural SDEs.

Section 2 of the paper gives a clear explanation of Neural ODEs and training them using the adjoint method. It then goes on to describe the work of Kidger, which points out that the adjoint equation for gradients with respect to the Neural ODE parameters can be seen as an integral. The terms of the integrand are the adjoint with respect to z and the gradient of the dynamics with respect to the parameters. In particular, there is no term in the integrand for the adjoint with respect to the parameters theta. So, we can solve this 1D integral using quadrature methods in-parallel across the dimensions of the parameters.

Instead of requiring a dense solution, the authors propose fixing the number of quadrature points in advance and maintaining a running sum. This avoids high memory consumption that other adjoint quadrature methods suffer from.

The authors propose a reasonable empirical heuristic for choosing the number of quadrature points. This heuristic requires the choice of a constant, for which an ablation study is done. There could be an experiment done to directly justify the choice of the form of this heuristic (i.e. comparing the NFEs on the forward and backwards passes when using the adjoint method), but I don’t think it is absolutely necessary since similar results have already been shown in other works.

The paper also extends their approach to training Neural SDEs, where memory requirements can be especially high. They make use of the Wong–Zakai theorem, so that an ODE can be numerically solved instead of an SDE. In particular, this approach requires a smooth and memory efficient approximation of the Wiener process.

The first experiment in the paper demonstrates that the GQ method is just as accurate as the adjoint method, and has similar (constant) memory requirements.

Secondly, they show these benefits carry over to making the training time faster on a toy spheres experiment.

There are additional experiments for image classification, time series, and SDEs.


**Audience:**

Yes

**Broader Impact Concerns:**

I have no concerns here.

**Claims And Evidence:**

Yes

**Requested Changes:**

The finer details of the approach for Neural SDEs needs further explanation in my opinion. I think having pseudocode in the Appendix (similar to that for the method for ODEs) would be very helpful.

The authors describe GQ methods in some detail, but I feel there could be further explanation given here. In particular, it would be helpful to describe how the weights need to be shifted to change to a general interval.

The experiment on time-series models demonstrates the method works well when there are intermediate points of integration. I would be interested to see if there were any ablations done on the number of integration points for GQ for time series modelling tasks, as I imagine the number of integration points might vary more between different examples in the dataset.

In section 5.6, can you clarify “We do not see the previous effect of model size on training time because we are still in the ‘small parameter regime’, which is why the training time fluctuates rather than increasing with model size.”

Are there any experiments where you compare the NFE on the backwards pass with the number of integration points? I know they are not directly comparable, since the NFE in the adjoint method includes terms for both df/dz and df/dtheta. But it would be nice to see that data directly, since it could disentangle total time spent from the memory requirements needs to solve the adjoint. This seems like it could be related to the ablation experiment done that justifies the choice of constant C in Equation 9, perhaps?


**Strengths And Weaknesses:**

The paper clearly communicates the basics of Neural ODEs and the existing literature. It clearly explains where existing methods fail, and how their approach tackles some of those issues.

The experiment on image classification demonstrates their method performs favourably in terms of training time on a larger scale. However, it seems that the GQ method only provides a marginal improvement over other methods.

---

> ### Author Response · Authors · 2023-05-11
> **Official Response to Reviewer 8wu8**
>
> We’d like to start by sincerely thanking the reviewer for their time and effort in reviewing our paper. We are glad you found our paper clear and that the experiments demonstrate our claims. We are grateful for the constructive feedback summarised into five points:
>
> 1. The explanation of Neural SDE should be expanded.
> 2. The explanation of shifting the integration interval should be extended.
> 3. An ablation on number of integration points in the time-series would be beneficial.
> 4. Small Parameter Regime.
> 5. Comparing backward NFE to number of integration points.
>
> We happily answer all of these points in separate replies so that they can be discussed individually. We have also updated the manuscript with changes, we look forward to your reply.

---

> > ### Author Response · Authors · 2023-05-11
> > **1. Explanation of SDE**
> >
> > Thank you for pointing this out and we absolutely agree it would help the clarity of the paper. We have added a description to the paper, it is now in Appendix B. The summary is that training the SDE is very similar to algorithm 1 for Neural ODEs or algorithm 2 for Neural ODEs with time-series.
> >
> > The big difference is that for standard Neural ODEs the rate of change of the state is given by
> >
> > $\frac{d\mathbf{z}}{dt} = f(\mathbf{z}, t, \theta)$,
> >
> > whereas now we have both a drift $f$ and diffusion $g$, so the rate of change of the state is given by
> >
> > $\frac{d \mathbf{z}}{dt} = f( \mathbf{z}, t, \theta) + g( \mathbf{z}, t, \theta)
> > \Biggr[
> > \sqrt{\frac{2}{t_K - t_0}}\sum_{i=1}^{m}\mathbf{\xi}_{i} \cos\left(\frac{(i-\tfrac{1}{2})
> > \pi (t-t_0)}{t_K - t_0}\right)
> > \Biggr].$
> >
> > Here the vectors $\mathbf{\xi}_i$ are sampled from a standard normal. We train this as an ODE using our GQ method, i.e. directly subsitituing the above rate of change into algorithm 1 or 2, and then at test time we can transfer the drift and diffusion to an SDE solver, so that at test time we are solving the SDE
> >
> > $d\mathbf{z} = f(\mathbf{z}, t, \theta)dt + g(\mathbf{z}, t, \theta)dW_t$
> >
> > for Weiner process $W_t$.
> >
> > The reason we can do this is that using the Wong-Zakai and Karhunen-Loeve theorems, we can approximate the SDE
> >
> > $d\mathbf{z} = f(\mathbf{z}, t, \theta)dt + g(\mathbf{z}, t, \theta)dW_t$
> >
> > with an ODE
> >
> > $\frac{d \mathbf{z}}{dt} = f( \mathbf{z}, t, \theta) + g( \mathbf{z}, t, \theta)
> > \Biggr[
> > \sqrt{\frac{2}{t_K - t_0}}\sum_{i=1}^{m}\mathbf{\xi}_{i} \cos\left(\frac{(i-\tfrac{1}{2})
> > \pi (t-t_0)}{t_K - t_0}\right)
> > \Biggr]$.
> >
> > This approximation improves with larger $m$. An important caveat is that for each solve we must use the same $\mathbf{\xi}_i$, these can either be iteratively sampled and included in the sum, reseeding the generator each time, to save even more memory, or they can be sampled and stored for each solve.
> >
> > As mentioned, the approximation improves with more terms in the cosine expansion. In Table 6 in the Appendix we look further into this and see that for our experiment the tradeoff between time to train and accuracy of the final solution happens at 10 terms in the expansion.
> >
> > This table is included below for the reviewer's benefit:
> >
> > | No. Cosines | Training Time (/minutes) | Mean KL Divergence ($\times10^{-2}$) |
> > | --- | --- |  --- |
> > | 0 | 9.62 $\pm$ 0.09 | 75.55 $\pm$ 32.30 |
> > |1 | 11.34 $\pm$ 0.04 | 28.01 $\pm$ 2.48 |
> > |5| 12.97 $\pm$ 0.06 | 7.49 $\pm$ 0.86 |
> > |10 | 19.88 $\pm$ 6.23 | 6.50 $\pm$ 0.74|
> > | 25 | 33.13 $\pm$ 0.96 | 6.18 $\pm$ 0.68 |
> > | 50 | 51.97 $\pm$ 0.69 | 6.12 $\pm$ 0.66|

---

> > > ### Comment · Reviewer_8wu8 · 2023-06-12
> > > **Response**
> > >
> > > Thanks for adding this explanation in!

---

> > ### Author Response · Authors · 2023-05-11
> > **2. Shifting the Integration Interval**
> >
> > Thank you for pointing this out, and again we agree this would benefit the paper. We have included a paragraph on this in Section 3.1 titled "Shifting Integration Domain". The summary is that the weights and position in Gauß–Legendre quadrature are only used in the $[-1, 1]$ integration interval, whereas our general interval is defined on $[t_0, t_K]$. Fortunatley we can linearly scale and shift our integrand so that we work on the $[-1, 1]$ domain.
> >
> > Consider the integral given by
> >
> > $I = \int_{t_0}^{t_K}f(t)dt,$
> >
> > by using the change of variables $\tilde{t} = \frac{2}{t_K - t_0}t - \frac{t_K + t_0}{t_K - t_0}$ we can rewrite the integral as
> >
> > $ \int_{t_0}^{t_K}f(t)dt = \int_{-1}^{1}
> > f
> > \bigg(
> > \frac{t_K-t_0}{2}\tilde{t}+\frac{t_K+t_0}{2}
> > \bigg)
> > \frac{t_K-t_0}{2}
> > d\tilde{t}$
> >
> > And this can then be approximated using Gauß-Legendre quadrature
> >
> > $\int_{t_0}^{t_K}f(t)dt \approx \sum_{i=1}^{n}w_i f
> > \bigg(
> > \frac{t_K-t_0}{2}\tau_i+\frac{t_K+t_0}{2}
> > \bigg)
> > \frac{t_K-t_0}{2},$
> >
> > where $w_i$ and $\tau_i$ are the standard weights and locations for the $[-1, 1]$ domain. This allows us to use Gauß-Legendre quadrature in our setting.
> >
> > As we describe in the paragraph “Why Gauß-Legendre Quadrature?” in section 3.1, this is one of the reasons we have to use Gauß-Legendre quadrature, since we are working on finite intervals with the end points included in the limits of integration. The other reasons are that Gaussian Quadrature schemes solve the integral $\int_a^b w(t) f(t) dt$, where w(t) is a weight function, we use $w(t) = 1$ restricting us to Gauß-Legendre, Gauß-Lobatto and Gauß-Chebyshev. Gauß-Lobatto converges slower than Gauß-Legendre, and Gauß-Chebyshev requires many points in the integral to be stored in memory for adaptive computation, removing the memory efficiency and the advantages of using the adjoint method over direct backprop.

---

> > > ### Comment · Reviewer_8wu8 · 2023-06-12
> > > **Response**
> > >
> > > Thanks for clarifying this.

---

> > ### Author Response · Authors · 2023-05-11
> > **3. Number of points in a Time-Series**
> >
> > In section C.2 of the Appendix there is an ablation looking at the effects of regularity on the same time-series. We look at four versions of the time series, with 10 time measurements or 50 and with those points being regular or irregular for each sample. When we have 50 time points we have more intermediate integrals and therefore more integration points. Additionally, when those points are irregular we have significantly more integration points - since we concatenate and sort all time points in a batch and apply a prediction mask to calculating the loss. And so from 10 regular points we have three ablations, having 50 regular points, having 10 irregular points and 50 irregular points. The increase in training time is significantly larger when using irregular points, and as expected when we have more measurement points the training time increases.
> >
> > For the  reviewer's benefit we include this table below, it can be found in Appendix C.2 (originally it was Appendix B.2 but we have updated the manuscript):
> >
> > | No. Times | Regularity | Time to Train (/s) | Final MSE   |
> > |-----------|------------|--------------------|-------------|
> > | 10        | Regular    | 48.5 ± 1.3         | 0.08 ± 0.07 |
> > | 10        | Irregular  | 239.2 ± 2.9        | 0.07 ± 0.06 |
> > | 50        | Regular    | 96.7 ± 4.9         | 0.11 ± 0.09 |
> > | 50        | Irregular  | 1880.5 ± 607.6     | 0.09 ± 0.08 |

---

> > > ### Comment · Reviewer_8wu8 · 2023-06-12
> > > **Response**
> > >
> > > Thanks for pointing me to this table, this resolves my concern here.

---

> > ### Author Response · Authors · 2023-05-11
> > **4.  Small Parameter Regime**
> >
> > When carrying out the standard Adjoint method, the general integral being solved backwards in time consists of the state $[z, a_z, a_\theta]$, where $z$ and $a_z$ are both the size of the fully batched state, $a_\theta$ is the size of the number of parameters. When using the GQ method, the backward ODE solve involves only $[z, a_z]$ where we use Gauß-Legendre quadrature to calculate the integral involving $a_\theta$. Therefore, when the number of parameters is large, the computational bottleneck is calculating the integral using $a_\theta$ for the GQ method and the ODE for $a_\theta$ for the standard adjoint method. But when the number of parameters is small the bottleneck is in calculating the ODE solve involving $[z, a_z]$. And so here we see values using the GQ method fluctuating since the model size is not causing the computational bottleneck. We also see for the SDE adjoint the training time does not increase for the same reason, the computational bottleneck is in the $[z, a_z]$ part of the backward solve rather than $a_\theta$. However, importantly we see here the direct backprop method runs out of memory and the standard adjoint was still slower. This point is made in section 6, that the GQ method is best applied when the model has many parameters and the state size (including batchsize) is small.

---

> > > ### Comment · Reviewer_8wu8 · 2023-06-12
> > > **Response**
> > >
> > > Thanks for the clarification here, that makes sense to me now. And thanks for pointing me to where this is explained in the paper.

---

> > ### Author Response · Authors · 2023-05-11
> > **5. Backward NFE vs Number of Integration Points**
> >
> > The backward NFE is related to the forward NFE, as shown in Figure 3 of Chen et al. 2018 (the original Neural ODE paper) the backward NFE is approximately half the forward NFE. Conversely the number of integration points is decided by our heuristic which is dependent on the forward NFE. Whilst technically they are not independent since they both depend on the forward NFE, for our purposes they are effectively independent. This is because when we choose the constant $C$ for the heuristic, this does not affect the number of backward NFE, since the backward ODE is solved without the GQ integral in mind.
> >
> > A very good related work is “'Hey, that’s not an ODE': Faster ODE Adjoints via Seminorms" by Kidger et al. 2020. The main work of that paper is only using the error estimation in $z$ and $a_z$ when using an adaptive solver, and not from $a_\theta$ during the backward solve. This means the backward NFE there are only coming from error estimates of $[z, a_z]$. Similarly since we are only solving the ODE for $[z, a_z]$ the backward NFE in our case only comes from there as well. With that in mind we point to Figure 1 from that paper, where the number of backward NFE significantly decreases when using their seminorms. Therefore we see that not only does our method require fewer evaluations for calculating the $a_\theta$ integral compared to the backward $a_\theta$ ODE (with our choice of $C$ it uses approximately 1/10 the number of forward NFE), but it also decreases the number of backward NFE since we do not consider the error estimates in $a_\theta$ during the backward solve.
> >
> > For this reason we do not believe  an ablation is necessary since it would be showing already known results.
> >
> > # References
> >
> > 1. Chen, R.T., Rubanova, Y., Bettencourt, J. and Duvenaud, D.K., 2018. Neural ordinary differential equations. Advances in neural information processing systems, 31. https://arxiv.org/abs/1806.07366
> >
> > 2. Kidger, P., Chen, R.T. and Lyons, T.J., 2021. " Hey, that's not an ODE": Faster ODE Adjoints via Seminorms. In ICML (pp. 5443-5452). https://arxiv.org/abs/2009.09457

---

> > > ### Comment · Reviewer_8wu8 · 2023-06-12
> > > **Response**
> > >
> > > Makes sense, thanks for pointing me to the figure in the reference.

---

> > ### Comment · Reviewer_8wu8 · 2023-06-12
> > **Response**
> >
> > Thank you for the detailed response. All of my concerns have been addressed, no further issues from me to raise.

---

> > > ### Author Response · Authors · 2023-06-13
> > > **Many thanks**
> > >
> > > Many thanks for taking the time to review and discuss the points. We're glad we have addressed all of your concerns.

---

### Review · Reviewer_4ho2 · 2023-06-02

**Summary Of Contributions:**

The authors propose a more efficient method for computing gradients of neural ODEs. They use the observation that the time derivative of one of the terms ($a_\theta$) in the "reverse time" ODE depends only on other terms and not on its own value. As a result, solving for $a_\theta(t_0)$ can be approached as a quadrature problem rather than a differential equation. This allows them to evaluate $a_\theta$ only at a subset of well-chosen quadrature points. They apply their method to a range of problems, evaluating its training speed and memory efficiency.

**Audience:**

Yes

**Claims And Evidence:**

Yes

**Requested Changes:**

- Some of the discussion of quadrature methods is imprecise, e.g. "Gaussian quadrature is the fastest method for 1-D integrals...". This should be replaced with a precise statement about the convergence and efficiency of GL quadrature.
- The speed up in this method is due to fewer evaluations of $\frac{da_\theta}{dt}$. How many evaluations does the conventional adjoint method perform? How many does this method save?
- In figure 2, why does GQ use _more_ memory than the adjoint method? Its state is the same size or smaller during the reverse time solve, isn't it?
- The authors point out that the true gradient is unknown, so it is difficult to assess accuracy. However, since their method is almost identical to the adjoint method except that it evaluates $\frac{da_\theta}{dt}$ at only a subset of points, we can't expect the GQ method to be _more_ accurate than the adjoint in general. It seems reasonable to treat deviation from the adjoint method gradient as the "error" caused by the GQ method. Again, it would be nice to see a plot of this error as a function of the number of quadrature points.

**Strengths And Weaknesses:**

### Strengths

- Neural ODEs and similar methods, such as diffusion models, are widely used but can be memory intensive and slow to train. Improvements in training efficiency are very useful, so the paper is well-motivated.
- The key observation that $a_\theta$ can be evaluated as a definite integral rather than an ODE is well explained, as is their proposed method.


### Concerns

- The authors provide a large set of experiments demonstrating their method, but I think a more focused analysis of a smaller set of experiments would provide more insight.
- The speedup in this method comes from evaluating $\frac{da_\theta}{dt}$ at only a subset of points. As with any quadrature problem, there is a tradeoff between accuracy and the number of function evaluations, but this tradeoff isn't thoroughly explored in the paper. They propose a heuristic for choosing the number of points in equation (9), but given that this is the crux of the method, it would be nice to see the more analysis of speed/accuracy trade-off.
- Some of the plots are very unclear. Figure 1 in particular is difficult to read---the results from their own method are almost invisible! It's also not clear to me what is being plotted here. What does 'Relative Error in Loss' mean? I think this plot in particular needs a lot of clarification.

---

> ### Author Response · Authors · 2023-06-06
> **Official Response to Reviewer 4ho2**
>
> To begin, we'd like to express our sincere gratitude to the reviewer for taking the time and effort to review our paper. We are glad you found the work well motivated and generally well explained. We are grateful for the constructive feedback summarised into these points:
>
> 1. Benefits of a smaller experiment
> 2. Speed vs accuracy tradeoff
> 3. Clarity around Figure 1
> 4. Further discussion around integration methods, particularly error bounds
> 5. Number of $\frac{da_\theta}{dt}$ evaluations
> 6. Peak memory usage in Figure 2
> 7.  Deviation from adjoint method as error
>
> We happily answer all of these points in separate replies so that they can be discussed individually. We have included all of these in an updated manuscript but have chosen to upload the updated manuscript after all reviews have been received as is recommend in the TMLR Submission Guidelines: "Authors can respond to a review as soon as it is posted, however we recommend waiting until all 3 reviews have been submitted before submitting any revised version of the PDF manuscript." https://jmlr.org/tmlr/editorial-policies.html

---

> ### Author Response · Authors · 2023-06-06
> **1. Benefits of a Smaller Experiment**
>
> We thank the reviewer for the suggestion, and as such we have run a further experiment on the $g_{1d}$ task from Dupont et al. 2019, we use the name Crossing Trajectories given by Massaroli et al. 2020. The task is to map $[[-1], [+1]]$ to $[[+1], [-1]]$, that is we have two trajectories that have to cross over. The original aim was to show with an elegant example that there are some tasks vanilla Neural ODEs cannot solve, we now use it as another experiment to show our GQ Adjoint is faster than the Standard Adjoint while achieving the same results. We use an Augmented Neural ODE (Dupont et al. 2019) with three augmented dimensions. The dynamics function is a two hidden layer MLP with softplus activations and varying hidden width. We train for 200 epochs and report the training times in seconds.
>
> In the updated manuscript we include a figure showing that as the hidden width increases the training time for GQ Adjoint scales more favourably than the other adjoint methods whilst achieving the same results. For now we include these times as a table below giving training times:
>
> | Method |100 | 500 | 1000 | 1500 | 2000 | 2500 | 3000 |
> | ---------- |  ----- | ----- | ------- | ------- | ------- | ------- | ------- |
> |Direct| $4.21 \pm 0.11$ |$5.20 \pm 0.05$ |$5.12 \pm 0.13$ |$5.44 \pm 0.29$ |$5.46 \pm 0.22$ |$5.57 \pm 0.19$ |$6.79 \pm 0.39$ |
> |GQ| $4.68 \pm 0.06$ |$5.51 \pm 0.28$ |$5.52 \pm 0.39$ |$6.69 \pm 0.39$ |$6.49 \pm 0.44$ |$6.92 \pm 0.27$ |$9.01 \pm 0.42$ |
> |Adjoint| $4.59 \pm 0.10$ |$5.54 \pm 0.29$ |$6.46 \pm 0.43$ |$14.15 \pm 0.61$ |$20.95 \pm 1.03$ |$31.04 \pm 1.38$ |$44.50 \pm 2.22$ |
> |Seminorm| $4.58 \pm 0.07$ |$5.88 \pm 0.43$ |$6.34 \pm 0.34$ |$13.97 \pm 0.45$ |$20.52 \pm 1.04$ |$30.42 \pm 1.27$ |$43.80 \pm 2.00$ |
>
> We see that as the hidden width increases the Direct method scales most favourably as we would expect. After that the GQ method scales most favourably as we would expect, showing our method scales more favourably than the Standard and Seminorm Adjoints.
>
> # References
> 1. Dupont, E., Doucet, A. and Teh, Y.W., 2019. Augmented neural odes. Advances in neural information processing systems, 32. https://arxiv.org/abs/1904.01681
>
> 2. Massaroli, S., Poli, M., Park, J., Yamashita, A. and Asama, H., 2020. Dissecting neural odes. Advances in Neural Information Processing Systems, 33, pp.3952-3963. https://arxiv.org/abs/2002.08071

---

> > ### Comment · Reviewer_4ho2 · 2023-06-09
> > **RE: 1. Benefits of a Smaller Experiment**
> >
> > I think this table and the corresponding plot are illuminating. It makes it clear that the GQ method shines when taking gradients through the NN is expensive. This makes perfect sense since these are the computations that the GQ method minimizes.

---

> > > ### Author Response · Authors · 2023-06-10
> > > **RE: RE: 1. Benefits of a Smaller Experiment**
> > >
> > > Many thanks, we are glad you agree, we will include this as a plot in the final manuscript and inform you when this is uploaded.

---

> ### Author Response · Authors · 2023-06-06
> **2. Speed Accuracy Tradeoff**
>
> In the Appendix there is an experiment exploring the accuracy vs speed tradeoff. In the previous version of the manuscript it was in Appendix B.1 in the most recent version it is in Appendix C.1. The paragraph is titled "Effect of C" and is an ablation looking at the effect of the constant $C$ in the heuristic, which effectively chooses how many points to use in the quadrature, larger $C$ leads to more points. In Table 2 we investigate the effect of $C$ on the test accuracy and training time for the Nested Spheres task. For convenience we include this table below:
>
> | C                          | Time to train (/s) | Test Accuracy (%) |
> | ----------------------- | ----------------------|--------------------------|
> |$1 \times 10 ^{-3}$| $61.4 \pm 3.0 $ | $79.0 \pm 26.9$    |
> |$1 \times 10 ^{-1}$| $89.8 \pm 3.1 $ | $95.6 \pm 8.8$      |
> |$1 \times 10 ^{2}$| $141.0 \pm 2.4 $ | $95.8  \pm 8.4$    |
> |$1 \times 10 ^{5}$| $141.7  \pm 2.4$ | $95.8 \pm 8.4$    |
>
> We see that the time to train is lowest when C is lowest, because the fewest terms are used, however we also
> see that the final accuracy is compromised as a result. This is what we would expect, the number of terms used in the integration is not enough so the gradients are inaccurate. We see that using a very high C leads to a slightly higher accuracy than when C is 0.1. However, this is insignificant compared to the increase in training time, and the accuracies are easily within a standard deviation of each other. There is no increase in training time between C being $1 \times 10^2$ and $1 \times 10^5$ because we manually cap the number of terms in the quadrature at 64.

---

> > ### Comment · Reviewer_4ho2 · 2023-06-09
> > **RE: 2. Speed Accuracy Tradeoff**
> >
> > This table is really interesting. It would be nice to see it as a plot with more points to see where 'diminishing returns' kick in, though I understand if this isn't possible due to time constraints.
> >
> > I'm a little concerned by the comment that you cap the number of terms in the quadrature to 64. IIUC you are saying that this doesn't actually increase the number of quadrature points from C=10^2 to C=10^5, in which case the table might be a bit misleading.

---

> > > ### Author Response · Authors · 2023-06-10
> > > **RE: RE: 2. Speed Accuracy Tradeoff**
> > >
> > > Thank you for the fast reply, we're glad you find the table interesting. We appreciate what you say about the 64 points, and we can remove when $C=10^5$, we do have a statement that explains this though. The reason we included this was because this allows us to model the integrand as a 127 degree polynomial which is very high. We found that this limit was never reached in our experiments and that the levels of performance were still high.
> > >
> > > Please let us know if this is a significant problem and we shall remove that part of the table but as we say this is both in the text and the table description.

---

> ### Author Response · Authors · 2023-06-06
> **3. Clarity around Figure 1**
>
> We apologise that Figure 1 is not as clear as we intended. We hope to clarify this here. The aim of Figure 1 and the analytical system is to demonstrate our claims on a system that we can control, where we have full knowledge of the true loss and gradients. Those claims are that the GQ Adjoint will produce the same gradients as the Standard Adjoint only faster. So we are looking for the errors in the loss and gradients to match those given by the Standard Adjoint, not be lower than them.
>
> The system is one of exponential growth (proposed by Zhuang et al. 2021) with state $z$, the dynamics are given by $\frac{dz}{dt} = az$, this gives the solution $z(t) = z_0 \exp(at)$, where $a$ is a parameter determining the growth and $z_0$ is a parameter giving the initial condition. We integrate up to time $T$ and use the loss $L = z(T)^2$ which is $z_0^2 \exp(2aT)$. We use this system since we have exact analytical solutions and we can test the method when finding gradients with respect to:
>
> - Dynamics parameters: $a$
> - Initial conditions: $z_0$
> - Solution times: $T$
>
> Our method supports differentiation with respect to all of these to match the Direct and Standard Adjoint's capabilities. The gradients are given by $\frac{\partial L}{\partial a} = 2Tz_0^2\exp(2aT)$, $\frac{\partial L}{\partial z_0} = 2z_0 \exp(2aT)$ and $\frac{\partial L}{\partial T} = 2az_0^2 \exp(2aT)$.
>
> Figure 1 plots how the errors between predicted values and analytical values (given above) change as integration time increases. We actually plot the relative error $\bigg| \frac{\text{True} - \text{Predicted}}{\text{True}} \bigg|$, we do this because the state is exponentially increasing, so by showing the relative error we see the differences for small $T$ as well as large $T$.
>
> We see that the errors in the loss are the same for the Direct, Standard Adjoint and GQ Adjoint, this is what we expect since the loss only depends on the forward solve and they all use the same method for the forward solve. We expect the errors **in the gradients** between the Standard/GQ Adjoint methods and the Direct method to be different, this is because we are directly seeing the difference between the two approaches, adjoint being optimize then discretize, direct being discretize then optimize. This is indeed what we see. However, we do expect the Standard Adjoint and GQ Adjoint methods to produce the same gradients since they are calculating the same integral only in different ways (with the GQ method being faster). This is our claim, that we obtain the same gradients only faster, and this is indeed what we see in the three relative error in gradient plots. Therefore, it is a positive that our method is indistinguishable from the Standard Adjoint, it supports our claim.
>
> We have included two other methods MALI (Zhuang et al. 2021) and ACA (Zhuang et al. 2020) to additionally show how the results change if we use a different method for the forward solve. We will clarify all of this in the manuscript.
>
> # References
> 1. Zhuang, J., Dvornek, N.C., Tatikonda, S. and Duncan, J.S., 2021. Mali: A memory efficient and reverse accurate integrator for neural odes. In International Conference on Learning Representations 2021. https://arxiv.org/abs/2102.04668
>
> 2. Zhuang, J., Dvornek, N., Li, X., Tatikonda, S., Papademetris, X. and Duncan, J., 2020, November. Adaptive checkpoint adjoint method for gradient estimation in neural ode. In International Conference on Machine Learning (pp. 11639-11649). PMLR. https://arxiv.org/abs/2006.02493

---

> > ### Comment · Reviewer_4ho2 · 2023-06-09
> > **RE: 3. Clarity around Figure 1**
> >
> > Thank you for the clarification. It might help to incorporate some of this text into the manuscript.
> >
> > I still find the plots visually very difficult to interpret. In particular, the line for GQ is practically invisible in each plot. I would suggest experiment with color schemes and possibly larger plots. I think you could also apply some smoothing or down-sampling to these time series---it doesn't seem like there is any important high-frequency information here. You could also try plotting each method separately on smaller axes e.g. with sns.FacetGrid.

---

> > > ### Author Response · Authors · 2023-06-10
> > > **RE: RE: 3. Clarity around Figure 1**
> > >
> > > We agree, we shall absolutely incorporate this text into the manuscript and will update you when we have uploaded it.
> > >
> > > Regarding the visibility we are happy to make any requested changes, we have downsampled the time-series, we have increased the size of the figure in the paper and we have swapped the linestyles between GQ and MALI. We believe the reason it is hard to make out the GQ method is because it essentially overlays the Standard Adjoint, which is supporting our claims that they produce the same results.

---

> ### Author Response · Authors · 2023-06-06
> **4. Quadrature Discussion**
>
> We agree that our discussion around integration methods is not precise enough and will update the manuscript accordingly to include the information below.
>
> We dissect three integration techniques where we solve a separate integral for each parameter, giving us many one dimensional integrals in parallel rather than one high dimensional integral. Assume we wish to solve the integral given by $\int_a^bf(x)dx$.
>
> # Newton-Cotes
> Newton-Cotes integration techniques split the interval $[a, b]$ into $N$ **equidistant** intervals. Then a polynomial of a given degree is fit on each interval which can be integrated. Classic examples are:
>
> - Trapezoid rule, this fits a linear interpolant to each interval. The error is $\mathcal{O}\big(\frac{f^{2}(\xi)}{N^2}\big)$, where $f^{m}(\xi)$ refers to the largest $m$-th derivative on the interval $[a, b]$ located at $\xi$.
> - Simpson's rule, this fits quadratic interpolants on the intervals. The error is $\mathcal{O}\big(\frac{f^{4}(\xi)}{N^4}\big)$.
> - Boole's rule, this fits quartic interpolants on the intervals. The error is $\mathcal{O}\big(\frac{f^{6}(\xi)}{N^6}\big)$.
>
> In the above, $f^m(\xi)$ is a constant, it has been included in the error complexity to show that if the true function is exactly a polynomial of a given degree the error can be zero. For example, if the function is exactly linear the second derivative everywhere is zero, so the Trapezoid and other Newton-Cotes rules give the exact results. However the crucial point is that these are still constant with respect to $N$ so the errors for the Trapezoid, Simpson and Boole's rules are $\mathcal{O}\big( \frac{1}{N^2} \big)$, $\mathcal{O}\big( \frac{1}{N^4} \big)$ and $\mathcal{O}\big( \frac{1}{N^6} \big)$. Therefore for Newton-Cotes the error is $\mathcal{O}\big( \frac{1}{N^k} \big)$ where $k$ depends on the rule for interpolation.
>
> # Gaussian Quadrature
> Gaussian  Quadrature **does not use equidistant points**, the points are given by principled locations on the interval, which allow us to approximate a $2N-1$ degree polynomial with only $N$ points in the quadrature sum. The error using Gaussian Quadrature with weight function $1$ is bounded by $\frac{(b-a)^{2N+1}(N!)^4}{(2N+1)((2N)!)^3}f^{2N}(\xi)$. There are two key terms in this error, the first is $f^{2N}(\xi)$, as we use more points we model a higher degree polynomial unlike Newton-Cotes which stays constant. As mentioned we are therefore already able to exactly solve a polynomial of degree $2N-1$, so if the function is well approximated by such a polynomial we will have an accurate approximation of the integral. The other key term is $((2N)!)^3$ in the denominator, which dominates the other terms in the error bound showing the error will shrink very quickly with $N$, significantly faster than if the error is $\mathcal{O}\big( \frac{1}{N^k} \big)$.
>
> Please note that the above assumes that $f$ is differentiable, if there are discontinuities in $f$ the integral is split up to make up for that. We elaborate on this in Appendix A where we describe how we adapt the GQ method to time-series.
>
> # Monte Carlo Integration
> Monte Carlo methods approximate the integral by sampling many points and calculating the mean of these. $\int_a^bf(x)dx = \int_a^bp(x)\frac{f(x)}{p(x)}dx \approx \frac{1}{N}\sum_{i=1}^{N}\frac{f(x_i)}{p(x_i)} \quad x_i \sim p(x)$. By sampling the points they are **not equidistant but also not in principled locations**. The distribution of these calculations follows a normal distribution with the mean being the true integral and the variance is $\mathcal{O}\big( \frac{1}{N} \big)$, so the error using Monte Carlo is $\mathcal{O}\big( \frac{1}{\sqrt{N}} \big)$. Monte Carlo has the worst error rates of the described methods which is why it is only used for high dimensional integrals where it is not possible to apply the other methods.
>
> Therefore we use Gaussian Quadrature for these integrals since it has the best error rate. This information can be found in Chapter 3 of Introduction to Numerical Analysis (Stoer and Bulirsch 1993).
>
> # References
> 1. Stoer, J., Bulirsch, R., Stoer, J. and Bulirsch, R., 1993. Topics in Integration. Introduction to Numerical Analysis, pp.125-166.

---

> > ### Comment · Reviewer_4ho2 · 2023-06-09
> > **RE: 4. Quadrature Discussion**
> >
> > Thanks for the detailed discussion. The part that I think would benefit the paper is around the convergence rate of GL quadrature. It would be particularly nice to connect this convergence rate to your heuristic for choosing the number of quadrature points (though, again, I understand if time doesn't allow it).
> >
> > Is there any evidence that the error converges like this bound as you adjust the parameter C?

---

> > > ### Author Response · Authors · 2023-06-10
> > > **RE: RE: 4. Quadrature Discussion**
> > >
> > > Ultimately this is an upper bound on the error, it is stated in Stoer and Bulirsch that the true error may be far less than this bound. As we adjust the parameter $C$ the number of points used is approximately linear in $C$, it is actually $\lceil C m \rceil$, where $m$ represents the other terms in the heurstic. So we replace $N$ in the error bound with $\lceil Cm \rceil$, so roughly the error converges in the same way as $C$ is increased, however the ceiling function compicates any exact analysis.

---

> ### Author Response · Authors · 2023-06-06
> **5. Number of $\frac{da_\theta}{dt}$ evaluations**
>
> The Standard Adjoint method produces approximately half the number of function evaluations during the backward solve as were performed during the forward solve. This is shown in Figure 3 of Chen et al. 2018 (the original Neural ODE paper). Our method then uses the number of evaluation points introduced by our heuristic $\big \lceil C \frac{\text{NFE}(t_k-t_{k-1})}{(t_K - t_0)}\big \rceil$. In our case we use $C=0.1$ giving us $\big \lceil \frac{\text{NFE}}{10}\frac{(t_k-t_{k-1})}{(t_K - t_0)}\big \rceil$, and if we assume we aren't working with a time series, then this simplifies to $\big \lceil \frac{\text{NFE}}{10}\big \rceil$. And so we save approximately $\frac{\text{NFE}}{2} - \frac{\text{NFE}}{10} = \frac{2 \text{NFE}}{5}$ fewer evaluations during the backward solve, where NFE is the number of function evaluations during the forward solve, which accounts for the complexity of the differential equation being solved. The Standard Adjoint method uses $\frac{\text{NFE}}{2}$ function evaluations and the GQ Adjoint uses $\frac{\text{NFE}}{10}$, so in relative terms the GQ method uses $\frac{1}{5}$ the number of evaluations as the Standard Adjoint.
>
> Note that this is infact likely an overestimate. In Kidger et al. 2020 it is demonstrated that if treating $a_\theta$ as an integral rather than differential equation far fewer function evaluations during the backward solve can be used. This paper only uses the error estimates of $[z, a_z]$ during the backward solve using an adaptive solver, the error estimate of $a_\theta$ is not included. Similarly since we are only solving the ODE for $[z, a_z]$ the error estimate in our case only comes from there as well. In Kidger et al. 2020 Figure 1 and Table 1 it is shown that the number of backward function evaluations is significantly fewer in this scenario (40%–62% fewer steps as quoted), and so we can expect even fewer evaluations.
>
> By using a range of 40%–60% fewer evaluations given by Kidger et al. 2020 we can put all this together. If we use NFE function evaluations during the forward solve:
>
> - The Standard Adjoint uses $0.5\text{NFE}$ evaluations of $[f(z, t), a_z^T{\frac{\partial f}{\partial z}}, a_z^T{\frac{\partial f}{\partial \theta}}]$
> - The Seminorm Adjoint uses between $0.2 \text{NFE}$ and $0.3\text{NFE}$ evaluations of $[f(z, t), a_z^T{\frac{\partial f}{\partial z}}, a_z^T{\frac{\partial f}{\partial \theta}}]$
> - The GQ adjoint uses between $0.2 \text{NFE}$ and $0.3\text{NFE}$ evaluations of $[f(z, t), a_z^T{\frac{\partial f}{\partial z}}]$ and $0.1\text{NFE}$ evaluations of $a_z^T{\frac{\partial f}{\partial \theta}}$
>
> And so we can see how the GQ method uses far fewer evaluations compared to both the Standard and Seminorm Adjoints. We shall include these calculations in the manuscript.
>
> # References
>
> 1. Chen, R.T., Rubanova, Y., Bettencourt, J. and Duvenaud, D.K., 2018. Neural ordinary differential equations. Advances in neural information processing systems, 31. https://arxiv.org/abs/1806.07366
>
> 2. Kidger, P., Chen, R.T. and Lyons, T.J., 2021. " Hey, that's not an ODE": Faster ODE Adjoints via Seminorms. In ICML (pp. 5443-5452). https://arxiv.org/abs/2009.09457

---

> > ### Comment · Reviewer_4ho2 · 2023-06-09
> >
> > This is a nice analysis---I think including it will improve the manuscript.

---

> > > ### Author Response · Authors · 2023-06-10
> > > **RE: RE: 5. Number of $\frac{da_{\theta}}{dt}$ Evaluations**
> > >
> > > We're glad you agree, we shall include this in the updated manuscript and inform you when it is uploaded.

---

> ### Author Response · Authors · 2023-06-06
> **6. Peak memory usage in Figure 2**
>
> Regarding memory, the key claim is that our method uses constant memory with respect to integration time or number of evaluations. This is due to using a rolling sum, for the quadrature. We see this in Figure 2 supporting our claim that the GQ Adjoint is memory efficient with the same memory complexity as the Standard Adjoint - constant, whereas the Direct and ACA methods which directly backpropagate through solver operations have non-constant memory usage.
>
> Nevertheless it is correct that Figure 2 shows a slight increase in the memory overhead used by the GQ Adjoint relative to the Standard Adjoint. In theory we should use the same amount of memory. During the backward solve the Standard Adjoint method has the states $[z, a_z, a_\theta]$ as well as velocities $[f(z, t), a_z^T\frac{\partial f}{\partial z}, a_z^T\frac{\partial f}{\partial \theta}]$. The GQ method is similar except we replace $a_\theta$ with the running sum of the quadrature calculation. So we expect the memory usage to be the same. It is likely that because the Standard Adjoint calculates all of the velocities at the same point, it only uses one forward evaluation of $f$ while tracking gradients to do this. Whereas the GQ method does not do this, it calculates $ a_z^T\frac{\partial f}{\partial \theta} $ at the specified points in the quadrature solve, which are not used to calculate $[f(z, t), a_z^T\frac{\partial f}{\partial z}]$ and so it uses one additional forward evaluation of $f$ while tracking gradients and so uses slightly more memory. As stated the main takeaway of Figure 2 is that the memory usage is constant, supporting our claim.

---

> ### Author Response · Authors · 2023-06-06
> **7.  Deviation from Adjoint Method as Error**
>
> The GQ Adjoint and Standard Adjoint calculate the same integral only in different ways. Our method uses Gaussian Quadrature and the Standard method treats it as a differential equation. Therefore we disagree that we can't expect the GQ method to be more accurate than the adjoint, because the Standard Adjoint evaluates $\frac{d a_\theta}{dt}$ assuming we are solving an ODE whereas we are actually solving an integral. We would expect that if we used as many points in the GQ calculation as the Standard calculation the GQ method would be more accurate. In order to overcome the difficulty of not knowing the true gradient we first investigate the analytical system given in Figure 1. As described in our other response we see that the GQ and Standard Adjoint produce the same gradients for this system supporting the claim that they produce the same gradients. Further to this we plot the performances of the different training methods for each experiment to show they are the same within error, which does not guarantee but is further evidence to suggest that the two methods produce the same gradient. Finally in the Appendix we have plotted the test performance against epoch during training given by Figures 8, 9, 10, 11, 12 and 14 to show that these curves match, further supporting that the calculated gradients match. Additionally in these plots we plot the test performance against wall-clock time to show that the GQ method trains in faster wall-clock time.
>
> For this reason we do not believe this change is necessary as we already demonstrate in multiple cases the gradients match.

---

### Review · Reviewer_rgHZ · 2023-06-07

**Summary Of Contributions:**

This paper proposes an acceleration technique for computing neural ODE adjoints (backpropagation). The key insight is that the backward pass integral can be approximated using quadrature, and the quadrature points are then used to guide the backward solver. The idea is principled, clever, and novel.

**Audience:**

Yes

**Broader Impact Concerns:**

no issues

**Claims And Evidence:**

Yes

**Requested Changes:**

Running comments

- The introduction contrast the ‘direct’ backprop with adjoint. The direct method is said to be “fast” but “memory-intensive”. I find both statements strange: this is just standard neural network backprop over many layers. I don’t really see in what sense do we characterise this as fast or memory intensive. Similarly saying that adjoint is “slow” is also strange: it solves a different problem and one can’t really compare Neural-ODE-adjoint with ResNet-Euler-backprop. Finally, there is also the issue that adjoint method “O(1)” method is biased due to numerical reversibility issues of the neural ODE. The statements need more precision.
- “arbitrarily large computational graphs”. Not sure I buy this. An ODE takes some amount of time increments, and each takes some linear number of intermediate steps (eg. 4 in RK4). The backpropagation recycles the gradients backwards, so I don’t see why the computational graphs would explode. I also don’t see why we need to store all intermediate states (in practise we probably do). The paper needs to be it more precise what is the complexity of standard backprop wrt number of increments and order of solver.
- “faster and potentially more accurately”. I don’t see how integral approximations can be more accurate than the true ODE. Surely you are incurring some error.
- Since w(t)=1, why do we include it at all?
- “adding each term in the sum..”. I don’t understand this part. This is all vague and I have no idea what terms or sums are we talking about.
- I don’t understand fig 1. What is integration time? It doesn’t make much sense that we integrate for 20..28 seconds: surely the t_K is constant? I don’t understand y-axis either: relative wrt what? It seems that ACA is the gold standard: how does ACA then incur error? This entire experiment is poorly described, and I can’t follow.
- In the experiments it’s unclear how many NFEs the backward pass takes between adjoint and GQ. Can you describe this?

**Strengths And Weaknesses:**

This trick improves adjoint efficiency quite significantly with clear empirical demonstrations. However, the adjoint is not a practical method in the first place, and naive autodiff is still better in practise.

The paper is well-written.

The claims are accurate, and are backed by sufficient evidence. This paper is interesting to the neural ODE community.

---

> ### Author Response · Authors · 2023-06-07
> **Official Response to Reviewer rgHZ**
>
> Firstly, many thanks for taking the time and effort to review our paper. We are pleased that you found the idea principled, clever and novel; we are also pleased you found the paper well-written, of interest to the Neural ODE community and supportive of our claims. We are grateful for the detailed comments summarised below:
>
> 1. Direct Backprop vs Adjoint Method
> 2. Imprecise Statements about the Computational Graph
> 3. Accuracy of an Integral vs ODE Approach
> 4. The weight function in quadrature sum
> 5. Imprecise statements about running sums
> 6. Figure 1
> 7. Number of Function Evaluations in the forward and backward passes
>
> We happily answer all of these concerns in separate replies below so that discussion around each point is independent. We have also included all of these points in an updated manuscript and shall inform all reviewers when this is uploaded.

---

> > ### Author Response · Authors · 2023-06-07
> > **3. Accuracy of an Integral vs ODE Approach**
> >
> > It is not entirely accurate to say that we are using an integral to approximate the true ODE. Both methods are being used to approximate the gradient of the forward differential equation solve - the optimize then discretize approach. The analytical gradient of this forward solve is given by the integral in Equations 7 and 11, which can then be approximated using ODE methods (the Standard Adjoint) or with Quadrature methods as we propose.
> >
> > The adjoint method for Neural ODEs is actually a special case of optimal control theory. In optimal control theory we have a dynamical system given by $\frac{dz}{dt} = f(z, t, \theta(t))$ where $\theta(t)$ is a control that we choose, for example the thrust of an airplane $\theta(t)$ controls the height $z(t)$ according to a differential equation given by Newton's Laws. Given some loss at the final time we solve the adjoining system $\lambda(t_K) = \frac{\partial L}{\partial z(t_K)}$ and $\frac{d \lambda}{dt} = -\lambda^T \frac{\partial f}{\partial z}$. The method of steepest descent is used to update the control so that $\theta(t)$ is updated to $\theta(t) - \eta \lambda(t)^T\frac{\partial f}{\partial \theta}(t)$, for small step size $\eta$. In the case of Neural ODEs the control is given by the parameters which are a constant, and so we sum all of the local changes to construct an integral, $\theta$ becomes $\theta - \eta \int_{t_0}^{t_K}\lambda^T(t)\frac{\partial f}{\partial \theta}(t) dt$.
> >
> > The point of this is to show that ultimately the gradients in the optimize then discretize approach are really an integral, it's just that previously this was solved as an ODE. So ultimately using quadrature - a method more suited to definite integrals than ODE methods - might produce a more accurate result for the integral than using the ODE method given by the Standard Adjoint, and the result given by the Standard Adjoint should not be considered the true gradient. Ultimately we understand that this statement could be misleading, implying that the Standard Adjoint is inaccurate, and will change it to "faster and to the same level of accuracy".

---

> ### Author Response · Authors · 2023-06-07
> **1. Direct Backprop vs Adjoint Method**
>
> Regarding the point "However, the adjoint is not a practical method in the first place, and naive autodiff is still better in practise." We definitely agree that naive autodiff (Direct Backpropagation) is used more in practice, mostly due to its speed in comparison to the adjoint method but also as mentioned the bias that can arise from numerical reversibility. However, that does not mean that the adjoint method is never used, and in particular it has to be used when the Direct method uses too much memory, for example:
>
> - Zhuang et al 2021: "Due to the heavy memory burden caused by large images, the naive method and ACA are unable to train a Neural ODE on ImageNet with 4 GPUs; only MALI and the adjoint method are feasible due to the constant memory".
> -  Morrill et al. 2021: "The lengths are sufficiently long that adjoint-based backpropagation was often needed simply to avoid running out of memory at any reasonable batch size".
>
> So it is used when the time-series is very long or the state size is large. As well as this we note that in Kidger 2022 that while the disadvantages of the adjoint method are noted "continuous adjoint methods often (but not always) still work in practice, without needing any special care". So the Adjoint method is not necessarily impractical, it is just that currently the Direct is the preferred method in practice. Whilst acknowledging this we still believe there is merit in studying the Adjoint method, and there are plenty of examples of works that also study it such as Zhuang et al. 2021, Kidger et al. 2020 and Daulbaev et al. 2020. As well as this we still believe it will be of interest to the TMLR community, taking a quote from the homepage: "we facilitate scientific discourse on topics that may not yet be accepted in mainstream venues but may be important in the future" https://jmlr.org/tmlr/index.html, and we also believe we agree with each other here since the review states "This paper is interesting to the neural ODE community".
>
>
> Next we consider the main point regarding comparisons in the introduction. We apologize that this was unclear, we did not mean that the Direct method is generally fast/memory intensive, or that the Standard Adjoint is generally slow/memory efficient, our intended meaning was that the Direct method is **faster** that the Standard Adjoint  but also **more memory intensive** than the Standard Adjoint, we were implicitly making the comparison but will make that change to be explicit. This is indeed supported by the literature, for example the beginning of Chapter 5 of Kidger 2022.
>
> To make the statements more precise, consider a solve with state size $N_z$, dynamics function with $N_f$ hidden layers, number of function evaluations in the forward pass $N_t$ and number of evaluations in the backward pass $N_r$. Then the memory usage for the Direct method is $\mathcal{O}(N_zN_fN_t)$ whereas for the Standard Adjoint method it is $\mathcal{O}(N_zN_f)$, which is why it is often referred to as constant with respect to the integration time, since it does not depend on $N_t$. This is because the Direct method has to store every activation every time the dynamics function is evaluated, whereas the Adjoint method only has to store the activations from one function evaluation. The computational complexity is then $\mathcal{O}(2N_zN_fN_t)$ for the Direct method and $\mathcal{O}(N_zN_f(N_t+N_r))$ for the Adjoint method, but the state in the backward solve is far larger increasing the computation. These values are taken from Table 1 of Zhuang et al. 2021. We shall include this information in the manuscript.
>
> This brings us to the final point, the comparison. The review states that the Direct and Adjoint methods cannot be compared since they do different things. However we argue that they can and should be compared, since they are used for the same purpose, that is to find the gradient of a Neural ODE, we should compare them to determine which is best for the use-case, since they have different merits due to solving the problem in different ways. This is analogous to comparing forward mode and reverse mode autodiff, they solve the same problem to differentiate but in different ways; by making the comparison it was determined that reverse mode is typically better and so we use backpropagation.

---

> > ### Author Response · Authors · 2023-06-07
> > **References for Response 1**
> >
> > Below are the references for this response:
> >
> > 1. Zhuang, J., Dvornek, N.C., Tatikonda, S. and Duncan, J.S., 2021. Mali: A memory efficient and reverse accurate integrator for neural odes. In International Conference on Learning Representations 2021. https://arxiv.org/abs/2102.04668
> >
> > 2. Morrill, J., Salvi, C., Kidger, P. and Foster, J., 2021, July. Neural rough differential equations for long time series. In International Conference on Machine Learning (pp. 7829-7838). PMLR. https://arxiv.org/abs/2009.08295
> >
> > 3. Kidger, P., 2022. On neural differential equations. https://arxiv.org/abs/2009.08295
> >
> > 4. Kidger, P., Chen, R.T. and Lyons, T.J., 2021. " Hey, that's not an ODE": Faster ODE Adjoints via Seminorms. In ICML (pp. 5443-5452). https://arxiv.org/abs/2009.09457
> >
> > 5. Daulbaev, T., Katrutsa, A., Markeeva, L., Gusak, J., Cichocki, A. and Oseledets, I., 2020. Interpolation technique to speed up gradients propagation in neural odes. Advances in Neural Information Processing Systems, 33, pp.16689-16700. https://arxiv.org/abs/2003.05271

---

> > ### Comment · Reviewer_rgHZ · 2023-06-07
> > **response**
> >
> > Thanks for the clarifications. The arguments of adjoint practicality are convincing, I agree. I would expect that the memory requirements for adjoint to be based on $N_z + N_z + N_f N_z$ to reflect memory for state, adjoint, and differential jacobian. In your response you instead have a product, and I wonder which way this goes.
> >
> > I think one perspective that is missing from the submission is that adjoint and direct method give a different result. When we solve an ODE forward we obtain a piecewise linear approximation of the true state over time. The direct method will return the **gradient of this ODE approximation**. In adjoint we again obtain an approximation of the forward solution, but however, the backward solution can be (and actually almost surely will be) a gradient of a *different* forward approximation (if we believe ANODE). I see the adjoint giving an **approximation of the ODE gradient**. There are then subtle differences between direct and adjoint, and one should be aware of these nuances. I agree that they are used for the same purpose, and comparing them is of course fine.

---

> > > ### Author Response · Authors · 2023-06-08
> > > **Many thanks for the fast response, we reply below**
> > >
> > > Many thanks for the fast response, we are pleased you are convinced with the arguments around Adjoint practicality. And we happily answer the remaining two points below:
> > >
> > > # Memory Usage
> > > You are indeed correct, the memory usage would include $N_z$ for the state, $N_z$ for $a_z$ and $N_f N_z$ to calculate the derivatives, we would further need $N_\theta$ for $a_\theta$ (the number of parameters). We wrote $N_fN_z$ as the dominant term in the scaling. These are not the exact memory usages since then we have to include various other quantities such as optimizer state, the parameters, the internal state of the ODE solver etc., so we've only included the dominant quantity. We shall make this clear when we update the manuscript.
> > >
> > > # Adjoint vs Direct
> > > We absolutely agree, the Direct and Adjoint do give different results, certainly for the gradients. Depending on the task and the parameter loss landscape this may not result in a difference in the final performance, the models may settle in different but equally good local optima. Or the loss landscape might be sufficiently shaped such that even with different gradients they still reach the same minimum, for example if the loss landscape is convex (this is very rarely the case). But as you say the gradients are different so we'll focus on that. We do not explicitly mention this in our manuscript but it is shown in Figure 1, we see that the errors in the loss are the same for the Adjoint and Direct method, but the errors in gradient are different showing they produce different gradients for the same loss. Interestingly for this experiment the errors are larger for the Direct than the Adjoint method, this is in contrast to ANODE. We hypothesize this is because in ANODE the dynamics are learnt so that images converge on a final value to be turned into a distribution over classes, whereas in our exponential experiment the dynamics diverge, therefore for practical purposes where we do not want diverging dynamics it is likely that the Direct method will produce preferable gradients.
> > >
> > > As you mention it is interesting to see how the different approaches differ, since the Direct method takes an exact gradient of an approximation to the forward ODE solve, whereas the adjoint method is mathematically the exact gradient of the ODE solve but this has to then be approximated, and in this case there are multiple levels of the approximation since we approximately solve the forward and reverse time ODEs. Intriguingly, a new paper has just been uploaded by Xu et al. 2023 which shows using the leapfrog ODE solver can produce incorrect gradients when directly backpropagating which oscillate around the true ODE gradient.
> > >
> > > We do wish to stress that ultimately our paper is about improving the Adjoint as we say in our introduction "Our focus in this work is building on the adjoint method.", however we do agree that one should be aware of the nuances and care must be taken. So to account for this we shall explicitly write that the gradients produced by the Direct and Adjoint methods tend to be different.
> > >
> > > # References
> > > 1. Xu, Y., Chen, S., Li, Q. and Wright, S.J., 2023. Correcting auto-differentiation in neural-ODE training. https://arxiv.org/abs/2306.02192

---

> > > > ### Comment · Reviewer_rgHZ · 2023-06-09
> > > > **thanks**
> > > >
> > > > Thanks for the responses. I have no more issues with the paper.

---

> > > > > ### Author Response · Authors · 2023-06-10
> > > > > **Thank You for the Review we Shall Update the Paper Accordingly**
> > > > >
> > > > > Many thanks for your response and again for the time to review the paper. We shall inform you and all reviewers when we update the manuscript.

---

> ### Author Response · Authors · 2023-06-07
> **2. Imprecise Statements about the Computational Graph**
>
> This is our mistake, thank you for observing it and we shall correct it. We incorrectly used computational graph where we meant memory. Our intention was to say that each time the function is evaluated during the forward solve this is added to the graph (to correctly backpropagate) and so the memory requirements can grow arbitrarily large. However, we need to stress that this is for adaptive step ODE solvers, that adjust the size of the step based on the complexity of the dynamics being solved. You are right in the case that the ODE solver uses a fixed number of steps such as an Euler solver or RK4 then the memory requirements and graph would not grow. However, since the adaptive solver may take arbitrarily small steps for arbitrarily complicated dynamics, the memory use can grow significantly as shown by the $\mathcal{O}(N_zN_fN_t)$ memory requirements for the Direct versus the $\mathcal{O}(N_zN_f)$ requirements for the Adjoint method.

---

> ### Author Response · Authors · 2023-06-07
> **4. The Weight Function in Quadrature Sum**
>
> The integral $\int_a^bf(t)dt$ is equal to $\int_a^bw(t) \frac{f(t)}{w(t)}dt$, normally written as $\int_a^b w(t)\tilde{f}(t)dt$, where $w(t)$ is a positive weight function. The weight function is included if the function $f(t)$ we are integrating has a particular form so that $\frac{f(t)}{w(t)}$ is considered simpler than $f(t)$. There are various Gaussian Quadratures, each with different weight functions, as well as integration intervals, and corresponding $w_i$ and $\tau_i$ in the quadrature sum. For example, Gauß-Chebyshev Quadrature of the second kind uses $w(t) = \sqrt{1-t^2}$. To see when this is useful consider solving the integral $\int_a^b \sqrt{1-t^2}f(t)dt$. We could solve this by solving the problem $\int_a^bw(t)\sqrt{1-t^2}f(t)dt$, using $w(t) = 1$ i.e. Gauß-Legendre Quadrature; or we could solve the problem $\int_a^bw(t)f(t)dt$ where $w(t) = \sqrt{1-t^2}$ using Gauß-Chebyshev Quadrature of the second kind.
>
> We include the weight function in our description so that we can explain the reasons that we **cannot** use the other Gaussian Quadratures in the paragraph titled "Why Gauß-Legendre Quadrature?". Since our dynamics function is given by a neural network, there is no obvious way to simplify using a weight function so we use $w(t)=1$, which (among with other reasons) means we have to use Gauß-Legendre quadrature over Gauß-Chebyshev Quadrature, Gauß-Jacobi Quadrature, Gauß-Hermite Quadrature and so on.

---

> ### Author Response · Authors · 2023-06-07
> **5. Imprecise Statements about Running Sums**
>
> We apologise that this is not clear, we do include both the running sum and terms in Algorithms 1 and 2 where we do precisely show their usage. The paragraph you refer to was used to intuitively describe how the memory consumption is reduced. We shall be more precise and incorporate the description below into that paragraph. Note that in the below $|\theta|$ refers to number of trainable parameters and $f(t)$ is used in place of $a_z^T\frac{\partial f}{\partial \theta}$.
>
> The quadrature sum is given by $\sum_i w_i f(\tau_i)$. In the case of Gauß-Kronrod quadrature (Rackauckas et al. 2021), each term $w_if(\tau_i)$ in the sum is stored during the backward solve and the sum is taken at the end. In fact many points along the trajectories of $z(t)$ and $a_z(t)$ are stored so that the quadrature can be carried out after the solve, using even more memory to store these trajectories. The use of this is that more terms can be introduced to the quadrature sum until a sufficient accuracy is reached (which is determined when there is negligible change to the result when more points are introduced in the sum).
>
> As mentioned this is highly memory intensive, so our approach is to determine the number of points we'll need before we start the backward solve, and then to add each term to a running sum during the backward solve as we go. That is at the start of the backward solve initalize a vector of zeros $g=\mathbf{0}$, and for each $\tau_i$ update $g$ to be $g + w_i f(\tau_i)$, so that we do not store any more than two vectors of size $|\theta|$ at any time, in contrast to the aforementioned method which stores many vectors of size $|\theta|$.
>
> # References
>
> 1. Rackauckas, C., Ma, Y., Martensen, J., Warner, C., Zubov, K., Supekar, R., Skinner, D., Ramadhan, A. and Edelman, A., 2020. Universal differential equations for scientific machine learning. https://arxiv.org/abs/2001.04385

---

> ### Author Response · Authors · 2023-06-07
> **6. Figure 1**
>
> We apologise that Figure 1 is not as clear as we intended. We hope to clarify this here. The aim of Figure 1 and the analytical system is to demonstrate our claims on a system that we can control, where we have full knowledge of the true loss and gradients. Those claims are that the GQ Adjoint will produce the same gradients as the Standard Adjoint only faster. So we are looking for the errors in the loss and gradients to match those given by the Standard Adjoint, not be lower than them.
>
> The system is one of exponential growth (proposed by Zhuang et al. 2021) with state $z$, the dynamics are given by $\frac{dz}{dt} = az$, this gives the solution $z(t) = z_0 \exp(at)$, where $a$ is a parameter determining the growth and $z_0$ is a parameter giving the initial condition. We integrate up to time $T$ and use the loss $L = z(T)^2$ which is $z_0^2 \exp(2aT)$. We use this system since we have exact analytical solutions and we can test the method when finding gradients with respect to:
>
> - Dynamics parameters: $a$
> - Initial conditions: $z_0$
> - Solution times: $T$
>
> Our method supports differentiation with respect to all of these to match the Direct and Standard Adjoint's capabilities. The gradients are given by $\frac{\partial L}{\partial a} = 2Tz_0^2\exp(2aT)$, $\frac{\partial L}{\partial z_0} = 2z_0 \exp(2aT)$ and $\frac{\partial L}{\partial T} = 2az_0^2 \exp(2aT)$.
>
> Figure 1 plots how the errors between predicted values and analytical values (given above) change as integration time $T$ increases. We actually plot the relative error $\bigg| \frac{\text{True} - \text{Predicted}}{\text{True}} \bigg|$, we do this because the state is exponentially increasing, so by showing the relative error we see the differences for small $T$ as well as large $T$.
>
> We see that the errors in the loss are the same for the Direct, Standard Adjoint and GQ Adjoint, this is what we expect since the loss only depends on the forward solve and they all use the same method for the forward solve. We expect the errors **in the gradients** between the Standard/GQ Adjoint methods and the Direct method to be different, this is because we are directly seeing the difference between the two approaches, adjoint being optimize then discretize, direct being discretize then optimize. This is indeed what we see. However, we do expect the Standard Adjoint and GQ Adjoint methods to produce the same gradients since they are calculating the same integral only in different ways (with the GQ method being faster). This is our claim, that we obtain the same gradients only faster, and this is indeed what we see in the three relative error in gradient plots.
>
> We have included two other methods MALI (Zhuang et al. 2021) and ACA (Zhuang et al. 2020). Their results are not particularly important, the main results that matter are those given by the Direct, Standard adjoint and GQ Adjoint, we included the additional results to show how the errors in the relative gradient change if we use a different method for the forward solve. We will clarify all of this in the manuscript.
>
> # References
> 1. Zhuang, J., Dvornek, N.C., Tatikonda, S. and Duncan, J.S., 2021. Mali: A memory efficient and reverse accurate integrator for neural odes. In International Conference on Learning Representations 2021. https://arxiv.org/abs/2102.04668
>
> 2. Zhuang, J., Dvornek, N., Li, X., Tatikonda, S., Papademetris, X. and Duncan, J., 2020, November. Adaptive checkpoint adjoint method for gradient estimation in neural ode. In International Conference on Machine Learning (pp. 11639-11649). PMLR. https://arxiv.org/abs/2006.02493

---

> ### Author Response · Authors · 2023-06-07
> **7. Number of Function Evaluations in the Forward and Backward Passes**
>
> Unlike in other methods for speeding up the adjoint method, the number of function evaluations is not indicative of wall-clock time for the GQ method. This is because in previous methods during the backward solve one function evaluation is used each time to calculate the velocity of $[z, a_z, a_\theta]$ which is $[f(z, t), -a_z^T\frac{\partial f}{\partial z}, -a_z^T\frac{\partial f}{\partial \theta}]$. And the gradient is still found as an ODE. However in our method we use one function evaluation to calculate the velocity of  $[z, a_z]$ which is $[f(z, t), -a_z^T\frac{\partial f}{\partial z}]$ and another function evaluation to calculate the term in the quadrature given by $a_z\frac{\partial f}{\partial \theta}$. So number of function evaluations is not indiciative of time to obtain the gradient. The reason we are faster is because the number of function evaluations does not take into account the time to backpropagate to calculate $a_z^T\frac{\partial f}{\partial z}$, or $a_z^T\frac{\partial f}{\partial \theta}$, as well generally solving a differential equation with state size $2|z| + |\theta|$ (accounting for $[z, a_z, a_\theta]$) versus one with size $2|z|$ (accounting for $[z, a_z]$). Nevertheless we can make approximate calculations for how many function evaluations are calculated:
>
> The Standard Adjoint method produces approximately half the number of function evaluations during the backward solve as were performed during the forward solve. This is shown in Figure 3 of Chen et al. 2018 (the original Neural ODE paper). Our method then uses the number of evaluation points introduced by our heuristic $\big \lceil C \frac{\text{NFE}(t_k-t_{k-1})}{(t_K - t_0)}\big \rceil$. In our case we use $C=0.1$ giving us $\big \lceil \frac{\text{NFE}}{10}\frac{(t_k-t_{k-1})}{(t_K - t_0)}\big \rceil$, and if we assume we aren't working with a time series, then this simplifies to $\big \lceil \frac{\text{NFE}}{10}\big \rceil$. And so the Standard Adjoint method uses $\frac{\text{NFE}}{2}$ function evaluations and the GQ Adjoint uses approximately $\frac{\text{NFE}}{10}$, so in relative terms the GQ method uses $\frac{1}{5}$ the number of evaluations as the Standard Adjoint.
>
> Note that this is in fact likely an overestimate. In Kidger et al. 2020 it is demonstrated that if treating $a_\theta$ as an integral rather than differential equation far fewer function evaluations during the backward solve can be used. This paper only uses the error estimates of $[z, a_z]$ during the backward solve using an adaptive solver, the error estimate of $a_\theta$ is not included. Similarly since we are only solving the ODE for $[z, a_z]$ the error estimate in our case only comes from there as well. In Kidger et al. 2020 Figure 1 and Table 1 it is shown that the number of backward function evaluations is significantly fewer in this scenario (40%–62% fewer steps as quoted), and so we can expect even fewer evaluations.
>
> By using a range of 40%–60% fewer evaluations given by Kidger et al. 2020 we can put all this together. If we use NFE function evaluations during the forward solve:
>
> - The Standard Adjoint uses $0.5\text{NFE}$ evaluations of $[f(z, t), a_z^T{\frac{\partial f}{\partial z}}, a_z^T{\frac{\partial f}{\partial \theta}}]$
> - The Seminorm Adjoint uses between $0.2 \text{NFE}$ and $0.3\text{NFE}$ evaluations of $[f(z, t), a_z^T{\frac{\partial f}{\partial z}}, a_z^T{\frac{\partial f}{\partial \theta}}]$
> - The GQ adjoint uses between $0.2 \text{NFE}$ and $0.3\text{NFE}$ evaluations of $[f(z, t), a_z^T{\frac{\partial f}{\partial z}}]$ and $0.1\text{NFE}$ evaluations of $a_z^T{\frac{\partial f}{\partial \theta}}$
>
> And so we can see how the GQ method uses fewer evaluations compared to both the Standard and Seminorm Adjoints, its just that this is not captured by NFE. We shall include these calculations in the manuscript.
>
> # References
>
> 1. Chen, R.T., Rubanova, Y., Bettencourt, J. and Duvenaud, D.K., 2018. Neural ordinary differential equations. Advances in neural information processing systems, 31. https://arxiv.org/abs/1806.07366
>
> 2. Kidger, P., Chen, R.T. and Lyons, T.J., 2021. " Hey, that's not an ODE": Faster ODE Adjoints via Seminorms. In ICML (pp. 5443-5452). https://arxiv.org/abs/2009.09457

---

### Review · Reviewer_WaJb · 2023-06-19

**Summary Of Contributions:**

This paper introduces a new approach to approximate the reverse-time integral that is needed to compute gradients of a neural ordinary differential equation (NODE). The authors propose to replace standard adjoints with a Gauß–Legendre quadrature. This is doable as the rather complicated reverse integral can be expressed in terms of multiple one-dimensional definite integrals, which are then computed in parallel. On a series of benchmarks, the model is shown to increase speed without hurting model performance.

**Audience:**

Yes

**Broader Impact Concerns:**

Everything alright here.

**Claims And Evidence:**

Yes

**Requested Changes:**

1. I think we need to see more comparisons. For instance, would more complicated NODE variants, e.g. latent NODEs, neural controlled differential equations, heavy-ball NODEs, etc, benefit from the proposed construction?
2. Comparisons against a more competitive baseline would strengthen the paper. The authors claim "regularizing solutions may affect the expressivity of the model". Putting these two together, I'm wondering if, e.g., Jacobian regularization, would help reduce memory footprint or execution time without adversely affecting the model performance.
3. "Preferably, the number of terms chosen adapts to the complexity of the problem, so that the smallest number of terms is used to compute an accurate gradient." Adjoints also can trade off the ODE accuracy with speed. So, would a less accurate adjoint solver (e.g., rtol=1e-2) lead to competitive performance? Related to this, isn't rtol=atol=1e-3 too high compared to the default values in MATLAB and torchdiffeq implementation?
4. "We make the assumption that if the (forward) trajectory of $z$ is complex then the corresponding gradient trajectory $a^T_z \nabla_\theta f$ is also complex". This should be empirically demonstrated.
5. A comparison of $n$ and model performance would be nice.
6. Why do we see memory efficiency on the analytical task but not the others?
7. The approach is tested on neural nets with varying widths but not depths. Why is this?
8. References are needed for the following claims:
    - Solving these ODEs in general can be slow, which is one of the major barriers preventing neural ODEs from being used more widely.
    - "Because the error in $a_\theta$ does not grow significantly compared to $z$ or $a_z$.
    - "Gaussian quadrature is the fastest method for 1-D integrals outside of analytical solutions"

### Minor:
1. "They introduce seminorms to take advantage of this" What is "seminorms"?
2. "Instead we can rewrite the ODE solve in a way that it calculates the gradients associated with the parameters as a definite integral" In what way (6) and (7) are different?
3. A discussion on whether the approach suffers from or helps against vanishing/exploding gradients would be nice.
4. In (9), when $n$ is divided by $t_k-t_0$?
5. What is $m$ in (13)?
6. Does the experiment in Sec 5.1 make sense? The approach is proposed for NODEs whereas the experiment considers a setup that is a lot simpler, for which a much smaller $C$ could be sufficient?

**Strengths And Weaknesses:**

### Strengths
- As far as I understand, the work is theoretically sound.
- The findings clearly favor the proposed approach.

### Weaknesses
- The approach is thoroughly evaluated on standard NODE architectures + benchmarks. I'm curious if more recent NODE variants would also benefit from the approach and how the approach would perform on more challenging setups.
- There are many claims without citations.

---

> ### Author Response · Authors · 2023-06-23
> **Official Response to Reviewer WaJb**
>
> Many thanks for the time and effort in writing a detailed review. We are glad you find the work theoretically sound and that the findings favour the GQ method over the standard adjoint method. We also thank you for the constructive feedback and action points summarised below:
>
> 1. More comparisons with further Neural ODE models
> 2. More comparisons with Neural ODE regularisation schemes
> 3. Higher tolerance for adjoint
> 4. Complexity of $a_z^T\nabla_\theta f$ assumption
> 5. Comparison of $n$ and model performance
> 6. Memory efficiency
> 7. Varying widths vs depths
> 8. References for claims
> 9. Minor points
>
> We happily answer all of these concerns in separate replies below so that discussion around each point is independent. We have also updated the manuscript and so any references in our answer will point to the updated manuscript unless explicitly stated.

---

> > ### Comment · Reviewer_WaJb · 2023-07-07
> > **My final response**
> >
> > Thanks for the detailed response. To summarize my final take on the paper; I have no concerns about the theoretical aspects of the work and its memory efficiency. Yet, the proposed approach is rather complicated and there are several moving pieces; hence the robustness of this setting must be shown clearly and I'm not yet convinced. Putting myself into the shoes of a NODE researcher, I'm not sure if I would train my models with the proposed method. I believe this is the primary deficiency of the work, for which I suggested including more comparisons. More concretely,
> > - The standard NODE model is improved in so many ways that make it necessary for this work to include additional comparisons. Even the original NODE paper introduces a latent model (NODE + VAE), let alone ODE-RNN and controlled NODEs...
> > - Again, if a much simpler approach, such as function regularization as in Finlay et al. 2020, leads to similar performance, I'd choose it over this more complicated formulation. Can we confidently tell this approach excels over simpler mechanisms such as regularization?
> >
> > This being said, if the other reviewers and AC do not share my concerns, I would be happy to lower the tone of my criticisms and see the paper accepted.

---

> > > ### Author Response · Authors · 2023-07-07
> > > **Reply to Reviewer's Response Part 1/2**
> > >
> > > Thank you for the response and the continued effort in reviewing the paper. We address the final concerns below, and as always we are willing to discuss any further concerns you may have.
> > >
> > > # Complicated and Many Moving Pieces
> > > Our paper has taken the existing adjoint method, recognized the presence of many 1D integrals in parallel being solved inefficiently, applied a well known integration technique to solve those integrals more efficiently and used a linear heuristic inside a ceiling function to aid this process. The method can be succinctly summarised as using a simple heuristic that determines how complicated a function is (defined as how high a degree polynomial would be needed to approximate it well), and then using a well-established method that uses a lookup table of function locations and weights to accurately determine a definite integral. We therefore disagree that the method is complicated and has too many moving parts. Whilst there have been no comments on the simplicity of the approach, the existing comments on the method are positive: “The idea is principled, clever, and novel… The authors propose a reasonable empirical heuristic for choosing the number of quadrature points.”
> > >
> > > # Use by NODE Researchers
> > > It’s unfortunate to hear that you would not use this method, since we have provided code in the supplementary and will publish the code after review, it would not be difficult to use our method. In response we respectfully reference the TMLR acceptance criteria (https://jmlr.org/tmlr/acceptance-criteria.html)
> > >
> > > 1. “Are the claims made in the submission supported by accurate, convincing and clear evidence?”
> > > 2. “Would some individuals in TMLR's audience be interested in the findings of this paper?... Crucially, it should **not** be used as a reason to reject work that isn't considered ‘significant’ or ‘impactful’ because it isn't achieving a new state-of-the-art on some benchmark. We explicitly avoid these terms (‘significant’, ‘impactful’, ‘novel’), and focus instead on the notion of ‘interest’. If the authors make it clear that there is something to be learned by some researchers in their area from their work, then the criteria of interest is considered satisfied.”
> > >
> > > Significance is not part of the criteria, and so predicted uptake should not form part of the decision. The main criteria is whether the claims support the evidence, which we believe they have which is also supported by your review. And whether the findings are of interest to the community, we argue that they are, we have introduced a new method to carry out the adjoint method, which tends to be faster, and have opened further directions for future research.
> > >
> > > # Comparison to Regularization
> > > Regularization is an orthogonal approach to ours to improve the speed of Neural ODEs. There is no reason why both cannot be used together to achieve further speedup, and as mentioned, our code will be released which does not require significant implementations, allowing the methods to be easily used together.
> > >
> > > Ultimately there is not really a notion of our method “excelling” over regularization since they are doing different complimentary things. Regularization is a high-level solution, it places restrictions on the dynamics of a Neural ODE which ultimately leads to it being faster. The GQ Adjoint is a low-level method, that increases the speed with which the gradients can be calculated; it places no restrictions on the dynamics. We can compare this to other papers that speed up Neural ODEs by also improving the low-level calculation of the gradients. For example, Daulbaev et al. 2021, Zhuang et al. 2020, Kidger et al. 2021. All of these papers compare to the direct method and the adjoint method but not to regularization schemes because these are all low-level methods to compare to, whereas regularization schemes are orthogonal techniques that are not meaningful to compare against. Another example is Zhuang et al. 2021 which looks more at the exact accuracy of these low level methods, and therefore only compares to the low level methods (direct, adjoint, ACA) and not to regularization schemes.
> > >
> > > To further justify this, consider the comparison with Kinetic Regularization. We need to backpropagate through the baseline method, so we can either use the Direct or Adjoint methods. If the baseline method is faster than the GQ method is it due to the regularization, the backprop method or both? In order to make a fair comparison we have to keep as many things as possible constant between the two methods.

---

> > > > ### Author Response · Authors · 2023-07-07
> > > > **Reply to Reviewer's Response Part 2/2**
> > > >
> > > > # Tests on Advanced Neural ODE Architectures
> > > > We agree that this would provide further evidence in support of our paper. However, we believe these experiments are not necessary. The underlying backbone of all of these advanced architectures is a Neural ODE. Our method will only speed up that specific part of any model, and so rather than using complex models we believe it is more clear to show our method acting on that part directly.
> > > >
> > > > We’d like to make an analogous comparison with the brilliant work AlphaTensor (Fawzi et al. 2022). This work used Reinforcement Learning to discover faster matrix multiplication algorithms than already existed. The paper tests the method directly against existing algorithms. There is no need to test the methods inside Neural Networks, Computer Graphics, etc. since the matrix multiplication is the underlying backbone of the methods. We agree that if the paper showed speed up of these applications that would be very interesting, however, it does not mean that the paper was not convincing or incomplete without these.
> > > >
> > > > # References
> > > > 1. Daulbaev, T., Katrutsa, A., Markeeva, L., Gusak, J., Cichocki, A. and Oseledets, I., 2020. Interpolation technique to speed up gradients propagation in neural odes. Advances in Neural Information Processing Systems, 33, pp.16689-16700.
> > > > 2. Zhuang, J., Dvornek, N., Li, X., Tatikonda, S., Papademetris, X. and Duncan, J., 2020, November. Adaptive checkpoint adjoint method for gradient estimation in neural ode. In International Conference on Machine Learning (pp. 11639-11649). PMLR.
> > > > 3. Kidger, P., Chen, R.T. and Lyons, T.J., 2021. " Hey, that's not an ODE": Faster ODE Adjoints via Seminorms. In ICML (pp. 5443-5452).
> > > > 4. Zhuang, J., Dvornek, N.C., Tatikonda, S. and Duncan, J.S., 2021. Mali: A memory efficient and reverse accurate integrator for neural odes. In International Conference on Learning Representations.
> > > > 5. Fawzi, A., Balog, M., Huang, A., Hubert, T., Romera-Paredes, B., Barekatain, M., Novikov, A., R Ruiz, F.J., Schrittwieser, J., Swirszcz, G. and Silver, D., 2022. Discovering faster matrix multiplication algorithms with reinforcement learning. Nature, 610(7930), pp.47-53.

---

> ### Author Response · Authors · 2023-06-23
> **1. More Comparisons with further Neural ODE models**
>
> All of the models suggested use a Neural ODE solve as the underlying forward propagation (or at least part of it). Our claim is that we make the adjoint method of backpropagation faster but we do not claim that the models’ performances will improve. In our work, we have tested an Augmented Neural ODE (Dupont et al. 2019), which relies only on the final state, which is similar to a Neural CDE (for time-series classification), but it just has a different dynamics function. We test a Second Order Neural ODE (Norcliffe et al. 2020), in a time-series task, which is a generalisation of Heavy Ball Neural ODEs (the damping term can be learnt rather than enforced). Finally we test on image classification, using an encoder to run dynamics in a latent space, which is similar to a Latent Neural ODE, which also encodes (with an ODE-RNN, not single network) and decodes with a Neural ODE. For this reason we believe we have explored sufficient cases and further comparisons would not add more insights.
>
> # References
>
> 1. Dupont, E., Doucet, A. and Teh, Y.W., 2019. Augmented neural odes. Advances in neural information processing systems, 32.
> 2. Norcliffe, A., Bodnar, C., Day, B., Simidjievski, N. and Liò, P., 2020. On second order behaviour in augmented neural odes. Advances in neural information processing systems, 33, pp.5911-5921.

---

> ### Author Response · Authors · 2023-06-23
> **2. More Comparisons with regularisation schemes**
>
> Whilst we agree investigating additional regularisation schemes would compliment the paper, ultimately it is not what the paper is about and we do not want to dilute the main message of the paper. Regularisation affects the trajectory taken whereas we wanted to investigate if we can speed up Neural ODEs without adding regularisation. We are confident regularisation would only improve the speed further compared to the adjoint method. We see that regularisation is able to improve time taken, for example in STEER: Simple Temporal Regularization for Neural ODE (Ghosh et al. 2020) and How to Train your Neural ODE (Finlay et al. 2020), but this is because the trajectory is made simpler. We investigated a lower level of speed up, that is can we solve the adjoint equations faster prior to regularisation and we found that we can.
>
> # References
>
> 1. Ghosh, A., Behl, H., Dupont, E., Torr, P. and Namboodiri, V., 2020. Steer: Simple temporal regularization for neural ode. Advances in Neural Information Processing Systems, 33, pp.14831-14843.
> 2. Finlay, C., Jacobsen, J.H., Nurbekyan, L. and Oberman, A.M., 2020. How to train your neural ode. arXiv preprint arXiv:2002.02798.

---

> ### Author Response · Authors · 2023-06-23
> **3. Higher Tolerance for Standard Adjoint**
>
> Indeed the standard adjoint also adapts to the problem. However, the GQ method still uses the same solver as the standard adjoint for the forward solve of $z$ and backward solve of $[z, a_z]$. Using a higher tolerance would also allow the GQ method to adapt to the problem in the same way. If we were to use a higher tolerance, then the GQ method would use $\big \lceil \frac{\text{NFE}}{10} \big \rceil$ terms (assuming we are dealing with a problem that isn’t a time-series). So provided the adjoint uses more than two function evaluations we can still expect to be faster (one for the quadrature point and one for the ODE solve). We also discuss in detail the number of function evaluations in Appendix E.1 which demonstrates that as the standard adjoint adapts to the problem, the GQ method adapts in the same way to still be faster (this can also be seen in responses 7 to Reviewer rgHZ and 5 to Reviewer 4ho2 for convenience).
>
> Regarding the tolerance used in the experiments, the tolerance needed depends on the problem. And we see in all the tasks we have strong performance, therefore, whilst higher than the default the tolerance we used is sufficient. We also see in other papers that this is the case, for example (Zhuang et al. 2021) use tolerances of 1.0, 0.1, 0.01.
>
> # References
>
> 1. Zhuang, J., Dvornek, N.C., Tatikonda, S. and Duncan, J.S., 2021. Mali: A memory efficient and reverse accurate integrator for neural odes. arXiv preprint arXiv:2102.04668.

---

> ### Author Response · Authors · 2023-06-23
> **4. Complexity of $a_z^T\nabla_\theta f$ assumption**
>
> We believe that this is not necessary, since our assumption is actually imposing that we use more terms than might be needed. The assumption is used in the heuristic for number of terms and we assume a complex forward trajectory of $z$ leads to a complex backward trajectory of $a_z^T\nabla_\theta f$, if we instead change the assumption to assume we have a simple backward trajectory then we can use fewer terms. However, we want to make sure we use at least as many terms as required for gradient accuracy. The question then becomes does this assumption lead to slower or worse performance, and our experiments show that it does not, as stated in the review “The findings clearly favor the proposed approach”. The heuristic works and therefore we argue that even though the assumption might lead to more terms than needed, we are still able to be faster than the standard adjoint.
>
> Further to this, the existing literature supports this assumption. For example the original Neural ODE paper (Chen et al. 2018) shows that the backward number of function evaluations is approximately half the number of forward function evaluations. So as the forward NFE increases the backward NFE also increases. Also in Hey that’s not an ODE (Kidger et al. 2021) the number of backward NFE increases during training, to show as the complexity of the forward solve increases during training, the backward solve also becomes more complex.
>
> # References
>
> 1. Chen, R.T., Rubanova, Y., Bettencourt, J. and Duvenaud, D.K., 2018. Neural ordinary differential equations. Advances in neural information processing systems, 31.
> 2. Kidger, P., Chen, R.T. and Lyons, T.J., 2021. " Hey, that's not an ODE": Faster ODE Adjoints via Seminorms. In ICML (pp. 5443-5452).

---

> ### Author Response · Authors · 2023-06-23
> **5. Comparison of $n$ and model performance**
>
> In the Appendix there is an ablation study exploring the accuracy vs $n$ tradeoff. In the previous version of the manuscript it was in Appendix B.1 in the most recent version it is in Appendix E.2. The paragraph is titled "Effect of C" and is an ablation looking at the effect of the constant $C$ in the heuristic, which effectively chooses how many points to use in the quadrature, larger $C$ leads to more points. In Table 2 we investigate the effect of $C$ on the test accuracy and training time for the Nested Spheres task. For convenience we include this table below:
>
> | C                          | Time to train (/s) | Test Accuracy (%) |
> | ----------------------- | ----------------------|--------------------------|
> |$1 \times 10 ^{-3}$| $61.4 \pm 3.0 $ | $79.0 \pm 26.9$    |
> |$1 \times 10 ^{-1}$| $89.8 \pm 3.1 $ | $95.6 \pm 8.8$      |
> |$1 \times 10 ^{2}$| $141.0 \pm 2.4 $ | $95.8  \pm 8.4$    |
> |$1 \times 10 ^{5}$| $141.7  \pm 2.4$ | $95.8 \pm 8.4$    |
>
> We see that the time to train is lowest when C is lowest, because the fewest terms are used. However we also see that the final accuracy is compromised as a result. This is what we would expect, the number of terms used in the integration is not enough so the gradients are inaccurate. We see that using a very high C leads to a slightly higher accuracy than when C is 0.1. However, this is insignificant compared to the increase in training time, and the accuracies are easily within a standard deviation of each other. There is no increase in training time between C being $1 \times 10^2$ and $1 \times 10^5$ because the number of terms in the quadrature is manually capped at 64 (which can integrate a polynomial of degree 127).

---

> ### Author Response · Authors · 2023-06-23
> **6. Memory Efficiency**
>
> There is memory efficiency in all tasks, due to using a running total of the quadrature sum as described. It is the case that we show it only on the analytical task. We see implicitly that we have memory efficiency in the SDE task, since the direct backpropagation method fails due to memory limits, whereas the adjoint and GQ method do not fail. We showed it on the analytical task simply to demonstrate that as more function evaluations are used the GQ and standard adjoint method use constant memory, whereas the direct and ACA methods which backpropagate through solver operations do not use constant memory. This is in contrast to other papers that claim memory efficiency ($\mathcal{O}(1)$) but do not explicitly record any memory usage, for example:
>
> - The Reversible Residual Network (Gomez et al. 2017)
> - Neural Ordinary Differential Equations (Chen et al. 2018) (the original Neural ODE paper)
> - Scalable Gradients for Stochastic Differential Equations (Li et al. 2020)
> - Interpolation Technique to Speed up Gradients Propagation in Neural ODEs (Daulbaev et al. 2020)
> - Efficient and Accurate Gradients for Neural SDEs (Kidger et al. 2021)
>
> The only paper to our knowledge that both claims and explicitly demonstrates memory efficiency is MALI: A memory efficient and reverse accurate integrator for Neural ODEs (Zhuang et al. 2021), and there it is only shown for three points on the analytical task.
>
> The main takeaway from the figure is that we are memory efficient, and this is both shown in the figure and based on the algorithm for calculating the gradient itself. We have also updated the discussion around this figure in the new manuscript.
>
> # References
>
> 1. Gomez, A.N., Ren, M., Urtasun, R. and Grosse, R.B., 2017. The reversible residual network: Backpropagation without storing activations. Advances in neural information processing systems, 30.
> 2. Chen, R.T., Rubanova, Y., Bettencourt, J. and Duvenaud, D.K., 2018. Neural ordinary differential equations. Advances in neural information processing systems, 31.
> 3. Li, X., Wong, T.K.L., Chen, R.T. and Duvenaud, D.K., 2020, February. Scalable gradients and variational inference for stochastic differential equations. In Symposium on Advances in Approximate Bayesian Inference (pp. 1-28). PMLR.
> 4. Daulbaev, T., Katrutsa, A., Markeeva, L., Gusak, J., Cichocki, A. and Oseledets, I., 2020. Interpolation technique to speed up gradients propagation in neural odes. Advances in Neural Information Processing Systems, 33, pp.16689-16700.
> 5. Kidger, P., Foster, J., Li, X.C. and Lyons, T., 2021. Efficient and accurate gradients for neural SDEs. Advances in Neural Information Processing Systems, 34, pp.18747-18761.
> 6. Zhuang, J., Dvornek, N.C., Tatikonda, S. and Duncan, J.S., 2021. Mali: A memory efficient and reverse accurate integrator for neural odes. arXiv preprint arXiv:2102.04668.

---

> ### Author Response · Authors · 2023-06-23
> **7. Varying Width vs Depth**
>
> We vary the width so that we can see the effect having more parameters has on the speed of training. This can also be achieved by varying the depth as you mention. This is inconsequential, there was no particular reason for choosing width apart from the fact it is slightly easier to control the number of parameters for the tasks: increasing width from $n$ to $n+1$ increases number of parameters roughly by $2n$ (with two hidden layers), whereas adding another hidden layer adds $n^2$ new parameters. So we can increase the number of parameters at a slower rate if desired by only varying one hyperparameter.

---

> ### Author Response · Authors · 2023-06-23
> **8.1 References for Claims - ODE Speed**
>
> “Solving these ODEs in general can be slow, which is one of the major barriers preventing neural ODEs from being used more widely.”
>
> This point was generally made to put the use of Neural ODEs into context and why it is important to work on speeding them up. It has proven difficult to find references that say this is why Neural ODEs are not used as widely. We argue that ultimately the proof is more in the **relative** lack of citations (currently the original Neural ODE paper (Chen et al. 2018) is at 3233) compared to works that have been significantly used such as Residual Networks (He et al. 2016) (at 169237 citations), Transformers (Vaswani et al. 2017) (at 78904 citations) and Adam Optimizer (Kingma and Ba 2014) (at 147868 citations). These figures were taken from Google Scholar on 22/06/2023. **We appreciate that this argument is flawed and not doing full justice to Neural ODEs, we will happily remove this phrase if the reviewer desires and change it to say that speeding up Neural ODEs is important and a large area of research.** We are able to find many papers that work on improving the speed of Neural ODEs, and claim that they are slow, for example:
>
> 1. Accelerating Neural ODEs Using Model Order Reduction (Lehtimäki et al. 2022)
> 2. Improving Neural Ordinary Differential Equations with Nesterov's Accelerated Gradient Method (Nguyen et al. 2022)
> 3. Hey that’s not an ODE (Kidger et al. 2021)
> 4. Adaptive Checkpoint Neural ODE (Zhuang et al. 2020)
> 5. Interpolation Technique to Speed Up Gradients Propagation in Neural ODEs (Daulbaev et al. 2020)
> 6. STEER: Simple Temporal Regularization (Ghosh et al. 2020)
> 7. How to Train your Neural ODE (Finlay et al. 2020)
>
> The majority of these papers are already cited in the paper, we shall add the rest.
>
> # References
>
> 1. Chen, R.T., Rubanova, Y., Bettencourt, J. and Duvenaud, D.K., 2018. Neural ordinary differential equations. Advances in neural information processing systems, 31.
> 2. He, K., Zhang, X., Ren, S. and Sun, J., 2016. Deep residual learning for image recognition. In Proceedings of the IEEE conference on computer vision and pattern recognition (pp. 770-778).
> 3. Vaswani, A., Shazeer, N., Parmar, N., Uszkoreit, J., Jones, L., Gomez, A.N., Kaiser, Ł. and Polosukhin, I., 2017. Attention is all you need. Advances in neural information processing systems, 30.
> 4. Kingma, D.P. and Ba, J., 2014. Adam: A method for stochastic optimization. arXiv preprint arXiv:1412.6980.
> 5. Lehtimäki, M., Paunonen, L. and Linne, M.L., 2022. Accelerating neural odes using model order reduction. IEEE Transactions on Neural Networks and Learning Systems.
> 6. Nguyen, H.H.N., Nguyen, T., Vo, H., Osher, S. and Vo, T., 2022. Improving Neural Ordinary Differential Equations with Nesterov's Accelerated Gradient Method. Advances in Neural Information Processing Systems, 35, pp.7712-7726.
> 7. Kidger, P., Chen, R.T. and Lyons, T.J., 2021. " Hey, that's not an ODE": Faster ODE Adjoints via Seminorms. In ICML (pp. 5443-5452).
> 8. Zhuang, J., Dvornek, N., Li, X., Tatikonda, S., Papademetris, X. and Duncan, J., 2020, November. Adaptive checkpoint adjoint method for gradient estimation in neural ode. In International Conference on Machine Learning (pp. 11639-11649). PMLR.
> 9. Daulbaev, T., Katrutsa, A., Markeeva, L., Gusak, J., Cichocki, A. and Oseledets, I., 2020. Interpolation technique to speed up gradients propagation in neural odes. Advances in Neural Information Processing Systems, 33, pp.16689-16700.
> 10. Ghosh, A., Behl, H., Dupont, E., Torr, P. and Namboodiri, V., 2020. Steer: Simple temporal regularization for neural ode. Advances in Neural Information Processing Systems, 33, pp.14831-14843.
> 11. Finlay, C., Jacobsen, J.H., Nurbekyan, L. and Oberman, A.M., 2020. How to train your neural ode. arXiv preprint arXiv:2002.02798.

---

> ### Author Response · Authors · 2023-06-23
> **8.2 Error growth in $a_\theta$**
>
> “Because the error in $a_\theta$ does not grow significantly compared to $z$ or $a_z$”
>
> The citation here is from Hey that’s Not an ODE (Kidger et al. 2021):
>
> “This means that (conditioned on knowing $z$ and $a_z$), the integral corresponding to $a_\theta$ is just an integral - not an ODE. As such, it is arguably inappropriate to solve it with an ODE solver, which makes the implicit assumption that small errors now may propagate to create large errors later.”
>
> We realise that despite our sentence being in the same paragraph describing the work by Kidger et al. 2021 it is not immediately clear that the two are related and will rephrase the paragraph to improve this.
>
> # References
>
> 1. Kidger, P., Chen, R.T. and Lyons, T.J., 2021. " Hey, that's not an ODE": Faster ODE Adjoints via Seminorms. In ICML (pp. 5443-5452).

---

> ### Author Response · Authors · 2023-06-23
> **8.3 Speed of Gaussian Quadrature**
>
> “Gaussian quadrature is the fastest method for 1-D integrals outside of analytical solutions”
>
> This assumes that an analytical solution is made of elementary functions that are fast to calculate such as sine, cosine, exponential etc. This would trivially be the fastest solution. Regarding the speed of Gaussian Quadrature compared to other numerical methods, we agree this needs further justification. The following discussion is in the updated manuscript in Appendix D with reference to it in the main text. The information was taken from the textbook Introduction to Numerical Analysis (Stoer and Bulirsch 1993).
>
> We dissect three integration techniques where we solve a separate integral for each parameter, giving us many one dimensional integrals in parallel rather than one high dimensional integral. Assume we wish to solve the integral given by $\int_a^bf(x)dx$.
>
> # Newton-Cotes
> Newton-Cotes integration techniques split the interval $[a, b]$ into $N$ **equidistant** intervals. Then a polynomial of a given degree is fit on each interval which can be integrated. Classic examples are:
>
> - Trapezoid rule, this fits a linear interpolant to each interval. The error is $\mathcal{O}\big(\frac{f^{2}(\xi)}{N^2}\big)$, where $f^{m}(\xi)$ refers to the largest $m$-th derivative on the interval $[a, b]$ located at $\xi$.
> - Simpson's rule, this fits quadratic interpolants on the intervals. The error is $\mathcal{O}\big(\frac{f^{4}(\xi)}{N^4}\big)$.
> - Boole's rule, this fits quartic interpolants on the intervals. The error is $\mathcal{O}\big(\frac{f^{6}(\xi)}{N^6}\big)$.
>
> In the above, $f^m(\xi)$ is a constant, it has been included in the error complexity to show that if the true function is exactly a polynomial of a given degree the error can be zero. For example if the function is exactly linear the second derivative everywhere is zero, so the Trapezoid and other Newton-Cotes rules give the exact results. However the crucial point is that these are still constant with respect to $N$ so the errors for the Trapezoid, Simpson and Boole's rules are $\mathcal{O}\big( \frac{1}{N^2} \big)$, $\mathcal{O}\big( \frac{1}{N^4} \big)$ and $\mathcal{O}\big( \frac{1}{N^6} \big)$. Therefore for Newton-Cotes the error is $\mathcal{O}\big( \frac{1}{N^k} \big)$ where $k$ depends on the rule for interpolation.
>
> # Gaussian Quadrature
> Gaussian  Quadrature **does not use equidistant points**, the points are given by principled locations on the interval - the roots of orthogonal polynomials which we can look up in a table. This allows us to approximate a $2N-1$ degree polynomial with only $N$ points in the quadrature sum. The error using Gaussian Quadrature with weight function $1$ is bounded by $\frac{(b-a)^{2N+1}(N!)^4}{(2N+1)((2N)!)^3}f^{2N}(\xi)$. There are two key terms in this error, the first is $f^{2N}(\xi)$, as we use more points we model a higher degree polynomial unlike Newton-Cotes which stays constant. As mentioned we are therefore already able to exactly solve a polynomial of degree $2N-1$, so if the function is well approximated by such a polynomial we will have an accurate approximation of the integral. The other key term is $((2N)!)^3$ in the denominator, which dominates the other terms in the error bound showing the error will shrink very quickly with $N$, significantly faster than if the error is $\mathcal{O}\big( \frac{1}{N^k} \big)$.
>
> Please note that the above assumes that $f$ is differentiable, if there are discontinuities in $f$ the integral is split up to make up for that. We elaborate on this in Appendix A where we describe how we adapt the GQ method to time-series.
>
> # Monte-Carlo Integration
> Monte-Carlo methods approximate the integral by sampling many points and calculating the mean of these. $\int_a^bf(x)dx = \int_a^bp(x)\frac{f(x)}{p(x)}dx \approx \frac{1}{N}\sum_{i=1}^{N}\frac{f(x_i)}{p(x_i)} \quad x_i \sim p(x)$. By sampling the points they are **not equidistant but also not in principled locations**. The distribution of these results follows a normal distribution with the mean being the true integral and the variance is $\mathcal{O}\big( \frac{1}{N} \big)$, so the error using Monte-Carlo is $\mathcal{O}\big( \frac{1}{\sqrt{N}} \big)$. Monte-Carlo has the worst error rate of the described methods which is why it is only used for high dimensional integrals where it is not possible to apply the other methods.
>
> Therefore we use Gaussian Quadrature for these integrals since it has the best error rate. This information can be found in Chapter 3 of Introduction to Numerical Analysis (Stoer and Bulirsch 1993).
>
> # References
> 1. Stoer, J., Bulirsch, R., Stoer, J. and Bulirsch, R., 1993. Topics in Integration. Introduction to Numerical Analysis, pp.125-166.

---

> ### Author Response · Authors · 2023-06-23
> **9. Minor Points Part 1/2**
>
> 1. What are seminorms: As introduced in Kidger et al. 2021, seminorms are where the error estimate of the solver during the backward solve only takes into account the error propagation of $[z, a_z]$, and not $[z, a_z, a_\theta]$ as is standard. We have explained this in the subsequent sentence in the original manuscript “During the backwards solve, the ODE solver does not consider $a_\theta$ when calculating the error to choose a step size, because the error in $a_\theta$ does not grow significantly compared to $z$ or $a_z$”. Which is justified by Kidger et al. 2021 as stated in the previous response. As stated in the previous response we shall rephrase this paragraph to improve the clarity.
>
> 2. Difference between (6) and (7): The final line of (6) gives the gradient, $\frac{\partial L}{\partial \theta} = a_{\theta}(t_0)$, where $a_\theta(t_K)=0$ and $\frac{d a_\theta}{dt} = -a_z^T \nabla_\theta f$. And since the differential equation for $a_\theta$ does not contain $a_\theta$, the gradient is given by $0 + \int_{t_K}^{t_0} - a_z^T\nabla_\theta f dt$, which is equal to $\int_{t_0}^{t_K} a_z^T \nabla_\theta f dt$, which is (7). There is no significant difference, the point of (7) is to highlight that this is an integral and can be solved with numerical integration techniques not only ODE solvers. As well as this it highlights the connection to Optimal Control Theory. In optimal control theory we have a dynamical system given by $\frac{dz}{dt} = f(z, t, \theta(t))$ where $\theta(t)$ is a control that we choose, for example the thrust of an airplane $\theta(t)$ controls the height $z(t)$ according to a differential equation given by Newton's Laws. Given some loss at the final time we solve the adjoining system $\lambda(t_K) = \frac{\partial L}{\partial z(t_K)}$ and $\frac{d \lambda}{dt} = -\lambda^T \frac{\partial f}{\partial z}$. The method of steepest descent is used to update the control so that $\theta(t)$ is updated to $\theta(t) - \eta \lambda(t)^T\frac{\partial f}{\partial \theta}(t)$, for small step size $\eta$. In the case of Neural ODEs the control is given by the parameters which are a constant, and so we sum all of the local changes to construct an integral, $\theta$ becomes $\theta - \eta \int_{t_0}^{t_K}\lambda^T(t)\frac{\partial f}{\partial \theta}(t) dt$.
>
> 3. Vanishing/Exploding Gradients: We do not believe a discussion around vanishing and exploding gradients is necessary, since the work is around changing the way the gradients are calculated to be faster. We agree it would be relevant if the work was about changing the Neural ODE architecture or the form of the dynamics to improve the vanishing/exploding gradient problem. To the best of our knowledge the only work that looks into this is Improving Neural Ordinary Differential Equations with Nesterov's Accelerated Gradient Method (Nguyen et al. 2022). One explanation for the lack of work on vanishing/exploding gradients in Neural ODEs is that they can be thought of as “infinitely deep” residual networks which can avoid the problem of vanishing/exploding gradients.  This is done by viewing the Neural ODE as an Euler discretisation $z_{t+\Delta t} = z_t + f_\theta(z_t, t) \Delta t$ in the infinitesimal limit. This shows that a Neural ODE can be thought of as a Residual Network with infinitesimal step sizes, and there is evidence to suggest Residual Networks may avoid vanishing/exploding networks due to the skip connections (see Veit et al. 2016 or  Zaeemzadeh et al. 2020 for example).
>
> 4. In (9) when $n$ is divided by $t_K - t_0$: $n$ is always divided by $t_K - t_0$. If the question is why, we can answer that. The number of terms is given by $\big \lceil \frac{\text{NFE}}{10}\frac{t_k - t_{k-1}}{t_{K} - t_{0}} \big \rceil$. In the case where the loss does not depend on intermediate times this simplifies to $\big \lceil \frac{\text{NFE}}{10}\big \rceil$. In the case of time-series we would like an estimate of how complex the trajectory was in the time range $[t_{k-1}, t_k]$, so that we can use a suitable number of terms in the Quadrature sum at that point. This is done by taking the fraction $\frac{t_k - t_{k-1}}{t_{K} - t_{0}}$ to determine what fraction of the total time range is of interest, then multiply by NFE assuming the function evaluations are spread evenly across the solve (which might not be entirely true but we see that this works in our time-series experiments). We then divide by 10 in accordance with the heuristic to use approximately NFE/10 terms in the quadrature sum.

---

> ### Author Response · Authors · 2023-06-23
> **Minor Points Part 2/2**
>
> 5. What is $m$ in (13): The Wong-Zakai theorem says we can approximate a Weiner process with a smooth function $B_m(t)$, where $m$ is an integer that defines the approximation, e.g. $B_m$ can be a polynomial with degree $m$, the theorem states that if $m \xrightarrow{} \infty$ causes $B_m$ to approach a Weiner process, then the solution of the ODE is a solution of the SDE. In our case $m$ determines how many cosines to use in the Karhunen-Loeve approximation of a Weiner process given by equation 14. The subscript $m$ in (13) is to show this is an approximation determined by the integer $m$. We shall edit the phrasing around (13) to make this more clear.
>
> 6. Experiment 5.1: The aim of Figure 1 and the analytical system is to demonstrate our claims on a system that we can control, where we have full knowledge of the true loss and gradients. Those claims are that the GQ Adjoint will produce the same gradients as the Standard Adjoint only faster. So we are looking for the errors in the loss and gradients to match those given by the Standard Adjoint, not be lower than them (we do not assess the speed here).
>
> The system is one of exponential growth (proposed by Zhuang et al. 2021) with state $z$, the dynamics are given by $\frac{dz}{dt} = az$, this gives the solution $z(t) = z_0 \exp(at)$, where $a$ is a parameter determining the growth and $z_0$ is a parameter giving the initial condition. We integrate up to time $T$ and use the loss $L = z(T)^2$ which is $z_0^2 \exp(2aT)$. We use this system since we have exact analytical solutions and we can test the method when finding gradients with respect to:
>
> - Dynamics parameters: $a$
> - Initial conditions: $z_0$
> - Solution times: $T$
>
> Our method supports differentiation with respect to all of these to match the Direct and Standard Adjoint's capabilities. The gradients are given by $\frac{\partial L}{\partial a} = 2Tz_0^2\exp(2aT)$, $\frac{\partial L}{\partial z_0} = 2z_0 \exp(2aT)$ and $\frac{\partial L}{\partial T} = 2az_0^2 \exp(2aT)$.
>
> Figure 1 plots how the errors between predicted values and analytical values (given above) change as integration time $T$ increases. We actually plot the relative error $\bigg| \frac{\text{True} - \text{Predicted}}{\text{True}} \bigg|$, we do this because the state is exponentially increasing, so by showing the relative error we see the differences for small $T$ as well as large $T$.
>
> We see that the errors in the loss are the same for the Direct, Standard Adjoint and GQ Adjoint, this is what we expect since the loss only depends on the forward solve and they all use the same method for the forward solve. We expect the errors **in the gradients** between the Standard/GQ Adjoint methods and the Direct method to be different, this is because we are directly seeing the difference between the two approaches, adjoint being optimize then discretize, direct being discretize then optimize. This is indeed what we see. However, we do expect the Standard Adjoint and GQ Adjoint methods to produce the same gradients since they are calculating the same integral only in different ways (with the GQ method being faster). This is our claim, that we obtain the same gradients only faster, and this is indeed what we see in the three relative error in gradient plots. The discussion around the figure has been updated and the figure has had minor aesthetic changes in the updated manuscript.
>
> # References
>
> 1. Kidger, P., Chen, R.T. and Lyons, T.J., 2021. " Hey, that's not an ODE": Faster ODE Adjoints via Seminorms. In ICML (pp. 5443-5452).
> 2. Nguyen, H.H.N., Nguyen, T., Vo, H., Osher, S. and Vo, T., 2022. Improving Neural Ordinary Differential Equations with Nesterov's Accelerated Gradient Method. Advances in Neural Information Processing Systems, 35, pp.7712-7726.
> 3. Veit, A., Wilber, M.J. and Belongie, S., 2016. Residual networks behave like ensembles of relatively shallow networks. Advances in neural information processing systems, 29.
> 4. Zaeemzadeh, A., Rahnavard, N. and Shah, M., 2020. Norm-preservation: Why residual networks can become extremely deep?. IEEE transactions on pattern analysis and machine intelligence, 43(11), pp.3980-3990.
> 5. Zhuang, J., Dvornek, N.C., Tatikonda, S. and Duncan, J.S., 2021. Mali: A memory efficient and reverse accurate integrator for neural odes. In International Conference on Learning Representations 2021. https://arxiv.org/abs/2102.04668
> 6. Zhuang, J., Dvornek, N., Li, X., Tatikonda, S., Papademetris, X. and Duncan, J., 2020, November. Adaptive checkpoint adjoint method for gradient estimation in neural ode. In International Conference on Machine Learning (pp. 11639-11649). PMLR. https://arxiv.org/abs/2006.02493

---

### Author Response · Authors · 2023-06-10
**Updated Manuscript**

To all reviewers,

Many thanks for taking the time and effort to both review our paper and actively engage in discussion. We have now updated the manuscript with the following changes (in order of how they appear in the paper):

- Changed the introduction to not say that the direct method is "fast and memory intensive" but "faster and more memory intensive than the adjoint", i.e. we compare them rather than make generic statements (Reviewer rgHZ)
- We have removed the statement in Section 2 about computational graphs growing arbitrarily and replaced this with the number of function evaluations (Reviewer rgHZ)
- Changed the statement about integration being "potentially more accurate" in Section 2 to "being the same level of accuracy" (Reviewer rgHZ)
- Added a few lines to Section 3.1 "Why Gauss-Legendre Quadrature?" to explain how weight functions are used more in general quadrature (Reviewer rgHZ)
- Removed ambiguity about the running total and terms in the running sum in Section 3.2 to specifically say that $g$ is initialised as a vector of zeros, and at each quadrature point $g$ is updated to $g + a_z^T \nabla_{\theta}f$ (Reviewer rgHZ)
- Explained Figure 1 to include the information in the author responses (Reviewers rgHZ and 4ho2)
- Updated Figure 1 so that it is larger, the time-series is downsampled and the linestyles are changed to make the results more clear (Reviewer 4ho2)
- Added a sentence about Figure 2 to explain why the GQ method has a slightly higher constant memory than the standard adjoint (Reviewer 4ho2)
- Added a description of the training of SDEs in Appendix B (Reviewer 8wu8)
- Added a description of the adjoint vs direct methods in Appendix C to explain the computational and memory complexities of the two methods, and how care must be taken since the gradients calculated can differ (Reviewer rgHZ)
- Added information about numerical integration in Appendix D, including information about error bounds of different methods (Reviewer 4ho2) and how we can shift from a general $[a, b]$ domain to $[-1, 1]$ to use Gaussian Quadrature (Reviewer 8wu8)
- Added information about the expected number of function evaluations in Appendix E.1 (Reviewers rgHZ, 4ho2, 8wu8)
- Added a new experiment based on the crossing trajectories task in Appendix G (Reviewer 4ho2)

We have also made all of our responses available to read, we apologize that this was not done before, this was a mistake with who was made readers of our comments, this is now fixed.

---

### Author Response · Authors · 2023-06-23
**Second Manuscript Update**

To all reviewers,

Many thanks again for the effort in reviewing our paper. In accordance with requests from Reviewer WaJb we have updated the manuscript again with the following changes:

- Added citation in related work (Xia et al. 2021, Nguyen et al. 2022, Lehtimäki et al. 2022, Onken et al. 2021)
- Removed the sentence claiming speed is a reason Neural ODEs are not as widely used as they could be, and said that solving ODEs can be slow and thus an important research topic
- Rephrased the paragraph in Section 2 about Seminorms to improve clarity
- Added a clarifying sentence to describe the use of $m$ in equation 13

# References

1. Xia, H., Suliafu, V., Ji, H., Nguyen, T., Bertozzi, A., Osher, S. and Wang, B., 2021. Heavy ball neural ordinary differential equations. Advances in Neural Information Processing Systems, 34, pp.18646-18659.
2. Nguyen, H.H.N., Nguyen, T., Vo, H., Osher, S. and Vo, T., 2022. Improving Neural Ordinary Differential Equations with Nesterov's Accelerated Gradient Method. Advances in Neural Information Processing Systems, 35, pp.7712-7726.
3. Lehtimäki, M., Paunonen, L. and Linne, M.L., 2022. Accelerating neural odes using model order reduction. IEEE Transactions on Neural Networks and Learning Systems.
4. Onken, D., Fung, S.W., Li, X. and Ruthotto, L., 2021, May. Ot-flow: Fast and accurate continuous normalizing flows via optimal transport. In Proceedings of the AAAI Conference on Artificial Intelligence (Vol. 35, No. 10, pp. 9223-9232).

---

### Decision · Action_Editors · 2023-07-28

**Recommendation:** Accept as is

**Comment:**

The paper presents a new way of looking at training neural ODEs using the adjoint method with demonstrable improvements in efficiency. There were questions about the practical benefits of the adjoint method itself that were largely resolved during the discussion period. While experiments that test alternative NODE variations and regularization would be very interesting, the approach itself was deemed to be relevant and sound in terms of the TMLR criteria.

**Audience:**

Yes, neural ODEs and SDEs are relevant to the community, more recently with diffusion models. This work provides a different perspective on training that will be of interest to some members of the community. The reviewers agree with this.

**Claims And Evidence:**

Yes, the reviewers for the most part believe this is true. One reviewer does believe that more work should be done to compare against improvements to basic NODEs and that comparisons should be done against regularization.

Other reviewers were not concerned with this, seeing as regularization achieves speedups by reducing model capacity. The authors also pointed out that regularization is complementary.

More experiments can always help, but there does need to be a limit. The other reviewers were happy with the experiments (including the additional results in the discussion) and the more critical reviewer did not see these points as a strong negative.

---

> ### Author Response · Authors · 2023-08-21
> **Many thanks for the effort to review**
>
> To the reviewers and action editor,
>
> As the authors, we'd like to offer our sincere thanks to the reviewers and action editor for their time and effort in reviewing the paper. We are grateful for the feedback which has improved the quality of the paper.